# PENG'S Q(λ) FOR CONSERVATIVE VALUE ESTIMATION IN OFFLINE REINFORCEMENT LEARNING

**Byeongchan Kim**
Seoul National University
bckim97@snu.ac.kr

**Min-hwan Oh**[*]
Seoul National University
minoh@snu.ac.kr

## ABSTRACT

We propose a model-free offline multi-step reinforcement learning (RL) algorithm, Conservative Peng's Q(λ) (CPQL). Our algorithm adapts the Peng's Q(λ) (PQL) operator for conservative value estimation as an alternative to the Bellman operator. To the best of our knowledge, this is the first work in offline RL to theoretically and empirically demonstrate the effectiveness of conservative value estimation with a *multi-step* operator by fully leveraging offline trajectories. The fixed point of the PQL operator in offline RL lies closer to the value function of the behavior policy, thereby naturally inducing implicit behavior regularization. CPQL simultaneously mitigates over-pessimistic value estimation, achieves performance greater than (or equal to) that of the behavior policy, and provides near-optimal performance guarantees — a milestone that previous conservative approaches could not achieve. Extensive numerical experiments on the D4RL benchmark demonstrate that CPQL consistently and significantly outperforms existing offline single-step baselines. In addition to the contributions of CPQL in offline RL, our proposed method also contributes to the offline-to-online learning framework. Using the Q-function pre-trained by CPQL in offline settings enables the online PQL agent to avoid the performance drop typically observed at the start of fine-tuning and to attain robust performance improvements. Our code is available at https://github.com/oh-lab/CPQL.

## 1 INTRODUCTION

*Offline* RL aims to learn policies from a static dataset collected under unknown behavior policies without further interactions with the actual environment. However, offline RL faces a major challenge known as distributional shift (Levine et al., 2020). A distributional shift arises when the state-action distribution under the learned policy diverges significantly from that under the behavior policy. This issue is exacerbated when the application of the Bellman updates to value functions requires querying values of out-of-distribution (OOD) state-action pairs, which can lead to an accumulation of extrapolation errors and ultimately result in poor performance of learned policies.

To tackle OOD actions in policy evaluation, conservative Q-learning (CQL) (Kumar et al., 2020) penalizes the learned Q-function for OOD actions induced by the learning policy. Building on CQL, subsequent algorithms (Ma et al., 2021; Lyu et al., 2022; Chen et al., 2023; Nakamoto et al., 2023; Shao et al., 2023; Yeom et al., 2024) address the potential over-pessimism in both in-distribution and OOD actions, which stems from excessively conservative value estimates. These approaches rely on additional components, such as estimating the unknown behavior policy to handle OOD actions (Lyu et al., 2022; Yeom et al., 2024) or introducing extra networks for a quantile (Ma et al., 2021) or a state value function (Chen et al., 2023; Nakamoto et al., 2023), which may lead to increased complexity and further drawbacks despite their intentions to address the over-pessimism — such drawbacks include distribution mismatches between the estimated behavior policy and the dataset (Zhuang et al., 2023; Kun et al., 2024), the need for extensive parameter tuning, and prolonged training times.

Intuitively, leveraging offline trajectories that span multiple timesteps, rather than individual single-timestep transitions, provides more information about the behavior policy and can potentially prevent

---

[*]Corresponding Author

the selection of OOD actions for offline datasets. Although trajectories are readily available in offline datasets, most previous model-free offline RL methods for policy evaluation have utilized these trajectories only in the form of fragmented single-step transitions (Fujimoto et al., 2019; Kumar et al., 2019; Wu et al., 2019; Fujimoto & Gu, 2021; Kostrikov et al., 2021). Hence, the following question arises:

*Can we design a value estimation method for offline RL that utilizes the multi-step learning?*

In *online* RL counterparts of offline RL, there is a line of work that extends a single-step temporal-difference (TD) learning (e.g., Q-learning (WATKINS, 1989)) to multi-step generalizations (Peng & Williams, 1994; Precup, 2000; Munos et al., 2016; Harutyunyan et al., 2016; Rowland et al., 2020; Kozuno et al., 2021), introducing multi-step operators that leverage temporally extended trajectories to update Q-values. These operators improve learning efficiency and provide a more forward-looking perspective, leading to enhanced performance in determining optimal actions compared to the single-step Bellman operator across various benchmarks (Harb & Precup, 2017; Mousavi et al., 2017; Hessel et al., 2018; Barth-Maron et al., 2018; Kapturowski et al., 2018; Daley & Amato, 2019). However, whether such an extension to multi-step TD learning is possible in offline RL is still unclear. Hence, the follow-up questions arise:

*What is a suitable multi-step operator for offline RL?*
*Is it possible to demonstrate that the multi-step operator enhances performance?*

In this paper, we propose an effective conservative multi-step Q-learning algorithm for model-free offline RL, *Conservative Peng's Q(λ)* (CPQL). Our algorithm builds on conservative value estimation by incorporating the Peng's Q(λ) (PQL) operator (Peng & Williams, 1994; Kozuno et al., 2021) instead of the standard Bellman operator. Unlike other multi-step operators (Precup, 2000; Munos et al., 2016; Harutyunyan et al., 2016; Rowland et al., 2020) that truncate trajectories, the PQL operator fully leverages whole trajectories to improve policy evaluation. Since the PQL operator does not use importance sampling, which requires estimating the behavior policy from offline datasets, it avoids the mismatch issues arising from inaccurate behavior policy estimation (Zhuang et al., 2023; Kun et al., 2024). Because the fixed point of the PQL operator in offline RL converges to the Q-function of a mixture policy that interpolates between the behavior policy and target policy, coupling it with a conservative value estimation method ensures that even mild conservatism is sufficient to mitigate Q-value overestimation caused by distribution shift. In contrast to other conservative methods (Kumar et al., 2020; Ma et al., 2021; Lyu et al., 2022; Chen et al., 2023; Nakamoto et al., 2023; Shao et al., 2023; Yeom et al., 2024), CPQL mitigates over-pessimistic value estimation in the Q-function (Theorem 1) without requiring additional estimation procedures or auxiliary networks. Our main contributions are summarized as follows:

- We propose CPQL, the first multi-step Q-learning algorithm for model-free offline RL. CPQL adapts the PQL operator to conservative value estimation and fully leverages offline trajectories without estimating additional models. To the best of our knowledge, our work is the first to demonstrate both theoretically and empirically the effectiveness of conservative multi-step value estimation.

- We provide rigorous theoretical analyses for CPQL. The mixture policy learned by CPQL is guaranteed to achieve performance greater than (or equal to) that of the behavior policy (Theorem 2) and further reduces the sub-optimality gap compared to CQL (Theorem 3). Our theoretical analyses effectively address the key limitations of over-pessimistic value estimation in offline RL, ensuring balanced conservatism and improved policy exploration.

- Extensive numerical experiments on the D4RL benchmark demonstrate that CPQL consistently and significantly outperforms existing offline single-step RL algorithms. In contrast to CQL, whose performance is highly sensitive to the choice of the conservatism parameter (An et al., 2021; Ghasemipour et al., 2022; Tarasov et al., 2024b), CPQL remains robust across a wide range of conservatism parameter values.

- Beyond the contribution of CPQL to mitigating over-pessimistic value estimation in offline RL, CPQL also contributes to the offline-to-online learning framework. Using the Q-function pre-trained by CPQL enables the online PQL agent to avoid the performance drop observed at the start of the online phase and attain robust performance improvements.

## 2 RELATED WORK

**Model-free Offline RL.** To overcome the distributional shift and extrapolation error, model-free offline RL methods focus on learning policies using techniques such as penalizing learned value functions to assign low values to unseen actions (Kumar et al., 2020; Ma et al., 2021; Lyu et al., 2022; Chen et al., 2023; Nakamoto et al., 2023; Shao et al., 2023; Ma et al., 2023; Yeom et al., 2024), constraining the learned policy to remain similar to the behavior policy (Fujimoto et al., 2019; Kumar et al., 2019; Wu et al., 2019; Fujimoto & Gu, 2021; Fakoor et al., 2021; Ghasemipour et al., 2021; Wu et al., 2022; Tarasov et al., 2024a), quantifying the uncertainty (Wu et al., 2021; Zanette et al., 2021) by adding ensemble techniques to obtain a robust value function (Bai et al., 2021; An et al., 2021; Ghasemipour et al., 2022; Yang et al., 2022; Nikulin et al., 2023), and learning without querying OOD actions (Chen et al., 2020; Kostrikov et al., 2022). However, most model-free offline RL algorithms use the single-step TD learning in off-policy methods based on TD3 (Fujimoto et al., 2018), SAC (Haarnoja et al., 2018), and AWR (Peng et al., 2019). Thus, our work addresses over-pessimistic value estimates by leveraging multi-step TD learning based on offline trajectories.

**Multi-step Operators.** Among off-policy multi-step operators (WATKINS, 1989; Peng & Williams, 1994; Cichosz, 1994; Sutton & Barto, 1998; Precup, 2000; Munos et al., 2016; Harutyunyan et al., 2016; Rowland et al., 2020; Kozuno et al., 2021; Daley et al., 2023) in *online* RL, the PQL operator consistently outperforms the Bellman operator and other multi-step operators across several complex *online* tasks (Harb & Precup, 2017; Mousavi et al., 2017; Hessel et al., 2018; Barth-Maron et al., 2018; Kapturowski et al., 2018; Daley & Amato, 2019). The fixed point of the PQL operator in *online* RL has been criticized for its inability to converge to the optimal Q-function without additional technical conditions, such as the updated behavior policy being close to the target policy (Harutyunyan et al., 2016; Kozuno et al., 2021). However, under the fixed behavior policy (*offline* settings), we exploit the property (Kozuno et al., 2021) that its fixed point is closer to the Q-function of the behavior policy. By integrating conservative value estimation into this property, CPQL tackles two central offline RL challenges: distributional shift and overly pessimistic value estimates.

**Offline-to-Online RL.** To prevent the forgetting of offline pre-training benefits and to enable efficient online exploration, offline-to-online RL methods have explored diverse techniques such as leveraging an offline dataset to sample-efficient online fine-tuning (Nair et al., 2020; Lee et al., 2022; Song et al., 2022), avoiding the need to retain offline data (Uchendu et al., 2023; Zhou et al., 2024), maintaining a balanced replay buffer (Ball et al., 2023; Ji et al., 2024; Luo et al., 2024), calibrating the value function (Nakamoto et al., 2023), adopting Bayesian approaches (Hu et al., 2024), bridging the value gap between offline and online RL (Yu & Zhang, 2023; Wagenmaker & Pacchiano, 2023), and proposing policy expansion schemes (Zhang et al., 2023). However, since CPQL mitigates over-pessimistic value estimation in the offline phase, it eliminates the need for additional mechanisms such as critic-actor calibration or alignment when transitioning to vanilla PQL in the online learning. This allows the online agent to directly leverage the pre-trained Q-function without further adjustment, ensuring a smoother transition to online fine-tuning. As a result, CPQL avoids the performance drop typically observed at the start of fine-tuning and attains robust performance improvements.

## 3 PRELIMINARIES

### 3.1 MARKOV DECISION PROCESS

We consider a Markov Decision Process (MDP) defined by a tuple $\mathcal{M} := (\mathcal{S}, \mathcal{A}, \mathcal{P}, \mathcal{R}, d_0, \gamma)$, where $\mathcal{S}$ is the state space, $\mathcal{A}$ is the action space, $\mathcal{P} : \mathcal{S} \times \mathcal{A} \to \Delta_{\mathcal{S}}$ represents the state transition probability kernel, $\mathcal{R}$ is the reward distribution, $d_0 \in \Delta_{\mathcal{S}}$ is the initial state distribution, and $\gamma \in [0, 1)$ is the discount factor. We let the reward function $r \in \mathbb{R}^{\mathcal{S} \times \mathcal{A}}$ be defined as $r(s, a) := \int r' \mathcal{R}(dr'|s, a)$, and assume that $|r(s, a)| \leq R_{\max}, \forall (s, a) \in \mathcal{S} \times \mathcal{A}$. Let a policy $\pi : \mathcal{S} \to \Delta_{\mathcal{A}}$ be a mapping from states to actions (deterministic) or a probability distribution over actions (stochastic). Given any policy $\pi$, an agent starts from an initial state $s_0$ and interacts with $\mathcal{M}$, repeatedly taking actions, receiving rewards, and observing subsequent states. This process generates a trajectory $\tau = \{(s_t, a_t, r(s_t, a_t))\}_{t \geq 0}$, where $a_t \sim \pi(\cdot|s_t)$ and $s_{t+1} \sim \mathcal{P}(\cdot|s_t, a_t)$. The state-value function and action-value function (Q-function) for the policy $\pi$ are defined as $V^\pi(s) := \mathbb{E}_\pi \left[ \sum_{t=0}^\infty \gamma^t r(s_t, a_t) \mid s_0 = s \right]$ and $Q^\pi(s, a) := \mathbb{E}_\pi \left[ \sum_{t=0}^\infty \gamma^t r(s_t, a_t) \mid s_0 = s, a_0 = a \right]$, respectively. We define the discounted state visitation distribution of a policy $\pi$ under the environment

$\mathcal{M}$ as $d_{\mathcal{M}}^{\pi}(s) := (1-\gamma)\sum_{t=0}^{\infty}\gamma^{t}\mathrm{Pr}^{\pi}(s_t = s \mid s_0 \sim d_0, \mathcal{P})$, where $\mathrm{Pr}^{\pi}(s_t = s \mid s_0 \sim d_0, \mathcal{P})$ denotes the probability of reaching state $s$ at time-step $t$ under $\pi$ and $\mathcal{P}$, starting from initial state $s_0$ distributed according to the initial state distribution $d_0$. Similarly, we define the discounted state-action visitation distribution as $d_{\mathcal{M}}^{\pi}(s,a) := d_{\mathcal{M}}^{\pi}(s)\,\pi(a|s)$.

## 3.2 OFF-POLICY OPERATORS

Off-policy RL consists of two main tasks: evaluation and improvement. The evaluation process is to learn the Q-function of a fixed policy. In the improvement setting, the goal is to obtain an optimal policy $\pi^*$ that maximizes the expected discounted return under $d_0$, represented as $\max_{\pi} J_{\mathcal{M}}(\pi) := \mathbb{E}_{s \sim d_0}[V^{\pi}(s)] = \frac{1}{1-\gamma}\mathbb{E}_{s,a \sim d_{\mathcal{M}}^{\pi}}[r(s,a)]$. Operators are a fundamental concept in RL because all value-based RL algorithms update the Q-function using a recursion $Q_{k+1} := \mathcal{O}_k Q_k$, where $\mathcal{O}_k : \mathbb{R}^{\mathcal{S} \times \mathcal{A}} \to \mathbb{R}^{\mathcal{S} \times \mathcal{A}}$ is an operator that specifies the update rule of the algorithm. We define $\mathcal{P}^{\pi}$ as the transition matrix coupled with the policy, given by $\mathcal{P}^{\pi}Q(s,a) := \mathbb{E}_{s' \sim \mathcal{P}(\cdot|s,a),\, a' \sim \pi(\cdot|s')}[Q(s',a')]$.

**Bellman Operator.** The Bellman operator $\mathcal{T}^{\pi} : \mathbb{R}^{\mathcal{S} \times \mathcal{A}} \to \mathbb{R}^{\mathcal{S} \times \mathcal{A}}$ is defined as $\mathcal{T}^{\pi}Q := r + \gamma \mathcal{P}^{\pi}Q$. We denote the set of all greedy policies with respect to $Q$ as $\mathbf{G}(Q)$. The Bellman optimality operator $\mathcal{T}^*$ is defined by $\mathcal{T}^*Q := \mathcal{T}^{\pi_Q}Q$, where $\pi_Q \in \mathbf{G}(Q)$.

**Peng's Q($\lambda$) (PQL).** For $\lambda \in [0,1)$, PQL updates the Q-function using the recursion $Q_{k+1} := \mathcal{T}_{\lambda}^{\pi_{\beta,k},\pi_k}Q_k$ (Peng & Williams, 1994; Kozuno et al., 2021), where $\pi_k \in \mathbf{G}(Q_k)$. The PQL operator $\mathcal{T}_{\lambda}^{\pi_{\beta},\pi} : \mathbb{R}^{\mathcal{S} \times \mathcal{A}} \to \mathbb{R}^{\mathcal{S} \times \mathcal{A}}$ is defined as $\mathcal{T}_{\lambda}^{\pi_{\beta},\pi}Q := (1-\lambda)\sum_{n=1}^{\infty}\lambda^{n-1}\mathcal{T}_n^{\pi_{\beta},\pi}Q$, where $\mathcal{T}_n^{\pi_{\beta},\pi}Q := (\mathcal{T}^{\pi_{\beta}})^{n-1}\mathcal{T}^{\pi}Q$ is the uncorrected $n$-step return operator (WATKINS, 1989; Cichosz, 1994; Sutton & Barto, 1998; Hessel et al., 2018).

## 3.3 OFFLINE RL

In offline RL, the learned policy is constrained to a static dataset without additional interactions with the environment during the control process. The offline dataset $\mathcal{D}$ consists of either trajectories $\{\tau_i\}_{i=1}^{n}$ gathered by unknown behavior policies $\pi_{\beta}$. On all states $s \in \mathcal{D}$, we denote the empirical behavior policy as $\hat{\pi}_{\beta}(a|s) := \frac{\sum_{s,a \in \mathcal{D}}\mathbf{1}[s=s,a=a]}{\sum_{s \in \mathcal{D}}\mathbf{1}[s=s]}$. We define the state space induced by $\mathcal{D}$ as $S(\mathcal{D})$, consisting of all states in $\mathcal{D}$. Since $\mathcal{D}$ typically covers a subset of the tuple space, offline RL algorithms based on the Bellman operator suffer from action distribution shift. Because the learning policy is updated to maximize Q-values, cumulative extrapolation errors in unseen actions can drive it toward OOD actions with erroneously high Q-values (Levine et al., 2020).

To address the overestimated Q-value problem, CQL (Kumar et al., 2020) penalizes the learned Q-function for OOD actions induced by the learning policy. The objective function of CQL with a non-negative conservatism parameter $\alpha$ is defined as follows:

$$\frac{1}{2}\mathbb{E}_{s,a,s' \sim \mathcal{D}}\left[(Q(s,a) - \mathcal{T}^{\pi}Q(s,a))^2\right] + \alpha\left(\mathbb{E}_{s \sim \mathcal{D}, a \sim \pi(\cdot|s)}[Q(s,a)] - \mathbb{E}_{s,a \sim \mathcal{D}}[Q(s,a)]\right). \quad (1)$$

## 4 CONSERVATIVE PQL

In this section, we develop the CPQL algorithm, where the learned Q-function mitigates overestimation bias in value estimation. We provide several novel theoretical results that include guarantees for the sub-optimality gap between the optimal policy and the mixture policy learned via CPQL. It is important to note that PQL has not been studied under *offline* RL settings. Hence, we first present how previous findings on online PQL can be adapted to offline PQL, addressing fundamental challenges of Q-learning in offline RL.

### 4.1 TOWARDS OFFLINE PQL

Prior works (Peng & Williams, 1994; Sutton & Barto, 1998; Kozuno et al., 2021) have investigated the PQL operator only in *online* RL. In this work, we focus on constructing the PQL operator in *offline* RL for the first time. Adapting PQL to offline RL not only facilitates faster convergence to the fixed point but also mitigates the effects of extrapolation errors and over-pessimistic value estimation, which are key issues in offline RL. We begin by recalling the fixed-point characterization of the PQL

operator and reinterpreting it from an *offline* RL perspective. We consider the exact case where no update errors exist in the value functions.

**Proposition 1 (Harutyunyan et al. (2016))** *The fixed point of the PQL operator, $Q^{\pi_\beta,\pi}$, satisfies:*

$$Q^{\pi_\beta,\pi} = (\lambda \mathcal{T}^{\pi_\beta} + (1-\lambda)\mathcal{T}^\pi) Q^{\pi_\beta,\pi}.$$

Proposition 1 states that a fixed point of the PQL operator coincides with the fixed point of $\lambda \mathcal{T}^{\pi_\beta} + (1-\lambda)\mathcal{T}^\pi$ for the target policy $\pi$. Since $\lambda \mathcal{T}^{\pi_\beta} + (1-\lambda)\mathcal{T}^\pi$ is a contraction with modulus $\gamma$ under $L^\infty$-norm, the existence and uniqueness of this fixed point are guaranteed. However, this fixed point does not ensure the convergence of the optimal Q-function in online RL unless $\pi_\beta$ is sufficiently close to $\pi$ (Harutyunyan et al., 2016; Kozuno et al., 2021). In contrast, when we use the *fixed* empirical behavior policy $\hat{\pi}_\beta$ from $\mathcal{D}$, the Q-function updated by the PQL operator converges to $Q^{\lambda \hat{\pi}_\beta + (1-\lambda)\pi}$.

**Proposition 2 (Kozuno et al. (2021))** *Let $\pi$ be a policy such that $Q^{\lambda \hat{\pi}_\beta + (1-\lambda)\pi} \geq Q^{\lambda \hat{\pi}_\beta + (1-\lambda)\bar{\pi}}$ holds pointwise for any policy $\bar{\pi}$. Then, $Q_k$ for the $k$-th iteration, updated by the PQL operator, uniformly converges to $Q^{\lambda \hat{\pi}_\beta + (1-\lambda)\pi}$ with a contraction rate of $\beta^k$, where $\beta := \frac{\gamma(1-\lambda)}{1-\gamma\lambda}$.*

Proposition 2 states a trade-off between bias and contraction rate, that is, PQL with the fixed behavior policy converges to a biased fixed point that differs from $Q^*$, with a contraction rate $\beta$.

**Interpretation to offline RL.** Prior work (Kozuno et al., 2021) focused on online RL, particularly on how updating the behavior policy is necessary for this fixed point to converge to $Q^*$. However, the fixed point $Q^\pi$ with $\lambda = 0$, corresponding to the value derived from the Bellman operator, can still deviate from $Q^*$ due to distribution shift in offline RL (Fujimoto et al., 2019; Kumar et al., 2019; Levine et al., 2020). Thus, one of our main points is that we should focus on *how an appropriately chosen $\lambda$ mitigates Q-value overestimation for the learned policy* by shifting the fixed point closer to $Q^{\hat{\pi}_\beta}$, rather than focusing on increasing the bias introduced by a large $\lambda$. Because the fixed point lies closer to the behavioral value, it naturally induces implicit behavior regularization. A carefully chosen $\lambda$ can effectively address the over-pessimism problem in conservative value estimation methods and yield a more robust learned Q-function, as it mitigates the influence of the learned policy.

## 4.2 THEORETICAL ANALYSIS

We aim to mitigate the over-pessimistic estimation of Q-values for the learned policy induced by conservatism. We integrate the PQL operator into the CQL loss, as it provides a simple and effective way to alleviate the over-pessimism of Q-values. We replace the standard Bellman operator in Equation 1 with the PQL operator. This leads to the following iterative Q-value update in CPQL:

$$\widehat{Q}_{k+1} \in \underset{Q}{\arg\min} \left\{ \frac{1}{2}\mathbb{E}_{s,a,s'\sim\mathcal{D}} \left[ \left( Q(s,a) - \mathcal{T}_\lambda^{\hat{\pi}_\beta,\pi_k}\widehat{Q}_k(s,a) \right)^2 \right] \right.$$
$$\left. + \alpha \left( \mathbb{E}_{s\sim\mathcal{D},a\sim\pi_k(\cdot|s)} [Q(s,a)] - \mathbb{E}_{s,a\sim\mathcal{D}} [Q(s,a)] \right) \right\}. \quad (2)$$

The following theorem shows that the expectation of the learned Q-function obtained by iterating Equation 2 lower bounds the expectation of the true Q-function. This result is an adaptation of Theorem 3.2 in Kumar et al. (2020). For readability, we state the main results without sampling-error terms. The corresponding finite-sample high-probability bounds are provided in Appendix B.1.

**Theorem 1 (Lower Bound on the State Value Function of CPQL)** *Let $\widehat{Q}^{\lambda\hat{\pi}_\beta + (1-\lambda)\pi}$ denote the Q-function derived from CPQL as defined in Equation 2. Then, the state value of $\lambda\hat{\pi}_\beta + (1-\lambda)\pi$, $\widehat{V}^{\lambda\hat{\pi}_\beta + (1-\lambda)\pi}(s) = \mathbb{E}_{a\sim(\lambda\hat{\pi}_\beta + (1-\lambda)\pi)(\cdot|s)} \left[ \widehat{Q}^{\lambda\hat{\pi}_\beta + (1-\lambda)\pi}(s,a) \right]$, lower bounds the true state value of the policy obtained via exact policy evaluation, $V^{\lambda\hat{\pi}_\beta + (1-\lambda)\pi}(s)$, for sufficiently large $\alpha$. Formally, for all $s \in \mathcal{S}(\mathcal{D})$,*

$$\widehat{V}^{\lambda\hat{\pi}_\beta + (1-\lambda)\pi}(s) \leq V^{\lambda\hat{\pi}_\beta + (1-\lambda)\pi}(s).$$

The next two theorems show that the mixture policy learned by CPQL is guaranteed to achieve the performance greater than (or equal to) that of $\hat{\pi}_\beta$ (Theorem 2) and reduces the sub-optimality gap (Theorem 3), which previous conservative value estimation methods had not achieved. The proofs with sampling error are deferred to Appendix B.2 and B.3.

**Theorem 2 (Comparison to the Behavior Policy)** *Let* $\hat{\pi} := \mathrm{argmax}_\pi \, \mathbb{E}_{s \sim d_0} \left[ \widehat{V}^{\lambda \hat{\pi}_\beta + (1-\lambda)\pi}(s) \right]$.
$\lambda \hat{\pi}_\beta + (1-\lambda)\hat{\pi}$ *achieves a policy improvement over* $\hat{\pi}_\beta$ *in the actual MDP* $\mathcal{M}$ *as follows:*

$$J_{\mathcal{M}} \left( \lambda \hat{\pi}_\beta + (1-\lambda)\hat{\pi} \right) \geq J_{\mathcal{M}} \left( \hat{\pi}_\beta \right) + \frac{\alpha(1-\lambda)}{1-\gamma} \mathbb{E}_{s \sim d_{\mathcal{M}}^{\lambda \hat{\pi}_\beta + (1-\lambda)\hat{\pi}}(s)} \left[ \mathbb{E}_{a \sim \hat{\pi}(\cdot|s)} \left[ \frac{\hat{\pi}(a|s)}{\hat{\pi}_\beta(a|s)} - 1 \right] \right].$$

**Theorem 3 (Sub-Optimality Gap)** *The gap of the expected discounted return between the optimal policy* $\pi^*$ *and the mixture policy* $\lambda \hat{\pi}_\beta + (1-\lambda)\hat{\pi}$ *under the actual MDP* $\mathcal{M}$ *satisfies*

$$J_{\mathcal{M}}(\pi^*) - J_{\mathcal{M}}(\lambda \hat{\pi}_\beta + (1-\lambda)\hat{\pi})$$

$$\leq \frac{2\lambda R_{\max}}{(1-\gamma)^2} \mathbb{E}_{s \sim d_{\mathcal{M}}^{\lambda \hat{\pi}_\beta + (1-\lambda)\pi^*}} \left[ d_{\mathrm{TV}}(\pi^*, \hat{\pi}_\beta)(s) \right]$$

$$+ \frac{2\alpha(1-\lambda)}{1-\gamma} \mathbb{E}_{s \sim d_{\mathcal{M}}^{\lambda \hat{\pi}_\beta + (1-\lambda)\pi^*}(s)} \left[ d_{\mathrm{TV}}(\pi^*, \hat{\pi})(s) \left( \xi(\hat{\pi})(s) + \frac{\gamma}{1-\gamma} \mathbb{E}_{a \sim \hat{\pi}(\cdot|s)} \left[ \frac{\hat{\pi}(a|s)}{\hat{\pi}_\beta(a|s)} - 1 \right] \right) \right],$$

*where* $\xi(\hat{\pi})(s) := \sum_{a \in \mathcal{A}} \frac{\pi^*(a|s) + \hat{\pi}(a|s)}{\hat{\pi}_\beta(a|s)}$ *and* $d_{\mathrm{TV}}(\pi_1, \pi_2)$ *is the total variation distance of* $\pi_1$ *and* $\pi_2$.

**Discussion of Theorems 2 and 3.** In Theorem 2, $\lambda \hat{\pi}_\beta + (1-\lambda)\hat{\pi}$ achieves at least the performance of $\hat{\pi}_\beta$ under the actual MDP $\mathcal{M}$. When accounting for sampling error, $\alpha$ is chosen such that the conservative term exceeds the sum of the sampling error terms. However, an excessively large $\alpha$ does not guarantee that the sub-optimality gap decreases. In Theorem 3, increasing $\alpha$ significantly can lead to larger influence of $\xi(\hat{\pi})$ and $\mathbb{E}_{a \sim \hat{\pi}(\cdot|s)}[\hat{\pi}(a|s)/\hat{\pi}_\beta(a|s)]$ on the RHS. To reduce these gaps, it is crucial to control the two unbounded terms, since their reduction has a greater effect than reducing the total variation distance. Thus, $\hat{\pi}$ approaches $\hat{\pi}_\beta$ when $\alpha$ takes on a large value by Theorem 2. However, the bound suggests that CPQL can reduce the sub-optimality bound relative to the $\lambda = 0$ case under suitable behavior-policy quality and coverage conditions. If $\hat{\pi}$ deviates from $\hat{\pi}_\beta$, then $\xi(\hat{\pi})$ and $\mathbb{E}_{a \sim \hat{\pi}(\cdot|s)}[\hat{\pi}(a|s)/\hat{\pi}_\beta(a|s)]$ can grow large to infinity. While CQL lacks a mechanism to directly mitigate this divergence, CPQL addresses it through $\lambda$, which balances between the first and second terms on the RHS. For example, since $\pi^*$ and $\hat{\pi}_\beta$ are fixed policies, if $\hat{\pi}_\beta$ is similar to $\pi^*$, choosing a large value of $\lambda$ further reduces the sub-optimality gap. Conversely, if $\hat{\pi}_\beta$ differs from $\pi^*$, adjusting $\lambda$ appropriately is more effective than CQL at reducing the sub-optimality gap.

### 4.3 PROPOSED ALGORITHM

---

**Algorithm 1** Conservative Peng's Q($\lambda$) (CPQL)

---

**Require:** Critic networks $Q_{\theta_1}, Q_{\theta_2}$, Actor network $\pi_\phi$, Dataset $\mathcal{D}$, Conservatism factor $\alpha$, and $\lambda$
1: Initialize target networks $\theta_1^- \leftarrow \theta_1$, $\theta_2^- \leftarrow \theta_2$
2: **for** gradient step $t = 1, 2, \cdots$ **do**
3:  Sample batch partial trajectories each of length $n$, $\{(s_0, a_0, r_0, s_1, a_1, r_1, \cdots, s_n)\}$, from $\mathcal{D}$
4:  **for** $i = n-1$ to $0$ **do**
5:   Compute $\widehat{Q}_{\theta_j}^i = r_i + \gamma Q_{\theta_j^-}(s_{i+1}, \pi_\phi(s_{i+1})) + \gamma\lambda \left( \widehat{Q}_{\theta_j}^{i+1} - Q_{\theta_j^-}(s_{i+1}, \pi_\phi(s_{i+1})) \right), j = 1,2$
6:  **end for**
7:  Construct target value $y = \min_{j=1,2} \widehat{Q}_{\theta_j}^0 - \gamma^n \alpha_{\mathrm{td}} \log \pi_\phi(\cdot|s_n)$
8:  Update critic $\theta_j$ for $j = 1, 2$ with gradient descent via minimizing

$$\alpha \mathbb{E}_{s \sim \mathcal{D}} \left[ \log \sum_a \exp\left( Q_{\theta_j}(s,a) \right) - \mathbb{E}_{a \sim \hat{\pi}_\beta(\cdot|s)} \left[ Q_{\theta_j}(s,a) \right] \right] + \frac{1}{2} \mathbb{E}_{s,a,s' \sim \mathcal{D}} \left[ \left( Q_{\theta_j}(s,a) - y \right)^2 \right]$$

9:  Update actor $\phi$ for learned policy $\pi_\phi$ with gradient ascent via maximizing

$$\mathbb{E}_{s \sim \mathcal{D}, a \sim \pi_\phi(\cdot|s)} \left[ \min_{j=1,2} Q_{\theta_j}(s,a) - \alpha_{\mathrm{pol}} \log \pi_\phi(\cdot|s) \right]$$

10:  Update target networks: $\theta_j^- \rightarrow \tau \theta_j + (1-\tau) \theta_j^-, j = 1, 2$
11: **end for**

---

Algorithm 1 presents a general version of our proposed method. In Line 5, given a partial trajectory of length $n$, we recursively compute the target Q-function using the trace parameter $\lambda$. While updates

Table 1: Results for MuJoCo locomotion, Adroit manipulation, and AntMaze navigation tasks in offline D4RL. * indicates reproduced results: (algorithm*) for all datasets, (score*) for a specific dataset. Bold numbers are the scores within 2% of the highest in each environment.

| Task | BC* | TD3+BC | CQL | IQL | MCQ | MISA | CSVE | EPQ* | CPQL (ours) |
|---|---|---|---|---|---|---|---|---|---|
| halfcheetah-random | 2.2 | 11.0 | 17.5* | 13.1* | 28.5 | 2.5* | 26.8 | 31.9 | **38.8 ± 1.0** |
| hopper-random | 3.7 | 8.5 | 7.9* | 7.9* | **31.8** | 9.9* | 26.1 | 30.3 | **31.5 ± 0.5** |
| walker2d-random | 1.3 | 1.6 | 5.1* | 5.4* | 17.0 | 9.0* | 6.2 | 11.2 | **21.2 ± 0.7** |
| halfcheetah-medium | 43.2 | 48.3 | 47.0* | 47.4 | 64.3 | 47.4 | 48.4 | **67.1** | **66.6 ± 0.9** |
| hopper-medium | 54.1 | 59.3 | 53.0* | 66.2 | 78.4 | 67.1 | 96.7 | **100.4** | **99.7 ± 2.0** |
| walker2d-medium | 70.9 | 83.7 | 73.3* | 78.3 | **91.0** | 84.1 | 83.2 | 86.4 | **90.0 ± 1.5** |
| halfcheetah-medium-replay | 37.6 | 44.6 | 45.5* | 44.2 | 56.8 | 45.6 | 54.5 | 51.4 | **60.3 ± 0.8** |
| hopper-medium-replay | 16.6 | 60.9 | 88.7* | 94.7 | **101.6** | 98.6 | 91.7 | 97.3 | **103.0 ± 0.6** |
| walker2d-medium-replay | 20.3 | 81.8 | 81.8* | 73.8 | 91.3 | 86.2 | 78.0 | 86.0 | **97.4 ± 4.0** |
| halfcheetah-medium-expert | 44.0 | 90.7 | 75.6* | 86.7 | 87.5 | **94.7** | 93.1 | 86.6 | **95.3 ± 0.6** |
| hopper-medium-expert | 53.9 | 98.0 | 105.6* | 91.5 | **111.2** | 109.8 | 94.1 | **110.4** | **111.3 ± 1.2** |
| walker2d-medium-expert | 90.1 | 110.1 | 107.9* | 109.6 | **114.2** | 109.4 | 109.0 | 110.9 | **112.9 ± 2.0** |
| halfcheetah-expert | 91.8 | 96.7 | 96.3* | 95.0* | 96.2 | 95.9* | 93.8 | **102.9** | 98.0 ± 1.6 |
| hopper-expert | 107.7 | 107.8 | 96.5* | 109.4* | **111.4** | **111.9** | 111.3 | 111.1 | **112.0 ± 0.6** |
| walker2d-expert | 106.7 | 110.2 | 108.5* | 109.9* | 107.2 | 109.3* | 108.5 | 109.8 | **114.1 ± 0.5** |
| MuJoCo Total | 744.1 | 1013.2 | 1010.2 | 1033.1 | 1188.4 | 1081.4 | 1121.4 | 1193.7 | **1252.1** |
| pen-human | 34.4 | 64.8* | 37.5 | 71.5 | 68.5 | 88.1 | **106.2** | 65.7 | 72.1 ± 4.6 |
| door-human | 0.5 | 0.0* | 9.9 | 4.3 | 2.3 | 5.2 | 2.8 | 5.1 | **14.3 ± 2.2** |
| hammer-human | 1.5 | 1.8* | 4.4 | 1.4 | 1.3 | **8.1** | 3.5 | 0.3 | 1.4 ± 0.9 |
| relocate-human | 0.0 | 0.1* | **0.2** | 0.1 | 0.1 | 0.1 | 0.1 | 0.1 | 0.1 ± 0.0 |
| pen-cloned | 56.9 | 49.0* | 39.2 | 37.3 | 49.4 | 58.6 | 54.5 | 55.8 | **70.9 ± 6.9** |
| door-cloned | -0.1 | 0.0* | 0.4 | 1.6 | 1.3 | 0.5 | 1.2 | 0.5 | **6.4 ± 5.0** |
| hammer-cloned | 0.8 | 0.2* | 2.1 | 2.1 | 1.4 | 2.2 | 0.5 | 1.2 | 1.6 ± 1.1 |
| relocate-cloned | -0.1 | -0.2* | -0.1 | -0.2 | **0.0** | -0.1 | -0.3 | -0.1 | -0.1 ± 0.0 |
| Adroit Total | 93.9 | 115.7 | 93.6 | 118.1 | 124.3 | 162.7 | **168.5** | 128.7 | **166.7** |
| antmaze-umaze | 65.0 | 78.6 | 74.0 | 87.5 | **98.3*** | 92.3 | - | 96.2 | **96.7 ± 1.9** |
| antmaze-umaze-diverse | 55.0 | 71.4 | 84.0 | 62.2 | 80.0* | **89.1** | - | 72.3 | 68.6 ± 0.5 |
| antmaze-medium-play | 0.0 | 10.6 | 61.2 | **71.2** | 52.5* | 63.0 | - | 59.0 | **72.4 ± 1.2** |
| antmaze-medium-diverse | 0.0 | 3.0 | 53.7 | **70.0** | 37.5* | 62.8 | - | 57.5 | **71.7 ± 0.8** |
| antmaze-large-play | 0.0 | 0.2 | 15.8 | 39.6 | 2.5* | 17.5 | - | 23.8 | **41.6 ± 5.2** |
| antmaze-large-diverse | 0.0 | 0.0 | 14.9 | **47.5** | 7.5* | 23.4 | - | 17.4 | **46.6 ± 4.9** |
| Antmaze Total | 120.0 | 163.8 | 303.6 | 378.0 | 278.3* | 348.1 | - | 326.2 | **397.6** |

are based on SAC (Haarnoja et al., 2018), we set $\alpha_{\mathrm{td}} = 0$ at all steps except the last (Line 7), ensuring stability during Q-function updates. This is because the entropy bonus term is added to the target Q-function at each step, amplifying its numerical scale and complicating value estimations (Kozuno et al., 2021). In Line 8, we adopt the log-sum-exp method from CQL (Kumar et al., 2020) to incorporate conservative value estimation. Compared to CQL, CPQL reduces the influence of the learned policy on Q-value estimates, enabling stable learning even with a small conservatism factor $\alpha$ (see Question (ii) in Section 5).

## 5 EXPERIMENTS

In this section, we describe our detailed experimental procedures and report the corresponding results to address the following pertinent questions:

(i) How does the performance of CPQL compare to prior single-step offline baselines, some of which incorporate conservative value estimation methods, across various tasks and datasets?

(ii) What advantage does CPQL provide over CQL in terms of sensitivity to the conservatism parameter $\alpha$, and does it mitigate over-conservatism while achieving strong performance?

(iii) How does CPQL compare with other multi-step operators (e.g., Uncorrected N-step, Retrace, and Tree-backup) when combined with conservative value estimation? (Note that there are no existing offline RL methods with a multi-step operator. Here, we are asking a question on ablation.)

(iv) Can the online PQL agent, using the Q-function pre-trained by CPQL in offline settings, mitigate performance drop at the start of the online phase and enable faster adaptation and improvement in online learning compared to offline-to-online baselines?

For a fair comparison, we evaluate all algorithms using results after 1M gradient steps in offline D4RL (Fu et al., 2020). In the offline-to-online setting, we first pre-train algorithms for 0.25M offline steps and then fine-tune them for 0.3M online steps. Our score is computed from the policy during the last 10 iterations, averaged over 5 seeds, with $\pm$ denoting the standard deviation across seeds. For CPQL evaluation, we set $n = 5$ to cap the length of the partial trajectories. (See Appendix C for details).

**Tasks.** MuJoCo (Todorov et al., 2012) consists of datasets from three environments (*HalfCheetah*, *Hopper*, and *Walker2d*), each with five dataset types (*Random*, *Medium*, *Medium-Replay*, *Medium-Expert*, and *Expert*). Adroit (Rajeswaran et al., 2017) involves two dataset types (*human* and *cloned*) and four Shadow Hand robot tasks (*hammer*, *door*, *pen*, and *relocate*). AntMaze provides three maze layouts (*umaze*, *medium*, and *large*) and three dataset types (*umaze*, *play*, and *diverse*).

**Baselines.** In the offline setting, we compare CPQL to prior model-free single-step offline RL algorithms: (i) TD3+BC (Fujimoto & Gu, 2021) that incorporates an explicit policy constraint through the behavior cloning (BC), (ii) CQL (Kumar et al., 2020) that penalizes the Q-function for OOD actions, (iii) IQL (Kostrikov et al., 2022) that learns the Q-function without querying OOD actions, (iv) MCQ (Lyu et al., 2022) that uses the mildly conservative Bellman operator, (v) MISA (Ma et al., 2023) that constrains the policy based on mutual information, (vi) CSVE (Chen et al., 2023) that learns a conservative state-value function, and (vii) EPQ (Yeom et al., 2024) that learns the Q-function by selectively penalizing states with insufficient action coverage. In offline-to-online RL, we evaluate the performance of CPQL (offline pretraining) followed by PQL (online fine-tuning), and compare it against several algorithms: (i) AWAC (Nair et al., 2020) that utilizes the advantage weighted actor-critic with weighted maximum likelihood, (ii) Cal-QL (Nakamoto et al., 2023) that calibrates the value-function, (iii) IQL (Kostrikov et al., 2022), (iv) SPOT (Wu et al., 2022) that uses density-based regularization, and (v) CQL (Kumar et al., 2020) (offline) to SAC (online).

## 5.1 RESULTS ON OFFLINE AND OFFLINE-TO-ONLINE D4RL

**Question (i):** Our experimental results, summarized in Table 1, are based on evaluations carried out across diverse tasks. CPQL achieves high performance in the vast majority of the tasks with **22** out of 29 tasks. In MuJoCo locomotion tasks, CPQL consistently achieves remarkable performance improvements across all tasks, regardless of data distribution—whether diverse (*Random*, *Medium-Replay*) or narrow (*Medium*, *Medium-Expert*). In Adroit manipulation tasks, CPQL surpasses all other algorithms for *door* on *human* and *cloned* datasets. Excluding only CSVE in the *pen-human* dataset, we achieve high performance in two *pen* tasks. In Antmaze navigation tasks, CPQL demonstrates outstanding performance despite sparse rewards and diverse datasets (undirected and multi-task). These results on diverse tasks demonstrate that CPQL effectively mitigates the problem of over-pessimistic value estimation by leveraging actual trajectories and the PQL operator.

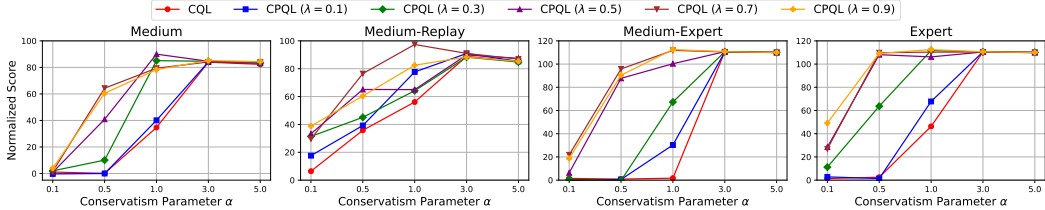

Figure 1: Normalized scores of different conservatism parameters $\alpha$ in *Walker2d* tasks.

**Question (ii):** CPQL maintains high performance even at small $\alpha$, unlike CQL. The smaller $\alpha$ helps CPQL address the issue of overly penalizing the Q-values of certain states in CQL, particularly less observed or unobserved states in $\mathcal{D}$. Prior works (An et al., 2021; Ghasemipour et al., 2022; Tarasov et al., 2024b) have pointed out that CQL is extremely sensitive to the choice of $\alpha$, as even small changes can lead to significant performance differences. In Figure 1, the red line representing CQL clearly illustrates this sensitivity issue. In contrast, CPQL outperforms CQL and exhibits less

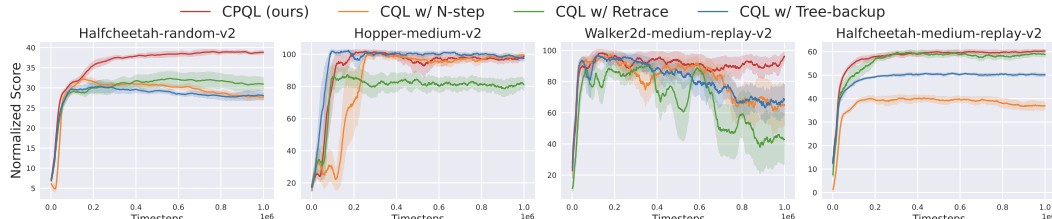

Figure 2: Comparisons of CPQL (ours) with CQL using alternative multi-step operators on MuJoCo tasks.

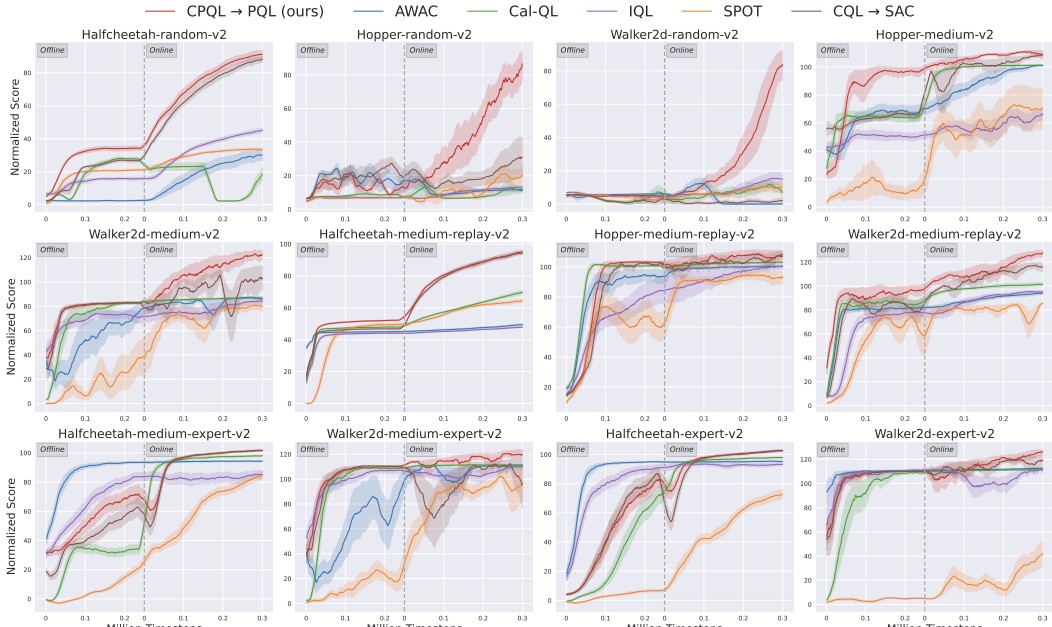

Figure 3: Comparing CPQL→PQL (ours) with several baselines for offline-to-online RL.

sensitivity to $\alpha$ across diverse datasets. By mitigating over-conservatism, CPQL enables the learned policy to better explore promising actions. As shown in Theorem 3, selecting an appropriate $\lambda$ reduces the sub-optimality gap and yields remarkable scores across diverse datasets.

**Question (iii):** In Figure 2, the Uncorrected $n$-step return, Retrace, and Tree-backup operators indeed learn faster during the first $0.2M$ steps, but their performance drops after reaching an early peak. Retrace (Munos et al., 2016) suffers from performance degradation because it relies on accurate behavior policy estimation, which is difficult to estimate (Zhuang et al., 2023; Kun et al., 2024). Tree-backup (Precup, 2000) is developed for discrete action spaces, and in continuous spaces, it leads to unstable updates due to the numerical scale of $\ln \pi$. The Uncorrected $n$-step return overly restricts exploration of OOD actions, which can lead to unstable performance in the later stages of training. However, CPQL achieves both stable and competitive performance without additional requirements.

**Question (iv):** In Figure 3, we show that initializing PQL with the Q-function pre-trained by CPQL helps the online agent avoid or quickly recover from the performance drop at the start of online fine-tuning and achieve robust improvement. First, CPQL outperforms other offline-to-online baselines with only $0.25M$ gradient steps, so the online agent is initialized with the well-trained Q-function, reducing exploration trials. Second, in Figure 4, the Q-values learned by PQL do not degrade at the start of the online phase. Since CPQL reduces the influence of the learned policy on Q-value estimation (Proposition 2 and Theorem 1), the average Q-value gradually increases after pretraining across different values of $\alpha$.

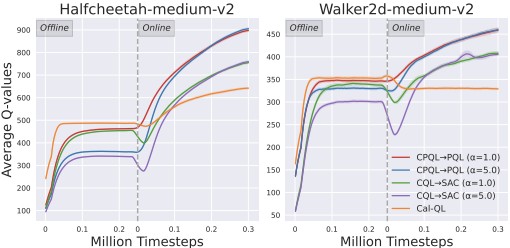

Figure 4: Average Q-values by conservatism parameter $\alpha$ for CPQL→PQL (ours), CQL→SAC, and Cal-QL.

In contrast, when transitioning from CQL to SAC, a larger $\alpha$ shows a more severe performance drop. While Cal-QL avoids the performance drop, its performance improvement is significantly slower.

## 6 CONCLUSION

CPQL proposes the first approach to a model-free offline multi-step RL algorithm by incorporating the PQL operator for conservative value estimation, mitigating over-pessimism in the Q-function, and reducing the sub-optimality gap. A key insight of CPQL is that the fixed point of the PQL operator lies closer to the value function of the behavior policy, thereby inducing implicit behavior regularization. CPQL outperforms existing offline RL algorithms, and its pre-trained Q-function enables PQL to avoid the performance drop at the start of fine-tuning and achieve robust performance improvements in the online phase. There are two limitations of CPQL. First, CPQL incurs additional computational cost due to multi-step backups, but in practice, the overhead and the increase in running time are small. Second, on low-quality datasets, performance may degrade, and single-step updates can be preferable to multi-step operators. However, CPQL can reproduce the single-step TD learning.

## ACKNOWLEDGEMENTS

This work was supported by the National Research Foundation of Korea (NRF) grant and the Institute of Information & communications Technology Planning & Evaluation (IITP) grant both funded by the Korea government (MSIT) (No. RS-2022-NR071853, RS-2023-00222663, RS-2025-25463302), by the Basic Science Research Program through the NRF funded by the Ministry of Education (No. RS-2023-00275091), and by AI-Bio Research Grant through Seoul National University.

## USE OF LARGE LANGUAGE MODELS

Large Language Models (LLMs) were used solely as an assistive tool for writing. Specifically, we employed an LLM to improve clarity, grammar, and style of exposition. No part of the research ideation, algorithm design, theoretical analysis, or experimental results involved the use of LLMs. The authors take full responsibility for the content of the paper.

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

## A    PROOF OF TECHNICAL LEMMAS FOR THEOREMS

First, we provide a Lemma and a proof for the sampling error bound of the PQL operator. We assume the concentration properties of the reward function and the transition dynamics:

**Assumption 1** *Given a state-action pair $(s, a) \in \mathcal{D}$, the following relationships hold with probability at least $1 - \delta$,*

$$|r(s, a) - \hat{r}(s, a)| \leq \frac{C_r^\delta}{\sqrt{N(s, a)}}, \quad \left\| \mathcal{P}(\cdot \mid s, a) - \widehat{\mathcal{P}}(\cdot \mid s, a) \right\|_1 \leq \frac{C_\mathcal{P}^\delta}{\sqrt{N(s, a)}},$$

*where $C_r^\delta$ and $C_\mathcal{P}^\delta$ are constants that depend on $\delta \in (0, 1)$, $N(s, a)$ is the number of samples for $(s, a)$, and the concentration properties of $r$ and $\mathcal{T}$, respectively.*

Under Assumption 1 and Proposition 1, the sampling error between the empirical PQL operator and the actual PQL operator can be bounded, as shown in the following proof:

**Lemma 1 (Sampling Error Bound of the PQL operator)** *Given a state-action pair $(s, a) \in \mathcal{D}$, with probability at least $1 - \delta$, the sampling error between the empirical PQL operator and the actual PQL operator for $(s, a)$ satisfies the following inequality:*

$$\left| \mathcal{T}_\lambda^{\hat{\pi}_\beta, \pi} Q(s, a) - \widehat{\mathcal{T}}_\lambda^{\hat{\pi}_\beta, \pi} Q(s, a) \right| \leq \frac{C_r^\delta + \gamma C_\mathcal{P}^\delta R_{\max}/(1 - \gamma)}{(1 - \gamma\lambda)\sqrt{N(s, a)}},$$

*where $C_{r,\mathcal{P}}^\delta$ is a constant dependent on the concentration properties $r$ and $\mathcal{P}$, with $\delta \in (0, 1)$.*

**Proof** For $(s, a)$,

$$\left| \mathcal{T}_\lambda^{\hat{\pi}_\beta, \pi} Q(s, a) - \widehat{\mathcal{T}}_\lambda^{\hat{\pi}_\beta, \pi} Q(s, a) \right|$$

$$= \left| \left( \mathcal{I} - \gamma\lambda\mathcal{P}^{\hat{\pi}_\beta} \right)^{-1} (r + \gamma(1 - \lambda)\mathcal{P}^\pi Q(s, a)) - \left( \mathcal{I} - \gamma\lambda\widehat{\mathcal{P}}^{\hat{\pi}_\beta} \right)^{-1} \left( \hat{r} + \gamma(1 - \lambda)\widehat{\mathcal{P}}^\pi Q(s, a) \right) \right|$$

$$\leq \left| \left( \mathcal{I} - \gamma\lambda\mathcal{P}^{\hat{\pi}_\beta} \right)^{-1} (r + \gamma(1 - \lambda)\mathcal{P}^\pi Q(s, a)) - \left( \mathcal{I} - \gamma\lambda\mathcal{P}^{\hat{\pi}_\beta} \right)^{-1} \left( \hat{r} + \gamma(1 - \lambda)\widehat{\mathcal{P}}^\pi Q(s, a) \right) \right|$$

$$\quad + \left| \left( \mathcal{I} - \gamma\lambda\mathcal{P}^{\hat{\pi}_\beta} \right)^{-1} \left( \hat{r} + \gamma(1 - \lambda)\widehat{\mathcal{P}}^\pi Q(s, a) \right) - \left( \mathcal{I} - \gamma\lambda\widehat{\mathcal{P}}^{\hat{\pi}_\beta} \right)^{-1} \left( \hat{r} + \gamma(1 - \lambda)\widehat{\mathcal{P}}^\pi Q(s, a) \right) \right|$$

$$\leq \left| \left( \mathcal{I} - \gamma\lambda\mathcal{P}^{\hat{\pi}_\beta} \right)^{-1} \right| \left( |r(s, a) - \hat{r}(s, a)| + \gamma(1 - \lambda) \left\| \mathcal{P}^\pi(\cdot|s, a) - \widehat{\mathcal{P}}^\pi(\cdot|s, a) \right\|_1 Q(s, a) \right)$$

$$\quad + \left| \left( \mathcal{I} - \gamma\lambda\mathcal{P}^{\hat{\pi}_\beta} \right)^{-1} - \left( \mathcal{I} - \gamma\lambda\widehat{\mathcal{P}}^{\hat{\pi}_\beta} \right)^{-1} \right| \left| \hat{r} + \gamma(1 - \lambda)\widehat{\mathcal{P}}^\pi Q(s, a) \right|$$

$$\leq \left| \left( \mathcal{I} - \gamma\lambda\mathcal{P}^{\hat{\pi}_\beta} \right)^{-1} \right| \left( |r(s, a) - \hat{r}(s, a)| + \gamma(1 - \lambda) \left\| \mathcal{P}^\pi(\cdot|s, a) - \widehat{\mathcal{P}}^\pi(\cdot|s, a) \right\|_1 Q(s, a) \right)$$

$$\quad + \lambda\gamma \left| \left( \mathcal{I} - \gamma\lambda\mathcal{P}^{\hat{\pi}_\beta} \right)^{-1} \right| \left\| \mathcal{P}^{\hat{\pi}_\beta}(\cdot|s, a) - \widehat{\mathcal{P}}^{\hat{\pi}_\beta}(\cdot|s, a) \right\|_1 \left| \left( \mathcal{I} - \gamma\lambda\widehat{\mathcal{P}}^{\hat{\pi}_\beta} \right)^{-1} \right| \frac{(1 - \gamma\lambda)R_{\max}}{1 - \gamma}$$

$$\leq \frac{C_r^\delta + \gamma(1 - \lambda)C_\mathcal{P}^\delta R_{\max}/(1 - \gamma)}{(1 - \gamma\lambda)\sqrt{N(s, a)}} + \frac{\gamma\lambda C_\mathcal{P}^\delta R_{\max}/(1 - \gamma)}{(1 - \gamma\lambda)\sqrt{N(s, a)}}$$

$$\leq \frac{C_r^\delta + \gamma C_\mathcal{P}^\delta R_{\max}/(1 - \gamma)}{(1 - \gamma\lambda)\sqrt{N(s, a)}}.$$

This completes the proof of Lemma 1.                                                                            ■

Based on the interpretation of the sampling error of the PQL operator, if $\lambda$ is zero, the sampling error of the PQL operator is equivalent to that of the Bellman operator. For example, when $\lambda = 0$, the sampling error between the empirical PQL operator and the actual PQL operator for $(s, a)$ is bounded by $\frac{C_r^\delta + \gamma C_\mathcal{P}^\delta R_{\max}/(1-\gamma)}{\sqrt{N(s,a)}}$. This result aligns with the sampling error between the empirical Bellman operator and the actual Bellman operator (Section D.3 in Kumar et al. (2020)).

Now, we provide proofs for several technical lemmas that utilize our theorems, such as the construction of the conservative value estimation and the sub-optimality gap between the optimal and learned policies. In Lemma 2, $\mathbb{E}_{a \sim \pi(\cdot|s)} \left[ \frac{\pi(a|s)}{\hat{\pi}_\beta(a|s)} - 1 \right]$ has non-negative values for all states in $\mathcal{S}(\mathcal{D})$. In other words, $\mathbb{E}_{a \sim \pi(\cdot|s)} \left[ \frac{\pi(a|s)}{\hat{\pi}_\beta(a|s)} \right]$ is greater than or equal to 1 for any $\pi$.

**Lemma 2** *For any state $s$ and any two policies $\pi_1$ and $\pi_2$, the following inequality holds:*

$$\mathbb{E}_{a \sim \pi_1(\cdot|s)} \left[ \frac{\pi_1(a|s)}{\pi_2(a|s)} - 1 \right] \geq 0.$$

*with equality if and only if $\pi_1 = \pi_2$.*

**Proof** For any state $s$,

$$\begin{aligned}
\mathbb{E}_{a \sim \pi_1(\cdot|s)} \left[ \frac{\pi_1(a|s)}{\pi_2(a|s)} - 1 \right] &= \sum_a \pi_1(a|s) \left( \frac{\pi_1(a|s)}{\pi_2(a|s)} - 1 \right) \\
&= \sum_a (\pi_1(a|s) - \pi_2(a|s) + \pi_2(a|s)) \left( \frac{\pi_1(a|s)}{\pi_2(a|s)} - 1 \right) \\
&= \sum_a (\pi_1(a|s) - \pi_2(a|s)) \left( \frac{\pi_1(a|s)}{\pi_2(a|s)} - 1 \right) + \sum_a \pi_2(a|s) \left( \frac{\pi_1(a|s)}{\pi_2(a|s)} - 1 \right) \\
&= \sum_a (\pi_1(a|s) - \pi_2(a|s)) \left( \frac{\pi_1(a|s) - \pi_2(a|s)}{\pi_2(a|s)} \right) + \sum_a (\pi_1(a|s) - \pi_2(a|s)) \\
&= \sum_a \frac{(\pi_1(a|s) - \pi_2(a|s))^2}{\pi_2(a|s)} \\
&\geq 0,
\end{aligned}$$

where the last equality follows from the fact that $\pi_1(a|s)$ and $\pi_2(a|s)$ are positive values for all actions and $\sum_a \pi_1(a|s) = \sum_a \pi_2(a|s) = 1$. This concludes the proof. ∎

Next, two lemmas are adaptations of Lemma 3 from Achiam et al. (2017).

**Lemma 3** *For any two policies $\pi_1$ and $\pi_2$, the vector difference of the discounted future state visitation distributions on two different policies holds:*

$$d^{\pi_1} - d^{\pi_2} = \gamma \left( I - \gamma \mathcal{P}^{\pi_1} \right)^{-1} \left( \mathcal{P}^{\pi_1} - \mathcal{P}^{\pi_2} \right) d^{\pi_2}.$$

**Proof** Recall that the discounted state visitation distribution of a policy $\pi$, $d^\pi$, which is defined as

$$d^\pi(s) = (1 - \gamma) \sum_{t=0}^\infty \gamma^t \Pr(s_t = s \mid \pi, \mathcal{P}).$$

For finite state spaces, $d^\pi$ can be expressed in vector form as follows:

$$d^\pi = (1 - \gamma) \sum_{t=0}^\infty (\gamma \mathcal{P}^\pi)^t d_0 = (1 - \gamma) (I - \gamma \mathcal{P}^\pi)^{-1} d_0,$$

where $d_0$ is the initial state distribution. Then, we obtain

$$\begin{aligned}
d^{\pi_1} - d^{\pi_2} &= (1 - \gamma) \left[ (I - \gamma \mathcal{P}^{\pi_1})^{-1} - (I - \gamma \mathcal{P}^{\pi_2})^{-1} \right] d_0 \\
&= (1 - \gamma) (I - \gamma \mathcal{P}^{\pi_1})^{-1} \left[ (I - \gamma \mathcal{P}^{\pi_2}) - (I - \gamma \mathcal{P}^{\pi_1}) \right] (I - \gamma \mathcal{P}^{\pi_2})^{-1} d_0 \\
&= \gamma (1 - \gamma) (I - \gamma \mathcal{P}^{\pi_1})^{-1} (\mathcal{P}^{\pi_1} - \mathcal{P}^{\pi_2}) (I - \gamma \mathcal{P}^{\pi_2})^{-1} d_0 \\
&= \gamma (I - \gamma \mathcal{P}^{\pi_1})^{-1} (\mathcal{P}^{\pi_1} - \mathcal{P}^{\pi_2}) d^{\pi_2}.
\end{aligned}$$

This concludes the proof. ∎

**Lemma 4** *The divergence between discounted state visitation distributions, $||d^{\pi_1} - d^{\pi_2}||$, is bounded by an average divergence of the policies $\pi_1$ and $\pi_2$:*

$$||d^{\pi_1} - d^{\pi_2}||_1 \leq \frac{\gamma}{1-\gamma} \mathbb{E}_{s \sim d^{\pi_2}} \left[ \sum_a \left| \pi_1(a|s) - \pi_2(a|s) \right| \right]$$

$$= \frac{2\gamma}{1-\gamma} \mathbb{E}_{s \sim d^{\pi_2}} \left[ d_{\mathrm{TV}}(\pi_1, \pi_2)(s) \right],$$

*where $d_{\mathrm{TV}}(\pi_1, \pi_2)(s) = (1/2) \sum_a |\pi_1(a|s) - \pi_2(a|s)|$.*

**Proof** First, from Lemma 3, we obtain

$$||d^{\pi_1} - d^{\pi_2}||_1 = \gamma || (I - \gamma \mathcal{P}^{\pi_1})^{-1} (\mathcal{P}^{\pi_1} - \mathcal{P}^{\pi_2}) d^{\pi_2} ||_1$$

$$\leq \gamma || (I - \gamma \mathcal{P}^{\pi_1})^{-1} ||_1 || (\mathcal{P}^{\pi_1} - \mathcal{P}^{\pi_2}) d^{\pi_2} ||_1.$$

$|| (I - \gamma \mathcal{P}^{\pi_1})^{-1} ||_1$ is bounded by:

$$|| (I - \gamma \mathcal{P}^{\pi_1})^{-1} ||_1 \leq \sum_{t=0}^{\infty} \gamma^t \left( ||\mathcal{P}^{\pi_1}||_1 \right)^t = (1 - \gamma)^{-1}.$$

To conclude the lemma, we bound $|| (\mathcal{P}^{\pi_1} - \mathcal{P}^{\pi_2}) d^{\pi_2} ||_1$.

$$|| (\mathcal{P}^{\pi_1} - \mathcal{P}^{\pi_2}) d^{\pi_2} ||_1 = \sum_{s'} \left| \sum_s (\mathcal{P}^{\pi_1} - \mathcal{P}^{\pi_2}) d^{\pi_2} \right|$$

$$= \sum_{s,s'} |\mathcal{P}^{\pi_1} - \mathcal{P}^{\pi_2}| d^{\pi_2}$$

$$= \sum_{s,s'} \left| \sum_a \mathcal{P}(s'|s,a) (\pi_1(a|s) - \pi_2(a|s)) d^{\pi_2}(s) \right|$$

$$\leq \sum_{s,a,s'} \mathcal{P}(s'|s,a) |\pi_1(a|s) - \pi_2(a|s)| d^{\pi_2}(s)$$

$$\leq \sum_{s,a} |\pi_1(a|s) - \pi_2(a|s)| d^{\pi_2}(s)$$

$$= \mathbb{E}_{s \sim d^{\pi_2}} \left[ \sum_a \left| \pi_1(a|s) - \pi_2(a|s) \right| \right]$$

Therefore, we obtain that:

$$||d^{\pi_1} - d^{\pi_2}||_1 \leq \frac{\gamma}{1-\gamma} \mathbb{E}_{s \sim d^{\pi_2}} \left[ \sum_a \left| \pi_1(a|s) - \pi_2(a|s) \right| \right].$$

If we express this inequality in terms of the total variation distance, it becomes the following inequality:

$$||d^{\pi_1} - d^{\pi_2}||_1 \leq \frac{2\gamma}{1-\gamma} \mathbb{E}_{s \sim d^{\pi_2}} \left[ d_{\mathrm{TV}}(\pi_1, \pi_2)(s) \right].$$

This concludes the proof. ∎

We prove the following lemma, which bounds the difference between the expected discounted return under $\mathcal{M}$ and $\widehat{\mathcal{M}}$.

**Lemma 5** *Given any policy $\pi$, for any MDP $\mathcal{M}$ and the empirical MDP $\widehat{\mathcal{M}}$, the following holds with probability at least $1 - \delta$:*

$$\left| J_{\mathcal{M}}(\lambda\hat{\pi}_{\beta} + (1-\lambda)\pi) - J_{\widehat{\mathcal{M}}}(\lambda\hat{\pi}_{\beta} + (1-\lambda)\pi) \right|$$
$$\leq \frac{C_r^{\delta} + \gamma R_{\max} C_{\mathcal{P}}^{\delta}/(1-\gamma)}{1-\gamma} \mathbb{E}_{s \sim d_{\widehat{\mathcal{M}}}^{\lambda\hat{\pi}_{\beta} + (1-\lambda)\pi}(s)} \left[ \frac{\sqrt{|\mathcal{A}|}}{\sqrt{N(s)}} \left( \lambda + (1-\lambda)\sqrt{\mathbb{E}_{a \sim \pi(\cdot|s)}\left[ \frac{\pi(a|s)}{\hat{\pi}_{\beta}(a|s)} \right]} \right) \right].$$

**Proof** To prove this inequality, we use the triangle inequality to separate the gap in the expected discounted return into differences in rewards and transition dynamics, as follows:

$$\left| J_{\widehat{\mathcal{M}}}(\lambda\hat{\pi}_{\beta} + (1-\lambda)\pi) - J_{\mathcal{M}}(\lambda\hat{\pi}_{\beta} + (1-\lambda)\pi) \right|$$
$$= \frac{1}{1-\gamma} \left| \sum_{s,a} d_{\widehat{\mathcal{M}}}^{\lambda\hat{\pi}_{\beta} + (1-\lambda)\pi}(s)\left(\lambda\hat{\pi}_{\beta}(a|s) + (1-\lambda)\pi(a|s)\right) r_{\widehat{\mathcal{M}}}(s,a) \right.$$
$$\left. - \sum_{s,a} d_{\mathcal{M}}^{\lambda\hat{\pi}_{\beta}(a|s) + (1-\lambda)\pi(a|s)}(s)\left(\lambda\hat{\pi}_{\beta}(a|s) + (1-\lambda)\pi(a|s)\right) r_{\mathcal{M}}(s,a) \right|$$
$$\leq \frac{1}{1-\gamma} \left| \sum_{s,a} d_{\widehat{\mathcal{M}}}^{\lambda\hat{\pi}_{\beta} + (1-\lambda)\pi}(s) \underbrace{\left(\lambda\hat{\pi}_{\beta}(a|s) + (1-\lambda)\pi(a|s)\right)\left(r_{\widehat{\mathcal{M}}}(s,a) - r_{\mathcal{M}}(s,a)\right)}_{=:\Delta_r(s)} \right|$$
$$+ \frac{1}{1-\gamma} \left| \sum_{s,a} \underbrace{\left(d_{\widehat{\mathcal{M}}}^{\lambda\hat{\pi}_{\beta} + (1-\lambda)\pi}(s) - d_{\mathcal{M}}^{\lambda\hat{\pi}_{\beta} + (1-\lambda)\pi}(s)\right)}_{=:\Delta_d(s)} \pi(a|s) r_{\mathcal{M}}(s,a) \right|.$$

We first bound the term that includes the difference between the actual rewards and the estimated rewards by applying concentration inequalities to derive an upper bound for $\Delta_r(s)$. Note that under concentration assumptions, and using the fact that $\mathbb{E}[\Delta_r(s)] = 0$ in the limit of infinite data, we obtain:

$$|\Delta_r(s)| \leq \sum_a \left(\lambda\hat{\pi}_{\beta}(a|s) + (1-\lambda)\pi(a|s)\right)\left| r_{\widehat{\mathcal{M}}}(s,a) - r_{\mathcal{M}}(s,a) \right|$$
$$\leq \sum_a \left(\lambda\hat{\pi}_{\beta}(a|s) + (1-\lambda)\pi(a|s)\right) \frac{C_r^{\delta}}{\sqrt{N(s) \cdot \hat{\pi}_{\beta}(a|s)}}$$
$$= \frac{C_r^{\delta}}{\sqrt{N(s)}} \sum_a \left(\lambda\sqrt{\hat{\pi}_{\beta}(a|s)} + (1-\lambda)\frac{\pi(a|s)}{\sqrt{\hat{\pi}_{\beta}(a|s)}}\right)$$
$$\leq \frac{C_r^{\delta}}{\sqrt{N(s)}} \left(\lambda\sqrt{|\mathcal{A}|} + (1-\lambda)\sum_a \frac{\pi(a|s)}{\sqrt{\hat{\pi}_{\beta}(a|s)}}\right). \tag{3}$$

Next, we bound the term that involves the difference between the actual and estimated transition dynamics by applying concentration inequalities to derive an upper bound for $\Delta_d(s)$. By Lemma 3, we obtain the following equation:

$$\Delta_d = \gamma \left(\mathcal{I} - \gamma\mathcal{P}_{\widehat{\mathcal{M}}}^{\lambda\hat{\pi}_{\beta} + (1-\lambda)\pi}\right)^{-1} \underbrace{\left(\mathcal{P}_{\mathcal{M}}^{\lambda\hat{\pi}_{\beta} + (1-\lambda)\pi} - \mathcal{P}_{\widehat{\mathcal{M}}}^{\lambda\hat{\pi}_{\beta} + (1-\lambda)\pi}\right)}_{=:\Delta_P} d_{\widehat{\mathcal{M}}}^{\lambda\hat{\pi}_{\beta} + (1-\lambda)\pi}.$$

We know that $\gamma$ is positive and $||(I - \gamma \mathcal{P}^\pi)^{-1}||_1 \leq (1-\gamma)^{-1}$ for any policy $\pi$, we only need to bound the remaining terms.

$$\left\| \Delta_P \, d_{\widehat{\mathcal{M}}}^{\lambda \hat{\pi}_\beta + (1-\lambda)\pi} \right\|_1$$

$$= \sum_{s'} \left| \sum_s \Delta_P(s'|s) \, d_{\widehat{\mathcal{M}}}^{\lambda \hat{\pi}_\beta + (1-\lambda)\pi}(s) \right|$$

$$\leq \sum_{s',s} |\Delta_P(s'|s)| \, d_{\widehat{\mathcal{M}}}^{\lambda \hat{\pi}_\beta + (1-\lambda)\pi}$$

$$= \sum_{s',s} \left| \sum_a \left( \mathcal{P}_{\widehat{\mathcal{M}}}(s'|s,a) - \mathcal{P}_{\mathcal{M}}(s'|s,a) \right) (\lambda \hat{\pi}_\beta(a|s) + (1-\lambda)\pi(a|s)) \right| d_{\widehat{\mathcal{M}}}^{\lambda \hat{\pi}_\beta + (1-\lambda)\pi}(s)$$

$$\leq \sum_{s',s} \left\| \mathcal{P}_{\widehat{\mathcal{M}}}(\cdot|s,a) - \mathcal{P}_{\mathcal{M}}(\cdot|s,a) \right\|_1 (\lambda \hat{\pi}_\beta(a|s) + (1-\lambda)\pi(a|s)) \, d_{\widehat{\mathcal{M}}}^{\lambda \hat{\pi}_\beta + (1-\lambda)\pi}(s)$$

$$\leq \sum_s d_{\widehat{\mathcal{M}}}^{\lambda \hat{\pi}_\beta + (1-\lambda)\pi}(s) \frac{C_\mathcal{P}^\delta}{\sqrt{N(s)}} \sum_a \frac{\lambda \hat{\pi}_\beta(a|s) + (1-\lambda)\pi(a|s)}{\sqrt{\hat{\pi}_\beta(a|s)}}$$

$$= \sum_s d_{\widehat{\mathcal{M}}}^{\lambda \hat{\pi}_\beta + (1-\lambda)\pi}(s) \frac{C_\mathcal{P}^\delta}{\sqrt{N(s)}} \sum_a \left( \lambda \sqrt{\hat{\pi}_\beta(a|s)} + (1-\lambda) \frac{\pi(a|s)}{\sqrt{\hat{\pi}_\beta(a|s)}} \right)$$

$$\leq \sum_s d_{\widehat{\mathcal{M}}}^{\lambda \hat{\pi}_\beta + (1-\lambda)\pi}(s) \frac{C_\mathcal{P}^\delta}{\sqrt{N(s)}} \left( \lambda \sqrt{|\mathcal{A}|} + (1-\lambda) \sum_a \frac{\pi(a|s)}{\sqrt{\hat{\pi}_\beta(a|s)}} \right)$$

where the last inequality is derived from the Cauchy–Schwarz inequality. Hence, we can bound $\Delta_d(s)$ as follows:

$$|\Delta_d(s)| \leq \frac{\gamma C_\mathcal{P}^\delta}{1-\gamma} \sum_s d_{\widehat{\mathcal{M}}}^{\lambda \hat{\pi}_\beta + (1-\lambda)\pi}(s) \frac{1}{\sqrt{N(s)}} \left( \lambda \sqrt{|\mathcal{A}|} + (1-\lambda) \sum_a \frac{\pi(a|s)}{\sqrt{\hat{\pi}_\beta(a|s)}} \right). \quad (4)$$

To derive the final upper bound of the objective function, it is necessary to bound $\sum_a \frac{\pi(a|s)}{\sqrt{\hat{\pi}_\beta(a|s)}}$, as follows:

$$\mathbb{E}_{a \sim \pi(\cdot|s)} \left[ \frac{\pi(a|s)}{\hat{\pi}_\beta(a|s)} \right] = \sum_a \frac{(\pi(a|s))^2}{\hat{\pi}_\beta(a|s)} = \sum_a \left( \frac{\pi(a|s)}{\sqrt{\hat{\pi}_\beta(a|s)}} \right)^2$$

$$\leq \left( \sum_a \frac{\pi(a|s)}{\sqrt{\hat{\pi}_\beta(a|s)}} \right)^2 \leq |\mathcal{A}| \, \mathbb{E}_{a \sim \pi(\cdot|s)} \left[ \frac{\pi(a|s)}{\hat{\pi}_\beta(a|s)} \right]$$

Then we obtain

$$\sqrt{\mathbb{E}_{a \sim \pi(\cdot|s)} \left[ \frac{\pi(a|s)}{\hat{\pi}_\beta(a|s)} \right]} \leq \sum_a \frac{\pi(a|s)}{\sqrt{\hat{\pi}_\beta(a|s)}} \leq \sqrt{|\mathcal{A}| \, \mathbb{E}_{a \sim \pi(\cdot|s)} \left[ \frac{\pi(a|s)}{\hat{\pi}_\beta(a|s)} \right]}. \quad (5)$$

By Equation 3, Equation 4, and Equation 5, we have that:

$$\left| J_{\mathcal{M}}(\lambda \hat{\pi}_\beta + (1-\lambda)\pi) - J_{\widehat{\mathcal{M}}}(\lambda \hat{\pi}_\beta + (1-\lambda)\pi) \right|$$

$$\leq \frac{C_r^\delta + \gamma R_{\max} C_\mathcal{P}^\delta / (1-\gamma)}{1-\gamma} \mathbb{E}_{s \sim d_{\widehat{\mathcal{M}}}^{\lambda \hat{\pi}_\beta + (1-\lambda)\pi}(s)} \left[ \frac{\sqrt{|\mathcal{A}|}}{\sqrt{N(s)}} \left( \lambda + (1-\lambda) \sqrt{\mathbb{E}_{a \sim \pi(\cdot|s)} \left[ \frac{\pi(a|s)}{\hat{\pi}_\beta(a|s)} \right]} \right) \right]$$

$$\quad (6)$$

Equation 6 reflects the trade-off between 1 and $\sqrt{\mathbb{E}_{a \sim \pi(\cdot|s)} \left[ \frac{\pi(a|s)}{\hat{\pi}_\beta(a|s)} \right]} (\geq 1)$, by weighting them with $\lambda$ and $1-\lambda$, respectively. This bound can be tighter than the single-step counterpart when the density-ratio term is controlled. This completes the proof of Lemma 5. ∎

## B    PROOF OF THEOREMS

In this appendix, we provide all proofs of our main theorems with sampling error.

### B.1    THEOREM 1

**Theorem 1 (Lower Bound on the State Value Function of CPQL)** *Let $\widehat{Q}^{\lambda\hat{\pi}_\beta+(1-\lambda)\pi}$ denote the Q-function derived from CPQL as defined in Equation 2. Then, the state value of $\lambda\hat{\pi}_\beta + (1-\lambda)\pi$, $\widehat{V}^{\lambda\hat{\pi}_\beta+(1-\lambda)\pi}(s) = \mathbb{E}_{a\sim(\lambda\hat{\pi}_\beta+(1-\lambda)\pi)(\cdot|s)}\left[\widehat{Q}^{\lambda\hat{\pi}_\beta+(1-\lambda)\pi}(s,a)\right]$, lower bounds the true state value of the policy obtained via exact policy evaluation, $V^{\lambda\hat{\pi}_\beta+(1-\lambda)\pi}(s)$, for sufficiently large $\alpha$. Formally, with probability at least $1 - \delta$, for all $s \in \mathcal{S}(\mathcal{D})$,*

$$\widehat{V}^{\lambda\hat{\pi}_\beta+(1-\lambda)\pi}(s) \le V^{\lambda\hat{\pi}_\beta+(1-\lambda)\pi}(s),$$

*if $\alpha \ge \frac{C_r^\delta+\gamma R_{\max}C_\mathcal{P}^\delta/(1-\gamma)}{(1-\gamma\lambda)(1-\lambda)(1-\gamma)}\max_{s,a\in\mathcal{D}}\frac{1}{\sqrt{N(s,a)}}\max_{s\in\mathcal{S}(\mathcal{D})}\left(\mathbb{E}_{a\sim\pi(\cdot|s)}\left[\frac{\pi(a|s)}{\hat{\pi}_\beta(a|s)}-1\right]\right)^{-1}$*

**Proof**    By setting the derivative of Equation 2 to zero, we obtain the following recursive update expression for $\widehat{Q}_{k+1}$ in terms of $\widehat{Q}_k$, incorporating the sampling error under Lemma 1. Given a state-action pair $(s, a)$, with high probability $\ge 1 - \delta$:

$$\widehat{Q}_{k+1}(s,a) = \widehat{\mathcal{T}}_\lambda^{\hat{\pi}_\beta,\pi_k}\widehat{Q}_k(s,a) - \alpha\left[\frac{\pi_k(a|s)}{\hat{\pi}_\beta(a|s)}-1\right]$$

$$\le \mathcal{T}_\lambda^{\hat{\pi}_\beta,\pi_k}\widehat{Q}_k(s,a) - \alpha\left[\frac{\pi_k(a|s)}{\hat{\pi}_\beta(a|s)}-1\right] + \frac{C_{r,\mathcal{P}}^\delta}{(1-\gamma\lambda)(1-\gamma)\sqrt{N(s,a)}}.$$

In Proposition 2, we know that $\lim_{k\to\infty}\widehat{Q}_k = \widehat{Q}^{\lambda\hat{\pi}_\beta+(1-\lambda)\pi}$ when the function approximation error is zero for every $(s, a) \in \mathcal{S} \times \mathcal{A}$. Thus, the state value function of $\lambda\hat{\pi}_\beta + (1-\lambda)\pi_k$, on the other hand, $\widehat{V}_{k+1}$ is underestimated, since:

$$\widehat{V}_{k+1}(s) = \mathcal{T}_\lambda^{\hat{\pi}_\beta,\pi_k}\widehat{V}_k(s) - \alpha\mathbb{E}_{a\sim(\lambda\hat{\pi}_\beta+(1-\lambda)\pi_k)(\cdot|s)}\left[\frac{\pi_k(a|s)}{\hat{\pi}_\beta(a|s)}-1\right]$$

$$+ \mathbb{E}_{a\sim(\lambda\hat{\pi}_\beta+(1-\lambda)\pi_k)(\cdot|s)}\left[\frac{C_{r,\mathcal{P}}^\delta}{(1-\gamma\lambda)(1-\gamma)\sqrt{N(s,a)}}\right].$$

Now, we can compute the fixed point of the recursion in the above equation. Because the fixed point of the PQL operator coincides with the unique fixed point of $\mathcal{T}^{\lambda\hat{\pi}_\beta+(1-\lambda)\pi}$, this gives us the following estimated policy value:

$$\widehat{V}^{\lambda\hat{\pi}_\beta+(1-\lambda)\pi}(s)$$

$$= V^{\lambda\hat{\pi}_\beta+(1-\lambda)\pi}(s) - \alpha\left[\left(\mathcal{I}-\gamma\mathcal{P}^{\lambda\hat{\pi}_\beta+(1-\lambda)\pi}\right)^{-1}\mathbb{E}_{a\sim(\lambda\hat{\pi}_\beta+(1-\lambda)\pi)(\cdot|s)}\left[\frac{\pi(a|s)}{\hat{\pi}_\beta(a|s)}-1\right]\right](s)$$

$$+ \left[\left(\mathcal{I}-\gamma\mathcal{P}^{\lambda\hat{\pi}_\beta+(1-\lambda)\pi}\right)^{-1}\mathbb{E}_{a\sim(\lambda\hat{\pi}_\beta+(1-\lambda)\pi_k)(\cdot|s)}\left[\frac{C_{r,\mathcal{P}}^\delta}{(1-\gamma\lambda)(1-\gamma)\sqrt{N(s,a)}}\right]\right](s). \quad (7)$$

In this case, the choice of $\alpha$, that prevents overestimation is given by:

$$\alpha \ge \frac{C_r^\delta + \gamma R_{\max}C_\mathcal{P}^\delta/(1-\gamma)}{(1-\gamma\lambda)(1-\lambda)(1-\gamma)}\max_{s,a\in\mathcal{D}}\frac{1}{\sqrt{N(s,a)}}\max_{s\in\mathcal{S}(\mathcal{D})}\left(\mathbb{E}_{a\sim\pi(\cdot|s)}\left[\frac{\pi(a|s)}{\hat{\pi}_\beta(a|s)}-1\right]\right)^{-1}$$

This completes the proof of Theorem 1.                                        ∎

## B.2 THEOREM 2

We prove that $\lambda\hat{\pi}_\beta + (1-\lambda)\hat{\pi}$ achieves at least the performance of $\hat{\pi}_\beta$ in the actual MDP $\mathcal{M}$.

**Theorem 2 (Comparison to the Behavior Policy)** *Let* $\hat{\pi} := \mathrm{argmax}_\pi \mathbb{E}_{s\sim d_0}\left[\widehat{V}^{\lambda\hat{\pi}_\beta+(1-\lambda)\pi}(s)\right]$. *With probability at least* $1-\delta$, $\lambda\hat{\pi}_\beta + (1-\lambda)\hat{\pi}$ *achieves a policy improvement over* $\hat{\pi}_\beta$ *in the actual MDP* $\mathcal{M}$ *as follows:*

$$
J_\mathcal{M}\left(\lambda\hat{\pi}_\beta + (1-\lambda)\hat{\pi}\right) \geq J_\mathcal{M}\left(\hat{\pi}_\beta\right) + \frac{\alpha(1-\lambda)}{1-\gamma}\mathbb{E}_{s\sim d_{\widehat{\mathcal{M}}}^{\lambda\hat{\pi}_\beta+(1-\lambda)\hat{\pi}}(s)}\left[\mathbb{E}_{a\sim\hat{\pi}(\cdot|s)}\left[\frac{\hat{\pi}(a|s)}{\hat{\pi}_\beta(a|s)} - 1\right]\right]
$$
$$
- \frac{C_{r,\mathcal{P}}^\delta}{1-\gamma}\mathbb{E}_{s\sim d_{\widehat{\mathcal{M}}}^{\lambda\hat{\pi}_\beta+(1-\lambda)\hat{\pi}}(s)}\left[\frac{\sqrt{|\mathcal{A}|}}{\sqrt{N(s)}}\left(1+\lambda+(1-\lambda)\sqrt{\mathbb{E}_{a\sim\hat{\pi}(\cdot|s)}\left[\frac{\hat{\pi}(a|s)}{\hat{\pi}_\beta(a|s)}\right]}\right)\right],
$$

*where* $C_{r,\mathcal{P}}^\delta$ *is a constant dependent on the concentration properties* $r$ *and* $\mathcal{P}$.

**Proof** The proof of this statement is divided into three parts:

$$
J_\mathcal{M}\left(\lambda\hat{\pi}_\beta + (1-\lambda)\hat{\pi}\right) - J_\mathcal{M}\left(\hat{\pi}_\beta\right)
$$
$$
= \underbrace{J_\mathcal{M}\left(\lambda\hat{\pi}_\beta + (1-\lambda)\hat{\pi}\right) - J_{\widehat{\mathcal{M}}}\left(\lambda\hat{\pi}_\beta + (1-\lambda)\hat{\pi}\right)}_{=:\Delta_1}
$$
$$
+ \underbrace{J_{\widehat{\mathcal{M}}}\left(\lambda\hat{\pi}_\beta + (1-\lambda)\hat{\pi}\right) - J_{\widehat{\mathcal{M}}}\left(\hat{\pi}_\beta\right)}_{=:\Delta_2} + \underbrace{J_{\widehat{\mathcal{M}}}\left(\hat{\pi}_\beta\right) - J_\mathcal{M}\left(\hat{\pi}_\beta\right)}_{=:\Delta_3}.
$$

By Lemma 5, we obtain the upper bound of $\Delta_1$ and $\Delta_3$, as follows:

$$
|\Delta_1| \leq \frac{C_r^\delta + \gamma R_{\max} C_\mathcal{P}^\delta/(1-\gamma)}{1-\gamma}\mathbb{E}_{s\sim d_{\widehat{\mathcal{M}}}^{\lambda\hat{\pi}_\beta+(1-\lambda)\hat{\pi}}(s)}\left[\frac{\sqrt{|\mathcal{A}|}}{\sqrt{N(s)}}\left(\lambda+(1-\lambda)\sqrt{\mathbb{E}_{a\sim\hat{\pi}(\cdot|s)}\left[\frac{\hat{\pi}(a|s)}{\hat{\pi}_\beta(a|s)}\right]}\right)\right]
$$
$$
|\Delta_3| \leq \frac{C_r^\delta + \gamma R_{\max} C_\mathcal{P}^\delta/(1-\gamma)}{1-\gamma}\mathbb{E}_{s\sim d_{\widehat{\mathcal{M}}}^{\hat{\pi}_\beta}(s)}\left[\frac{\sqrt{|\mathcal{A}|}}{\sqrt{N(s)}}\right].
$$

Next, we obtain the lower bound of $\Delta_2$ by the definition of $\hat{\pi}$ and Equation 7:

$$
J_{\widehat{\mathcal{M}}}\left(\lambda\hat{\pi}_\beta + (1-\lambda)\hat{\pi}\right) - \frac{\alpha(1-\lambda)}{1-\gamma}\mathbb{E}_{s\sim d_{\widehat{\mathcal{M}}}^{\lambda\hat{\pi}_\beta+(1-\lambda)\hat{\pi}}(s)}\left[\mathbb{E}_{a\sim\hat{\pi}(\cdot|s)}\left[\frac{\hat{\pi}(a|s)}{\hat{\pi}_\beta(a|s)} - 1\right]\right] \geq J_{\widehat{\mathcal{M}}}\left(\hat{\pi}_\beta\right) - 0.
$$

Thus, we have that:

$$
\Delta_2 \geq \frac{\alpha(1-\lambda)}{1-\gamma}\mathbb{E}_{s\sim d_{\widehat{\mathcal{M}}}^{\lambda\hat{\pi}_\beta+(1-\lambda)\hat{\pi}}(s)}\left[\mathbb{E}_{a\sim\hat{\pi}(\cdot|s)}\left[\frac{\hat{\pi}(a|s)}{\hat{\pi}_\beta(a|s)} - 1\right]\right]
$$

Therefore, by integrating the bound of $\Delta_1$, $\Delta_2$, and $\Delta_3$, we obtain that:

$$
J_\mathcal{M}\left(\lambda\hat{\pi}_\beta + (1-\lambda)\hat{\pi}\right)
$$
$$
\geq J_\mathcal{M}\left(\hat{\pi}_\beta\right) + \frac{\alpha(1-\lambda)}{1-\gamma}\mathbb{E}_{s\sim d_{\widehat{\mathcal{M}}}^{\lambda\hat{\pi}_\beta+(1-\lambda)\hat{\pi}}(s)}\left[\mathbb{E}_{a\sim\hat{\pi}(\cdot|s)}\left[\frac{\hat{\pi}(a|s)}{\hat{\pi}_\beta(a|s)} - 1\right]\right]
$$
$$
- \frac{C_{r,\mathcal{P}}^\delta}{1-\gamma}\mathbb{E}_{s\sim d_{\widehat{\mathcal{M}}}^{\lambda\hat{\pi}_\beta+(1-\lambda)\hat{\pi}}(s)}\left[\frac{\sqrt{|\mathcal{A}|}}{\sqrt{N(s)}}\left(1+\lambda+(1-\lambda)\sqrt{\mathbb{E}_{a\sim\hat{\pi}(\cdot|s)}\left[\frac{\hat{\pi}(a|s)}{\hat{\pi}_\beta(a|s)}\right]}\right)\right],
$$

where $C_{r,\mathcal{P}}^\delta = C_r^\delta + \gamma R_{\max} C_\mathcal{P}^\delta/(1-\gamma)$.

This completes the proof of Theorem 2. ∎

When $\lambda = 0$, we obtain the following lower bound:

$$
J_\mathcal{M}\left(\hat{\pi}\right) - J_\mathcal{M}\left(\hat{\pi}_\beta\right)
$$
$$
\geq \frac{\alpha}{1-\gamma}\mathbb{E}_{s\sim d_{\widehat{\mathcal{M}}}^{\hat{\pi}}(s)}\left[\mathbb{E}_{a\sim\hat{\pi}(\cdot|s)}\left[\frac{\hat{\pi}(a|s)}{\hat{\pi}_\beta(a|s)} - 1\right]\right] - \frac{2C_{r,\mathcal{P}}^\delta}{1-\gamma}\mathbb{E}_{s\sim d_{\widehat{\mathcal{M}}}^{\hat{\pi}}(s)}\left[\frac{\sqrt{|\mathcal{A}|}}{\sqrt{N(s)}}\sqrt{\mathbb{E}_{a\sim\hat{\pi}(\cdot|s)}\left[\frac{\hat{\pi}(a|s)}{\hat{\pi}_\beta(a|s)}\right]}\right].
$$

This result coincides with Theorem 3.6 from CQL (Kumar et al., 2020). Our theorem converges under the same conditions, thereby ensuring consistency with the CQL framework.

## B.3 Theorem 3

We first present, to the best of our knowledge, theoretical guarantees concerning the sub-optimality gap between the optimal policy and the mixture policy.

**Theorem 3 (Sub-Optimality Gap)** *With probability at least $1-\delta$, the gap of the expected discounted return between the optimal policy $\pi^*$ and the mixture policy $\lambda\hat{\pi}_\beta + (1-\lambda)\hat{\pi}$ under the actual MDP $\mathcal{M}$ satisfies*

$$
J_{\mathcal{M}}\left(\pi^*\right) - J_{\mathcal{M}}\left(\lambda\hat{\pi}_\beta + (1-\lambda)\hat{\pi}\right)
$$
$$
\leq \frac{2\lambda R_{\max}}{(1-\gamma)^2}\mathbb{E}_{s\sim d_{\widehat{\mathcal{M}}}^{\lambda\hat{\pi}_\beta+(1-\lambda)\pi^*}}\left[d_{\mathrm{TV}}\left(\pi^*,\hat{\pi}_\beta\right)(s)\right]
$$
$$
+ \frac{2\alpha(1-\lambda)}{1-\gamma}\mathbb{E}_{s\sim d_{\widehat{\mathcal{M}}}^{\lambda\hat{\pi}_\beta+(1-\lambda)\pi^*}(s)}\left[d_{\mathrm{TV}}\left(\pi^*,\hat{\pi}\right)(s)\left(\xi(\hat{\pi})(s)+\frac{\gamma}{1-\gamma}\mathbb{E}_{a\sim\hat{\pi}(\cdot|s)}\left[\frac{\hat{\pi}(a|s)}{\hat{\pi}_\beta(a|s)}-1\right]\right)\right]
$$
$$
+ \frac{C_{r,\mathcal{P}}^\delta}{1-\gamma}\mathbb{E}_{s\sim d_{\widehat{\mathcal{M}}}^{\lambda\hat{\pi}_\beta+(1-\lambda)\hat{\pi}}(s)}\left[\frac{\sqrt{|\mathcal{A}|}}{\sqrt{N(s)}}\left(\lambda+(1-\lambda)\sqrt{\mathbb{E}_{a\sim\hat{\pi}(\cdot|s)}\left[\frac{\hat{\pi}(a|s)}{\hat{\pi}_\beta(a|s)}\right]}\right)\right]
$$
$$
+ \frac{C_{r,\mathcal{P}}^\delta}{1-\gamma}\mathbb{E}_{s\sim d_{\widehat{\mathcal{M}}}^{\pi^*}(s)}\left[\frac{\sqrt{|\mathcal{A}|}}{\sqrt{N(s)}}\sqrt{\mathbb{E}_{a\sim\pi^*(\cdot|s)}\left[\frac{\pi^*(a|s)}{\hat{\pi}_\beta(a|s)}\right]}\right],
$$

*where $\xi(\hat{\pi})(s):=\sum_{a\in\mathcal{A}}\frac{\pi^*(a|s)+\hat{\pi}(a|s)}{\hat{\pi}_\beta(a|s)}$ and $d_{\mathrm{TV}}\left(\pi_1,\pi_2\right)$ is the total variation distance of $\pi_1$ and $\pi_2$.*

**Proof** The proof of this statement is divided into four parts:

$$
J_{\mathcal{M}}\left(\pi^*\right) - J_{\mathcal{M}}\left(\lambda\hat{\pi}_\beta + (1-\lambda)\hat{\pi}\right)
$$
$$
= \underbrace{J_{\mathcal{M}}\left(\pi^*\right) - J_{\widehat{\mathcal{M}}}\left(\pi^*\right)}_{=:\Delta_1} + \underbrace{J_{\widehat{\mathcal{M}}}\left(\pi^*\right) - J_{\widehat{\mathcal{M}}}\left(\lambda\hat{\pi}_\beta + (1-\lambda)\pi^*\right)}_{=:\Delta_2}
$$
$$
+ \underbrace{J_{\widehat{\mathcal{M}}}\left(\lambda\hat{\pi}_\beta + (1-\lambda)\pi^*\right) - J_{\widehat{\mathcal{M}}}\left(\lambda\hat{\pi}_\beta + (1-\lambda)\hat{\pi}\right)}_{=:\Delta_3}
$$
$$
+ \underbrace{J_{\widehat{\mathcal{M}}}\left(\lambda\hat{\pi}_\beta + (1-\lambda)\hat{\pi}\right) - J_{\mathcal{M}}\left(\lambda\hat{\pi}_\beta + (1-\lambda)\hat{\pi}\right)}_{=:\Delta_4}.
$$

By Lemma 5, we obtain the upper bound of $\Delta_1$ and $\Delta_4$, as follows:

$$
|\Delta_1| \leq \frac{C_r^\delta + \gamma R_{\max}C_{\mathcal{P}}^\delta/(1-\gamma)}{1-\gamma}\mathbb{E}_{s\sim d_{\widehat{\mathcal{M}}}^{\pi^*}(s)}\left[\frac{\sqrt{|\mathcal{A}|}}{\sqrt{N(s)}}\left(\sqrt{\mathbb{E}_{a\sim\pi^*(\cdot|s)}\left[\frac{\pi^*(a|s)}{\hat{\pi}_\beta(a|s)}\right]}\right)\right]
$$
$$
|\Delta_4| \leq \frac{C_r^\delta + \gamma R_{\max}C_{\mathcal{P}}^\delta/(1-\gamma)}{1-\gamma}\mathbb{E}_{s\sim d_{\widehat{\mathcal{M}}}^{\lambda\hat{\pi}_\beta+(1-\lambda)\hat{\pi}}(s)}\left[\frac{\sqrt{|\mathcal{A}|}}{\sqrt{N(s)}}\left(\lambda+(1-\lambda)\sqrt{\mathbb{E}_{a\sim\hat{\pi}(\cdot|s)}\left[\frac{\hat{\pi}(a|s)}{\hat{\pi}_\beta(a|s)}\right]}\right)\right]
$$

Next, we derive the upper bound of $\Delta_2$, as follows:

$$
|\Delta_2| = \frac{1}{1-\gamma}\left|\sum_{s,a}\left(d_{\widehat{\mathcal{M}}}^{\pi^*}(s)\pi^*(a|s) - d_{\widehat{\mathcal{M}}}^{\lambda\hat{\pi}_\beta+(1-\lambda)\pi^*}(s)(\lambda\hat{\pi}_\beta+(1-\lambda)\pi^*)(a|s)\right)r_{\widehat{\mathcal{M}}}(s,a)\right|
$$
$$
\leq \frac{R_{\max}}{1-\gamma}\left|\sum_s\left(d_{\widehat{\mathcal{M}}}^{\pi^*}(s) - d_{\widehat{\mathcal{M}}}^{\lambda\hat{\pi}_\beta+(1-\lambda)\pi^*}(s)\right)\right| + \frac{\lambda R_{\max}}{1-\gamma}\left|\sum_{s,a}d_{\widehat{\mathcal{M}}}^{\lambda\hat{\pi}_\beta+(1-\lambda)\pi^*}(s)(\pi^*-\hat{\pi}_\beta)(s,a)\right|
$$
$$
= \frac{\gamma\lambda R_{\max}}{(1-\gamma)^2}\mathbb{E}_{s\sim d_{\widehat{\mathcal{M}}}^{\lambda\hat{\pi}_\beta+(1-\lambda)\pi^*}}\left[\sum_a\left|\pi^*(a|s)-\hat{\pi}_\beta(a|s)\right|\right] + \frac{\lambda R_{\max}}{1-\gamma}\mathbb{E}_{s\sim d_{\widehat{\mathcal{M}}}^{\lambda\hat{\pi}_\beta+(1-\lambda)\pi^*}}\left[\sum_a\left|\pi^*(a|s)-\hat{\pi}_\beta(a|s)\right|\right]
$$
$$
= \frac{2\lambda R_{\max}}{(1-\gamma)^2}\mathbb{E}_{s\sim d_{\widehat{\mathcal{M}}}^{\lambda\hat{\pi}_\beta+(1-\lambda)\pi^*}}\left[d_{\mathrm{TV}}\left(\pi^*,\hat{\pi}_\beta\right)(s)\right],
$$

where the second inequality follows from Lemma 4 and the last equality holds with the definition of total variation distance, $d_{\mathrm{TV}}\left(\pi_1,\pi_2\right)(s) = \sum_a\left|\pi_1(a|s)-\pi_2(a|s)\right|/2$.

By the definition of $\hat{\pi}$ and Equation 7, we derive the upper bound of $\Delta_3$, as follows:

$$
|\Delta_3| \leq \frac{\alpha}{1-\gamma} \left| \mathbb{E}_{s \sim d_{\widehat{\mathcal{M}}}^{\lambda\hat{\pi}_\beta+(1-\lambda)\pi^*}(s)} \left[ \mathbb{E}_{a \sim (\lambda\hat{\pi}_\beta+(1-\lambda)\pi^*)(\cdot|s)} \left[ \frac{\pi^*(a|s)}{\hat{\pi}_\beta(a|s)} - 1 \right] \right] \right.
$$

$$
\left. - \mathbb{E}_{s \sim d_{\widehat{\mathcal{M}}}^{\lambda\hat{\pi}_\beta+(1-\lambda)\hat{\pi}}(s)} \left[ \mathbb{E}_{a \sim (\lambda\hat{\pi}_\beta+(1-\lambda)\hat{\pi})(\cdot|s)} \left[ \frac{\hat{\pi}(a|s)}{\hat{\pi}_\beta(a|s)} - 1 \right] \right] \right|
$$

$$
\leq \frac{\alpha(1-\lambda)}{1-\gamma} \left| \mathbb{E}_{s \sim d_{\widehat{\mathcal{M}}}^{\lambda\hat{\pi}_\beta+(1-\lambda)\pi^*}(s)} \left[ \mathbb{E}_{a \sim \pi^*(\cdot|s)} \left[ \frac{\pi^*(a|s)}{\hat{\pi}_\beta(a|s)} \right] - \mathbb{E}_{a \sim \hat{\pi}(\cdot|s)} \left[ \frac{\hat{\pi}(a|s)}{\hat{\pi}_\beta(a|s)} \right] \right] \right|
$$

$$
+ \frac{\alpha(1-\lambda)}{1-\gamma} \sum_s \left| d_{\widehat{\mathcal{M}}}^{\lambda\hat{\pi}_\beta+(1-\lambda)\hat{\pi}}(s) - d_{\widehat{\mathcal{M}}}^{\lambda\hat{\pi}_\beta+(1-\lambda)\pi^*}(s) \right| \mathbb{E}_{a \sim \hat{\pi}(\cdot|s)} \left[ \frac{\hat{\pi}(a|s)}{\hat{\pi}_\beta(a|s)} - 1 \right]
$$

$$
\leq \frac{\alpha(1-\lambda)}{1-\gamma} \mathbb{E}_{s \sim d_{\widehat{\mathcal{M}}}^{\lambda\hat{\pi}_\beta+(1-\lambda)\pi^*}(s)} \left[ \sum_a \frac{(\pi^*(a|s) + \hat{\pi}(a|s)) |\pi^*(a|s) - \hat{\pi}(a|s)|}{\hat{\pi}_\beta(a|s)} \right]
$$

$$
+ \frac{\alpha(1-\lambda)}{1-\gamma} \sum_s \left| d_{\widehat{\mathcal{M}}}^{\lambda\hat{\pi}_\beta+(1-\lambda)\hat{\pi}}(s) - d_{\widehat{\mathcal{M}}}^{\lambda\hat{\pi}_\beta+(1-\lambda)\pi^*}(s) \right| \mathbb{E}_{a \sim \hat{\pi}(\cdot|s)} \left[ \frac{\hat{\pi}(a|s)}{\hat{\pi}_\beta(a|s)} - 1 \right]
$$

$$
\leq \frac{\alpha(1-\lambda)}{1-\gamma} \mathbb{E}_{s \sim d_{\widehat{\mathcal{M}}}^{\lambda\hat{\pi}_\beta+(1-\lambda)\pi^*}(s)} \left[ \underbrace{\sum_a \frac{\pi^*(a|s) + \hat{\pi}(a|s)}{\hat{\pi}_\beta(a|s)} \sum_a |\pi^*(a|s) - \hat{\pi}(a|s)|}_{:=\xi(\hat{\pi})(s)} \right]
$$

$$
+ \frac{\alpha\gamma(1-\lambda)}{(1-\gamma)^2} \mathbb{E}_{s \sim d_{\widehat{\mathcal{M}}}^{\lambda\hat{\pi}_\beta+(1-\lambda)\pi^*}(s)} \left[ \mathbb{E}_{a \sim \hat{\pi}(\cdot|s)} \left[ \frac{\hat{\pi}(a|s)}{\hat{\pi}_\beta(a|s)} - 1 \right] \sum_a \left| \hat{\pi}(a|s) - \pi^*(a|s) \right| \right],
$$

$$
\leq \frac{2\alpha(1-\lambda)}{1-\gamma} \mathbb{E}_{s \sim d_{\widehat{\mathcal{M}}}^{\lambda\hat{\pi}_\beta+(1-\lambda)\pi^*}(s)} \left[ d_{\mathrm{TV}}(\pi^*, \hat{\pi})(s) \left( \xi(\hat{\pi})(s) + \frac{\gamma}{1-\gamma} \mathbb{E}_{a \sim \hat{\pi}(\cdot|s)} \left[ \frac{\hat{\pi}(a|s)}{\hat{\pi}_\beta(a|s)} - 1 \right] \right) \right],
$$

where the last inequality follows from the definition of $\xi(\hat{\pi})$ and Lemma 4.

Therefore, by integrating the bound of $\Delta_1$, $\Delta_2$, $\Delta_3$ and $\Delta_4$, we have that:

$$
J_{\mathcal{M}}(\pi^*) - J_{\mathcal{M}}(\lambda\hat{\pi}_\beta + (1-\lambda)\hat{\pi})
$$

$$
\leq \frac{2\lambda R_{\max}}{(1-\gamma)^2} \mathbb{E}_{s \sim d_{\widehat{\mathcal{M}}}^{\lambda\hat{\pi}_\beta+(1-\lambda)\pi^*}} \left[ d_{\mathrm{TV}}(\pi^*, \hat{\pi}_\beta)(s) \right]
$$

$$
+ \frac{2\alpha(1-\lambda)}{1-\gamma} \mathbb{E}_{s \sim d_{\widehat{\mathcal{M}}}^{\lambda\hat{\pi}_\beta+(1-\lambda)\pi^*}(s)} \left[ d_{\mathrm{TV}}(\pi^*, \hat{\pi})(s) \left( \xi(\hat{\pi})(s) + \frac{\gamma}{1-\gamma} \mathbb{E}_{a \sim \hat{\pi}(\cdot|s)} \left[ \frac{\hat{\pi}(a|s)}{\hat{\pi}_\beta(a|s)} - 1 \right] \right) \right]
$$

$$
+ \frac{C_{r,\mathcal{P}}^\delta}{1-\gamma} \mathbb{E}_{s \sim d_{\widehat{\mathcal{M}}}^{\lambda\hat{\pi}_\beta+(1-\lambda)\hat{\pi}}(s)} \left[ \frac{\sqrt{|\mathcal{A}|}}{\sqrt{N(s)}} \left( \lambda + (1-\lambda) \sqrt{\mathbb{E}_{a \sim \hat{\pi}(\cdot|s)} \left[ \frac{\hat{\pi}(a|s)}{\hat{\pi}_\beta(a|s)} \right]} \right) \right]
$$

$$
+ \frac{C_{r,\mathcal{P}}^\delta}{1-\gamma} \mathbb{E}_{s \sim d_{\widehat{\mathcal{M}}}^{\pi^*}(s)} \left[ \frac{\sqrt{|\mathcal{A}|}}{\sqrt{N(s)}} \sqrt{\mathbb{E}_{a \sim \pi^*(\cdot|s)} \left[ \frac{\pi^*(a|s)}{\hat{\pi}_\beta(a|s)} \right]} \right],
$$

This completes the proof of Theorem 3. ∎

## C  EXPERIMENTAL DETAILS AND PARAMETER SETUP

In this appendix, we first briefly introduce how normalized scores are calculated in the D4RL benchmark. We then describe our implementation and experimental details.

### C.1  D4RL BENCHMARKS

D4RL provides a metric, the normalized score, which represents a normalized undiscounted average return, to evaluate the performance of offline RL algorithms. It is calculated as follows:

$$\text{Normalized score} = \frac{\text{average return - return of the random policy}}{\text{return of the expert policy - return of the random policy}} \times 100.$$

Note that 0 represents the performance of a random policy, and 100 represents the performance of an expert policy. In D4RL, if the task is in the same environment, different types of datasets share the same reference minimum and maximum scores. We summarize the reference score for each environment in Table 2. For AntMaze, we set the number of episodes to 100 and evaluate the number of times the goal is reached. If the ant successfully reaches the goal location, it is rewarded with 1.0, indicating a successful episode. Conversely, if the ant fails to reach the goal, it receives a reward of 0.0, reflecting an unsuccessful attempt.

Table 2: The reference minimum and maximum scores for MuJoCo, Adroit, and AntMaze datasets.

| Domain | Task | Reference Min Score | Reference Max Score |
|---|---|---|---|
| MuJoCo | Halfcheetah | -280.18 | 12135.0 |
| MuJoCo | Hopper | -20.27 | 3234.3 |
| MuJoCo | Walker2d | 1.63 | 4592.3 |
| Adroit | Pen | 96.26 | 3076.83 |
| Adroit | Door | -56.51 | 2880.57 |
| Adroit | Hammer | -274.86 | 12794.13 |
| Adroit | Relocate | -6.43 | 4233.88 |
| AntMaze | Umaze / Medium / Large | 0.0 | 1.0 |

### C.2  BASELINES

#### C.2.1  OFFLINE BASELINES

To generate the results reported in Table 1, we conduct experiments on MuJoCo "-v2", Adroit "-v0", and Antmaze "-v0" datasets. We adopt behavior cloning (BC), several canonical offline RL algorithms (TD3+BC (Fujimoto & Gu, 2021), CQL (Kumar et al., 2020), and IQL (Kostrikov et al., 2022)), and more recent extensions of CQL (MCQ (Lyu et al., 2022), MISA (Ma et al., 2023), CSVE (Chen et al., 2023), and EPQ (Yeom et al., 2024)). For a fair comparison, we evaluate all algorithms using results after 1M gradient steps. Thus, certain algorithms must be reproduced for all datasets, while for some datasets, several algorithms with missing values must also be reproduced.

**MuJoCo Locomotion Tasks.** We take the results for TD3+BC (Table 9 in Fujimoto & Gu (2021)), MCQ (Table 1 in Lyu et al. (2022)), and CSVE (Table 1 in Chen et al. (2023)) as reported in their original papers. Since the reported scores in the CQL paper are based on "-v0" datasets, and the scores for BC are needed, we take the scores for BC and CQL from Table 1 in Lyu et al. (2022). Since the IQL and MISA papers do not report performance on the Random and Expert datasets, we take the results for IQL from Table 1 in Lyu et al. (2022) and for MISA from Table 1 in Yeom et al. (2024). For the Medium, Medium-Replay, and Medium-Expert datasets, we directly take the results of IQL (Table 1 in Kostrikov et al. (2022)) and MISA (Table 2 in Ma et al. (2023)) from their original papers. Since the EPQ paper reports scores after 3M gradient steps, we run the official implementation of EPQ on all datasets for 1M gradient steps, available at https://github.com/hyeon1996/EPQ.

**Adroit Manipulation Tasks.** We take the results for CQL (Table 2 in Kumar et al. (2020)), IQL (Table 1 in Kostrikov et al. (2022)), MCQ (Table 9 in Lyu et al. (2022)), MISA (Table 2 in Ma et al.

(2023)), and CSVE (Table 2 in Chen et al. (2023)) as reported in their original papers. Since the scores for BC are needed, we take the scores for BC from Table 2 in Kumar et al. (2020). Since the TD3+BC paper do not report performance on Adroit tasks, we take the results for TD3+BC from Table 1 in Yeom et al. (2024). Since the EPQ paper reports scores after 0.3M gradient steps, we run the official implementation of EPQ on all Adroit datasets for 1M gradient steps, available at https://github.com/hyeon1996/EPQ.

**AntMaze Navigation Tasks.** We take the results for TD3+BC (Table 8 in Fujimoto & Gu (2021)), CQL (Table 2 in Kumar et al. (2020)), IQL (Table 1 in Kostrikov et al. (2022)), and MISA (Table 2 in Ma et al. (2023)) as reported in their original papers. Since the scores for BC are needed, we take the scores for BC from Table 2 in Kumar et al. (2020). Since the MCQ paper does not report performance on AntMaze tasks, we take the results for MCQ from Table 1 in Yeom et al. (2024). Although a repository (https://github.com/2023AnnonymousAuthor/csve) appears to be the code for the paper, it does not provide parameters for the AntMaze dataset, preventing us from conducting experiments. Since the EPQ paper reports scores after 3M gradient steps, we run the official implementation of EPQ on all AntMaze datasets for 1M gradient steps, available at https://github.com/hyeon1996/EPQ.

### C.2.2 OFFLINE-TO-ONLINE BASELINES

To generate the performance curve reported in Figure 3, we conduct experiments on MuJoCo "-v2" datasets. We adopt canonical offline-to-online RL algorithms (AWAC (Nair et al., 2020) and Cal-QL (Nakamoto et al., 2023)), offline RL algorithms that achieve high performance in online RL (IQL (Kostrikov et al., 2022) and SPOT (Wu et al., 2022)), and CQL (Kumar et al., 2020) (offline) to SAC (online). For a fair comparison, we evaluate all algorithms using results after 0.25M gradient steps for offline settings and 0.3M gradient steps for online settings. We run the implementations of the five algorithms based on the CORL (Tarasov et al., 2024b) GitHub repository, available at https://github.com/tinkoff-ai/CORL.

### C.3 CPQL IMPLEMENTATION DETAILS

Table 3: Hyperparameter setup for CPQL

|  | Hyperparameter | Value |
| --- | --- | --- |
| SAC hyperparameters | Optimizer | Adam (Kingma, 2014) |
|  | Critic learning rate | 3e-4 |
|  | Actor learning rate | 1e-4 |
|  | Batch size | 256 |
|  | Discount factor | 0.99 / MuJoCo and AntMaze |
|  |  | 0.90 / Adroit |
|  | Target update rate | 5e-3 |
|  | Target entropy | $-1 \cdot$ Action Dimension |
|  | Entropy in Q-target | False |
| Architecture | Critic hidden dim | 256 |
|  | Critic hidden layers | 3 / MuJoCo and Adroit |
|  |  | 5 / AntMaze |
|  | Critic activation function | ReLU |
|  | Actor hidden dim | 256 |
|  | Actor hidden layers | 3 |
|  | Actor activation function | ReLU |
| CPQL hyperparameters | Lagrange | True / AntMaze |
|  |  | False / MuJoCo and Adroit |
|  | conservatism parameter | $\{0.1, 0.5, 1.0, 3.0, 5.0, 7.0, 10.0\}$ |
|  | Lagrange gap | 0.8 / AntMaze |
|  | Pre-training steps | 0 |
|  | Num sampled actions (during eval) | 10 |
|  | Num sampled actions (logsumexp) | 10 |
|  | Trajectory Length | 5 |
|  | $\lambda$ | $\{0.0, 0.1, 0.3, 0.5, 0.7, 0.9, 0.95, 0.99\}$ |

We set the trajectory length $n = 5$ for CPQL to cap the length of the partial trajectories. Across all of our experiments, we tune the conservatism parameter $\alpha$ and $\lambda$ from the following potential values using grid search: $\alpha \in \{0.1, 0.5, 1, 3, 5, 7, 10\}$ and $\lambda \in \{0, 0.1, 0.3, 0.5, 0.7, 0.9, 0.95, 0.99\}$. In offline-to-online RL, we set the conservatism parameter $\alpha$ to either 1 or 5. We extend our experiments to include $\alpha$ values lower than the previously typical choices of 5 and 10 used in CQL. We optimize the learned policy following the standard SAC (Haarnoja et al., 2018) approach. We run the CPQL implementation based on the CORL (Tarasov et al., 2024b) GitHub repository, available at https://github.com/tinkoff-ai/CORL. The hyperparameter setup for CPQL, including the default SAC configuration, is detailed in Table 3. We summarize the hyperparameters used for running the MuJoCo, Adroit, and AntMaze tasks in Table 4. We plot the performance of CPQL in Figure 5 using the best parameters from Table 4.

Table 4: Detailed hyperparameters of CPQL, where we conduct experiments on MuJoCo-Gym ("v2") and Adroit and AntMaze ("v0") datasets.

| Task | conservatism parameter $\alpha$ | PQL parameter $\lambda$ |
|---|---|---|
| halfcheetah-random | 0.1 | 0.3 |
| halfcheetah-medium | 0.1 | 0.0 |
| halfcheetah-medium-replay | 0.1 | 0.3 |
| halfcheetah-medium-expert | 10.0 | 0.1 |
| halfcheetah-expert | 3.0 | 0.0 |
| hopper-random | 0.1 | 0.0 |
| hopper-medium | 0.1 | 0.7 |
| hopper-medium-replay | 0.5 | 0.1 |
| hopper-medium-expert | 5.0 | 0.1 |
| hopper-expert | 10.0 | 0.9 |
| walker2d-random | 0.5 | 0.9 |
| walker2d-medium | 1.0 | 0.5 |
| walker2d-medium-replay | 1.0 | 0.7 |
| walker2d-medium-expert | 1.0 | 0.95 |
| walker2d-expert | 1.0 | 0.99 |
| pen-human | 10.0 | 0.5 |
| door-human | 5.0 | 0.7 |
| hammer-human | 7.0 | 0.9 |
| relocate-human | 1.0 | 0.9 |
| pen-cloned | 1.0 | 0.5 |
| door-cloned | 3.0 | 0.1 |
| hammer-cloned | 5.0 | 0.7 |
| relocate-cloned | 10.0 | 0.1 |
| antmaze-umaze | 7.0 | 0.1 |
| antmaze-diverse | 5.0 | 0.9 |
| antmaze-medium-play | 10.0 | 0.3 |
| antmaze-medium-diverse | 5.0 | 0.1 |
| antmaze-large-play | 10.0 | 0.1 |
| antmaze-large-diverse | 5.0 | 0.0 |

Analysis of the Halfcheetah-expert-v2 dataset suggests that the underlying behavior policy is not truly near-optimal (the normalized score of the trajectories is approximately 85, compared to approximately 100 for Hopper and Walker2d). Thus, a large $\lambda$ may degrade performance.

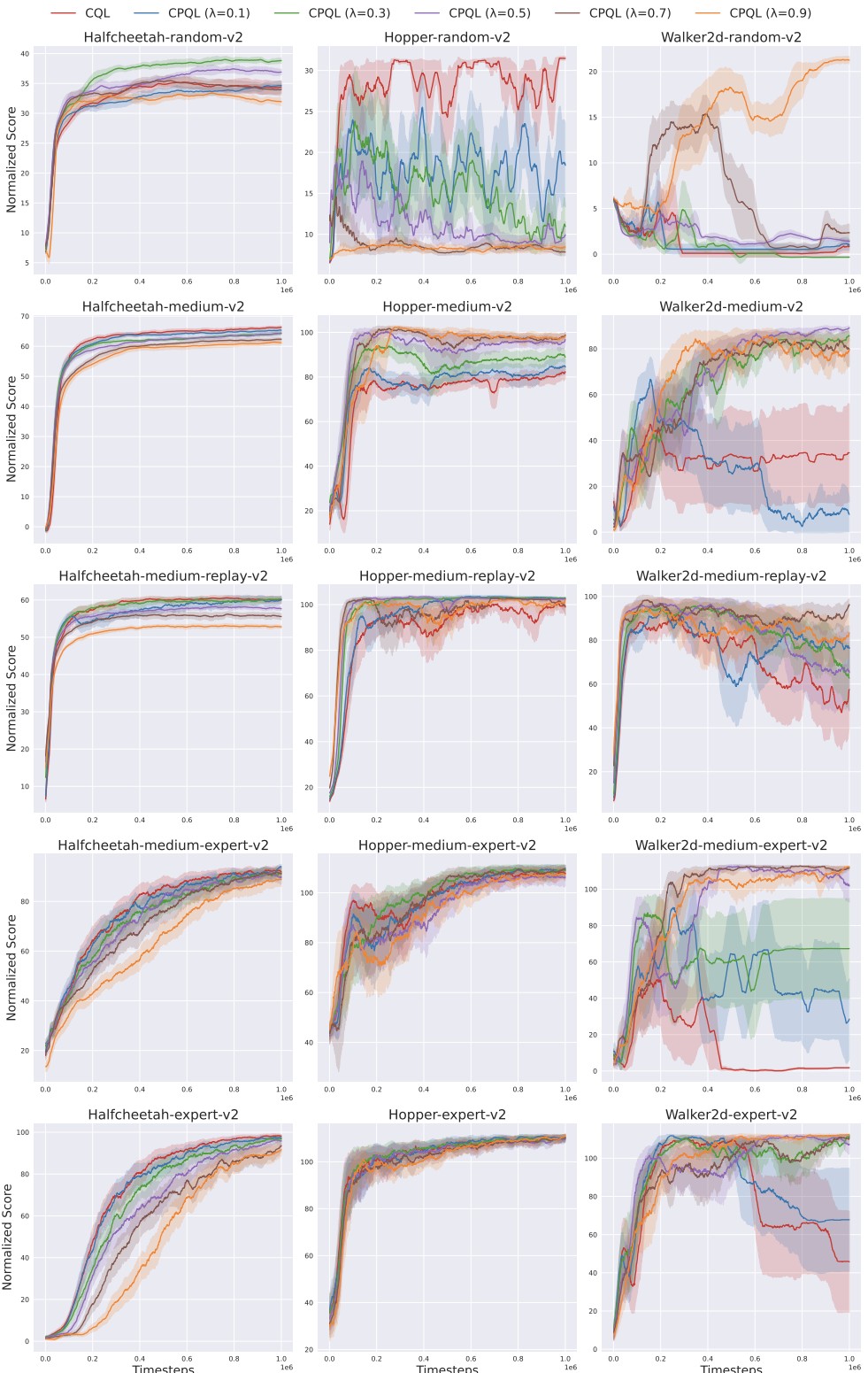

Figure 5: Performance of CPQL in MuJoCo locomotion tasks.

## C.4 RUNNING TIME

We compare the computational cost of CQL and CPQL, which use a single-step operator and a multi-step operator, respectively. We run this comparison based on the *Hopper-Medium-v2* dataset with a single GeForce RTX 3090 GPU. We measure the average runtime per epoch (1K training steps) except for the evaluation step. The results are reported in Table 5. We observe that CQL and CPQL have average runtimes of 40.6 and 42.4 seconds, respectively. The runtime difference between the two algorithms is minimal, but as shown in Table 1, we observe significant performance differences.

Table 5: Computational costs of CQL and CPQL.

| Epoch runtime (s) | CQL | CPQL |
|---|---|---|
| 1,000 gradient steps | 40.6 | 42.4 |

Compared to two recent conservative value estimation algorithms, MCQ (Lyu et al., 2022) and EPQ (Yeom et al., 2024), CPQL not only outperforms on diverse tasks but also has a lower runtime. According to Table 3 in Yeom et al. (2024), the reported runtimes using a single NVIDIA RTX A5000 GPU are as follows: CQL (43.1 seconds), MCQ (58.1 seconds), and EPQ (54.8 seconds). For a fair comparison, we compute the ratio that indicates how much the training time increases in Table 6. We confirm that CPQL is the most efficient compared to other algorithms.

Table 6: Epoch runtime-increase relative to CQL.

| Ratio of epoch runtime (%) | CPQL | MCQ | EPQ |
|---|---|---|---|
| Epoch time growth | 4.4 | 34.8 | 27.1 |

Additionally, MCQ and EPQ require more training time because they rely on autoencoder-based OOD action estimation (Lyu et al., 2022) and additional penalty adaptation factors (Yeom et al., 2024), respectively. Therefore, CPQL achieves superior performance with significantly lower computational cost, outperforming MCQ and EPQ while requiring less training time by avoiding autoencoder-based OOD action estimation and additional penalty adaptation factors.

Additionally, because the runtime per 1K gradient steps differs by approximately a factor of **3.2** between IQL and CQL/CPQL, we compare their sample efficiency under a comparable wall-clock budget. We therefore report the normalized scores after **1M** gradient steps for IQL, **0.32M** for CQL, and **0.3M** for CPQL. The normalized scores of IQL are taken from (Kostrikov et al., 2022).

Table 7: Normalized scores under a comparable wall-clock budget.

| Task | IQL (1M grad steps) | CQL (0.32M grad steps) | CPQL (0.3M grad steps) |
|---|---|---|---|
| halfcheetah-random | 13.1 | $25.1 \pm 1.6$ | $\mathbf{37.5 \pm 1.2}$ |
| hopper-random | 7.9 | $8.2 \pm 1.1$ | $\mathbf{20.2 \pm 0.2}$ |
| walker2d-random | 5.4 | $2.5 \pm 5.5$ | $\mathbf{13.4 \pm 8.4}$ |
| halfcheetah-medium | 47.4 | $48.4 \pm 0.3$ | $\mathbf{64.4 \pm 1.0}$ |
| hopper-medium | 66.2 | $62.6 \pm 6.1$ | $\mathbf{102.0 \pm 1.6}$ |
| walker2d-medium | 78.3 | $\mathbf{81.8 \pm 1.9}$ | $82.0 \pm 2.4$ |
| halfcheetah-medium-replay | 44.2 | $46.6 \pm 0.3$ | $\mathbf{59.1 \pm 1.7}$ |
| hopper-medium-replay | 94.7 | $99.6 \pm 1.9$ | $\mathbf{103.4 \pm 0.6}$ |
| walker2d-medium-replay | 73.8 | $84.7 \pm 4.5$ | $\mathbf{95.9 \pm 3.2}$ |
| halfcheetah-medium-expert | $\mathbf{86.7}$ | $70.3 \pm 9.2$ | $76.2 \pm 10.9$ |
| hopper-medium-expert | $\mathbf{91.5}$ | $88.0 \pm 22.0$ | $\mathbf{90.4 \pm 23.4}$ |
| walker2d-medium-expert | $\mathbf{109.6}$ | $\mathbf{110.4 \pm 0.2}$ | $110.0 \pm 0.7$ |

CPQL achieves substantially better performance than CQL and IQL when normalized by the number of gradient steps, further strengthening the practical claim of our algorithm.

# D COMPARISON WITH OTHER MULTI-STEP OPERATORS

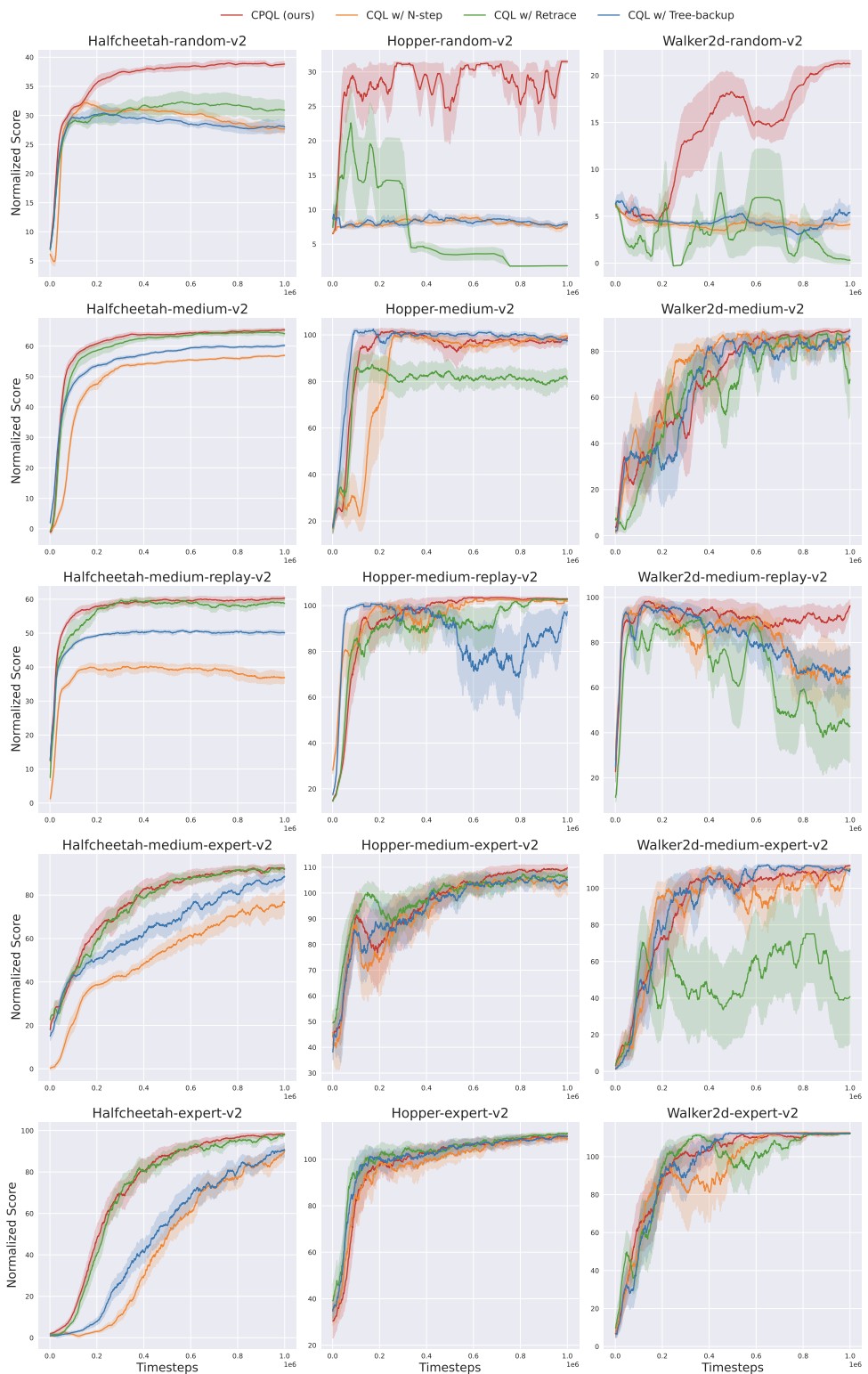

Figure 6: Comparison of CPQL (ours) with CQL using alternative multi-step operators (Uncorrected N-step, Retrace, and Tree-backup) on MuJoCo locomotion tasks from D4RL.

Table 8: Computational costs of CQL (baseline), CPQL (ours), Uncorrected N-step, Retrace, and Tree-backup on Hopper-medium-v2. We report the runtime per 1,000 gradient steps (in seconds). For Retrace, the additional cost $+ \alpha$ accounts for the extra time required to estimate the behavior policy, typically using behavior cloning. CPQL's computational cost is comparable to the single-step operator, with only a marginal increase in runtime.

| Epoch runtime (s) | CQL (baseline) | CPQL (ours) | CQL w/ N-step | CQL w/ Retrace | CQL w/ Tree-backup |
|---|---|---|---|---|---|
| 1,000 gradient steps | 40.6 | 42.4 | 41.3 | $43.0 + \alpha$ | 43.0 |

## E  COMPARISON WITH PQL

In this section, we address the following question:

*How does CPQL compare to a method that purely uses PQL, without the conservatism penalty?*

In the main text, we focused primarily on evaluating performance across the D4RL benchmarks (MuJoCo, AntMaze, and Adroit) in both offline and offline-to-online settings. We interpret the results to clarify why the PQL operator is useful in offline RL and why CPQL is needed. To this end, we evaluate CPQL and PQL on MuJoCo locomotion tasks, using the normalized return and the critic model's average Q-values over a batch size of samples as evaluation metrics.

Table 9: Normalized Return (Real Performance) of CPQL and PQL.

| Task | Algorithm | $\lambda = 0.3$ | $\lambda = 0.7$ |
|---|---|---|---|
| hopper-medium-replay | CPQL ($\alpha = 1$) | $102.6 \pm 0.8$ | $102.5 \pm 0.7$ |
| | PQL | $24.9 \pm 10.8$ | $45.3 \pm 27.9$ |
| walker2d-medium | CPQL ($\alpha = 1$) | $85.1 \pm 5.5$ | $79.4 \pm 18.6$ |
| | PQL | $-0.2 \pm 0.0$ | $-0.2 \pm 0.0$ |

Table 10: Average Q-values (Estimated Values) of CPQL and PQL.

| Task | Algorithm | $\lambda = 0.3$ | $\lambda = 0.7$ |
|---|---|---|---|
| hopper-medium-replay | CPQL ($\alpha = 1$) | $235.8 \pm 5.6$ | $222.2 \pm 4.9$ |
| | PQL | $314.7 \pm 3.2$ | $271.5 \pm 4.0$ |
| walker2d-medium | CPQL ($\alpha = 1$) | $335.4 \pm 7.9$ | $332.6 \pm 7.7$ |
| | PQL | $4 \times 10^{11} \pm 10^{10}$ | $475.8 \pm 6.1$ |

We observe several notable findings in Table 9 and 10. Simply applying PQL to the offline dataset substantially mitigates one of the most important challenges in offline RL, the overestimation of Q-values caused by distribution shift. For instance, in the *hopper-medium-replay* dataset, SAC reports the normalized score of only around 3.5 (from Table 1 in CQL paper), indicating a failure to learn the optimal policy, whereas PQL achieves significantly higher performance. Nevertheless, the distribution shift induced by the learned policy persisted, underscoring the necessity of CPQL to address this limitation more effectively. In the *walker2d-medium* dataset, PQL with $\lambda = 0.7$ reduced average Q-value overestimation compared to $\lambda = 0.3$, yet this reduction did not translate into an improved normalized return.

In contrast, CPQL combines conservative value estimation with the PQL operator, suppressing overestimation while incorporating long-horizon information. As a result, in both *hopper-medium-replay* and *walker2d-medium* datasets, CPQL achieves much more stable and higher returns than PQL, demonstrating that the synergistic integration of conservatism and the multi-step operator plays a critical role in improving offline RL performance.

## F    CUSTOMIZED OFFLINE DATASETS

From the D4RL datasets, it is difficult to determine the exact behavior policy, which makes it challenging to precisely measure the role of $\lambda$. To address this issue, we constructed customized offline datasets in the *Halfcheetah* and *Walker2d* environments. Using SAC, we collected 200K samples with the policy obtained at the point where the normalized score reached 20. We continued training until the normalized score reached 100, designating this policy as the optimal policy. Based on these setups, we conducted several ablation studies to better understand the effects of CPQL and $\lambda$.

### F.1    COMPARISON OF CQL, PQL, AND CPQL

Table 11: Normalized Return (Real Performance) and Average Q-values (Estimated Values) for the customized dataset of *Walker2d*.

| Walker2d | Behavior Policy ($\pi_\beta$) | Optimal Policy ($\pi^*$) | CQL | - |
|---|---|---|---|---|
| Normalized Return | 20 | 100 | $45.9 \pm 9.5$ | - |
| Average Q-values | $\approx 192.74$ | $\approx 267.76$ | $75.9 \pm 5.4$ | - |

| Walker2d | PQL ($\lambda = 0.3$) | PQL ($\lambda = 0.7$) | CPQL ($\lambda = 0.3$) | CPQL ($\lambda = 0.7$) |
|---|---|---|---|---|
| Normalized Return | $-0.5 \pm 0.0$ | $0.1 \pm 0.1$ | $63.5 \pm 8.9$ | $\mathbf{81.3 \pm 4.5}$ |
| Average Q-values | $4 \times 10^{10} \pm 2.3 \times 10^9$ | $438.7 \pm 20.1$ | $129.6 \pm 5.9$ | $174.3 \pm 8.2$ |

In Table 11, we set the conservatism parameter $\alpha$ to 5.0 for both CQL and CPQL. Comparing CQL and PQL, CQL produces relatively low average Q-values due to the conservatism term, achieving a performance of around 45.9. In contrast, PQL with $\lambda = 0.3$ suffers from the typical overestimation problem in offline RL, but as $\lambda$ increased to 0.7, its average Q-value decreased to around 438.7. It shows the migration of the over-conservatism effect with the PQL operator. However, PQL still failed to learn the optimal policy, because the learned policy still suffers from a large distribution shift, leading to high Q-values.

By adding the conservatism term to PQL, CPQL alleviates this issue and outperforms CQL in terms of performance. This improvement occurs because, under the same conservatism parameter, CPQL has mildly conservative Q-values. This aligns with the theoretical insights in Theorems 1-3. Furthermore, we observe that as $\lambda$ was increased, PQL's average Q-values approached those of the behavior policy, whereas CPQL's average Q-values approached those of the optimal policy.

### F.2    COMPARISON OF CPQL AND OTHER MULTI-STEP OPERATORS

Multi-step operators without a conservatism term are expected to fail to learn a policy that approaches the optimal policy. Thus, we add the conservatism term with $\alpha = 1.0$ for all algorithms.

Table 12: Normalized Return (Real Performance) and Average Q-values (Estimated Values) for the customized dataset of *Halfcheetah*.

| Halfcheetah | CPQL | CQL w/ Nstep | CQL w/ Retrace | CQL w/ Tree-backup |
|---|---|---|---|---|
| Normalized Return | $\mathbf{39.6 \pm 2.6}$ | $31.0 \pm 2.8$ | $\mathbf{39.5 \pm 2.8}$ | $34.4 \pm 3.1$ |
| Average Q-values | $213.7 \pm 10.1$ | $127.6 \pm 5.2$ | $212.4 \pm 10.5$ | $130.5 \pm 7.6$ |

In Table 12, CPQL and CQL with Retrace achieved the highest performance (39.6 and 39.5), maintaining relatively high average Q-values, 213, which indicates a milder conservatism. In contrast, CQL with $n$-step returns and Tree-backup showed lower returns (31.0 and 34.4) and substantially lower average Q-values, suggesting stronger conservatism. In this case, the $n$-step return prevents the agent from exploring OOD actions. Tree-backup, on the other hand, was developed for discrete action spaces, and in continuous spaces it leads to very unstable updates due to the numerical scale of $\ln \pi$.

In the above case of an offline dataset collected from a single policy, as in the previous experiments, estimating the behavior policy is relatively straightforward. This explains why Retrace achieved performance comparable to CPQL. However, an open question is whether Retrace would still perform well when the offline dataset is generated by multiple behavior policies. To investigate this, we collected four datasets in *Walker2d* with normalized scores of 20, 60, and 100, containing 200K, 120K, and 80K samples (ratio 5 : 3 : 2), resulting in a total of 400K samples for training. We add the conservatism term with $\alpha = 5.0$ for all algorithms.

Table 13: Normalized Return for a toy example of the mixture dataset of *Walker2d*.

| Walker2d | CPQL | CQL w/ Retrace |
|---|---|---|
| Normalized Return | **98.6 ± 3.5** | 87.8 ± 22.8 |

CPQL outperforms CQL with Retrace, indicating that CPQL has more robust performance for datasets collected from multiple behavior policies. This trend is consistent with the results observed on the D4RL *random* and *medium-replay* datasets.

# G  ADDITIONAL BASELINES

## G.1  OFFLINE RL

We evaluate our method on MuJoCo locomotion tasks in offline settings, comparing it against Q-value uncertainty approaches for conservative estimation (ensemble-based: EDAC (An et al., 2021), PBRL (Bai et al., 2021); non-ensemble: UWAC (Wu et al., 2021), QDQ (Zhang et al., 2024)) as well as trajectory-based methods (DT (Chen et al., 2021), TT (Janner et al., 2021)). We also compare against additional single-step baselines in effectively regulating OOD actions ($\mathcal{X}$-QL (Garg et al., 2023), SQL, EQL (Xu et al., 2023), and InAC (Xiao et al., 2023)).

Table 14: Results for MuJoCo locomotion tasks. * indicates methods trained with 3M gradient steps as reported in original papers. All other methods are trained with 1M gradient steps. Bold numbers are the scores within 2% of the highest in each environment.

| Task | EDAC* | PBRL | UWAC | QDQ | DT | TT | CPQL (ours) |
|---|---|---|---|---|---|---|---|
| halfcheetah-random | 28.4 | 11.0 | 2.3 | - | - | - | **38.8 ± 1.0** |
| hopper-random | 25.3 | 26.8 | 2.7 | - | - | - | **31.5 ± 0.5** |
| walker2d-random | 16.6 | 8.1 | 2.0 | - | - | - | **21.2 ± 0.7** |
| halfcheetah-medium-v2 | 65.9 | 58.2 | 42.2 | **74.1** | 42.6 | 46.9 | 66.6 ± 0.9 |
| hopper-medium-v2 | **101.6** | 81.6 | 50.9 | 99.0 | 67.6 | 61.1 | **99.7 ± 2.0** |
| walker2d-medium-v2 | 92.5 | **90.3** | 75.4 | 86.9 | 74.0 | 79.0 | **90.0 ± 1.5** |
| halfcheetah-medium-replay-v2 | 61.3 | 49.5 | 35.9 | **63.7** | 36.6 | 41.9 | 60.3 ± 0.8 |
| hopper-medium-replay-v2 | **101.0** | 100.7 | 25.3 | **102.4** | 82.7 | 91.5 | **103.0 ± 0.6** |
| walker2d-medium-replay-v2 | 87.1 | 86.2 | 23.6 | 93.2 | 66.6 | 82.6 | **97.4 ± 4.0** |
| halfcheetah-medium-expert-v2 | **106.3** | 93.6 | 42.7 | 99.3 | 86.8 | 95.0 | 95.3 ± 0.6 |
| hopper-medium-expert-v2 | 110.7 | **111.2** | 44.9 | **113.5** | 107.6 | 110.0 | **111.3 ± 1.2** |
| walker2d-medium-expert-v2 | **114.7** | 109.8 | 96.5 | **115.9** | 108.1 | 101.9 | 112.9 ± 2.0 |
| halfcheetah-expert-v2 | **106.8** | 96.2 | 92.9 | - | - | - | 98.0 ± 1.6 |
| hopper-expert-v2 | **110.1** | 110.4 | **110.5** | - | - | - | **112.0 ± 0.6** |
| walker2d-expert-v2 | **115.1** | 108.8 | 108.4 | - | - | - | 114.1 ± 0.5 |

Across MuJoCo locomotion tasks, in Table 14, CPQL consistently achieves competitive or superior performance compared to both Q-value uncertainty methods (with and without ensembles) and trajectory-based approaches. In particular, it matches or exceeds the strongest baselines in *medium*, *medium-replay*, and *expert* datasets, demonstrating robustness across varying data qualities. Furthermore, in Table 15, when compared to recent single-step baselines designed to regulate OOD actions, CPQL achieves the highest or near-highest scores across all benchmark settings. These results confirm that CPQL is not only effective in addressing conservatism but also reliable in balancing exploration and value estimation, leading to strong and stable returns across diverse offline RL tasks.

Table 15: Results for MuJoCo locomotion tasks. Bold numbers are the scores within 2% of the highest in each environment.

| Task | $\mathcal{X}$-QL | SQL | EQL | InAC | CPQL (ours) |
|---|---|---|---|---|---|
| halfcheetah-medium | 48.3 | 48.3 | 47.2 | 48.3 | **66.6 ± 0.9** |
| hopper-medium | 74.2 | 75.5 | 74.6 | 60.3 | **99.7 ± 2.0** |
| walker2d-medium | 84.2 | 84.2 | 83.2 | 82.7 | **90.0 ± 1.5** |
| halfcheetah-medium-replay | 45.2 | 44.8 | 44.5 | 44.3 | **60.3 ± 0.8** |
| hopper-medium-replay | 100.7 | 99.7 | 98.1 | 92.1 | **103.0 ± 0.6** |
| walker2d-medium-replay | 82.2 | 81.2 | 76.6 | 69.8 | **97.4 ± 4.0** |
| halfcheetah-medium-expert | **94.2** | 94.0 | 90.6 | 83.5 | **95.3 ± 0.6** |
| hopper-medium-expert | 111.2 | 111.8 | 105.5 | 93.8 | **111.3 ± 1.2** |
| walker2d-medium-expert | 112.7 | 110.0 | 110.2 | 109.0 | **112.9 ± 2.0** |
| halfcheetah-expert | - | - | - | 93.6 | **98.0 ± 1.6** |
| hopper-expert | - | - | - | 103.4 | **112.0 ± 0.6** |
| walker2d-expert | - | - | - | 110.6 | **114.1 ± 0.5** |

## G.2 OFFLINE-TO-ONLINE RL

We evaluate our method on the MuJoCo locomotion tasks and AntMaze navigation tasks after fine-tuning with 300k online samples. We report the final normalized score average over five random seeds, with ± indicating the 95%-confidence interval.

- MuJoCo locomotion tasks: We compare CPQL (offline) to PQL (online) against several algorithms: (i) CQL (offline) to SAC (online), (ii) PEX (Zhang et al., 2023) that expands the policy set during online fine-tuning using optimistic exploration (iii) RLPD (Ball et al., 2023) that regularizes online updates using value and policy constraints from offline data, and (iv) Cal-QL that calibrates the value-function. (see Table 16)

- AntMaze navigation tasks: These environments are known to be extremely challenging for standard off-policy RL algorithms like SAC, due to their sparse rewards and complex exploration requirements. As vanilla online algorithm fails to learn successful policies in these tasks, we compare CPQL against several algorithms, including CQL. (see Table 17)

Table 16: Results for MuJoCo locomotion tasks in offline-to-online settings. Bold numbers are the scores within 2% of the highest in each environment.

| MuJoCo | CQL→SAC | PEX | RLPD | Cal-QL | CPQL→PQL |
|---|---|---|---|---|---|
| halfcheetah-random | 90.3 ± 3.1 | 60.9 ± 6.2 | 91.5 ± 3.1 | 32.9 ± 10.1 | **93.8 ± 6.3** |
| hopper-random | 33.7 ± 34.9 | 48.5 ± 48.3 | 90.2 ± 23.7 | 17.7 ± 32.3 | **102.0 ± 1.7** |
| walker2d-random | 3.8 ± 7.9 | 9.8 ± 2.0 | 87.7 ± 17.5 | 9.4 ± 7.0 | **88.6 ± 20.1** |
| halfcheetah-medium | 96.3 ± 1.6 | 70.4 ± 2.9 | 95.5 ± 1.9 | 77.0 ± 2.7 | **96.5 ± 1.7** |
| hopper-medium | 109.3 ± 1.1 | 86.2 ± 32.7 | 91.4 ± 34.5 | 100.7 ± 1.0 | **111.5 ± 0.7** |
| walker2d-medium | 114.4 ± 3.2 | 91.4 ± 17.8 | 121.6 ± 2.9 | 97.0 ± 10.2 | **127.8 ± 3.4** |
| halfcheetah-medium-replay | 94.8 ± 1.9 | 55.4 ± 6.3 | 90.1 ± 1.6 | 62.1 ± 1.4 | **95.8 ± 2.2** |
| hopper-medium-replay | 108.4 ± 3.4 | 95.3 ± 8.9 | 78.9 ± 30.4 | 101.4 ± 2.6 | **112.1 ± 2.5** |
| walker2d-medium-replay | 114.7 ± 11.1 | 87.2 ± 16.9 | 119.0 ± 2.6 | 98.4 ± 4.1 | **128.6 ± 4.8** |

From the results presented in the table above, CPQL to PQL method achieves significantly better performance compared to other baselines. Several factors contribute to this advantage. First, the Q-function learned by PQL does not degrade at the beginning of the online phase. This is because CPQL reduces the influence of the learned policy on Q-value estimation, resulting in more stable value learning. Second, compared to PEX, RLPD, and Cal-QL, PQL benefits from a stronger exploration capability, as it is guided by a well-trained Q-function obtained from CPQL.

Table 17: Results for AntMaze tasks in offline-to-online settings. Bold numbers are the scores within 2% of the highest in each environment.

| AntMaze | CQL | PEX | RLPD | Cal-QL | CPQL |
|---|---|---|---|---|---|
| antmaze-umaze | **99.0 ± 0.7** | 95.2 ± 2.0 | **99.4 ± 1.0** | 90.1 ± 13.4 | **98.2 ± 1.0** |
| antmaze-umaze-diverse | 76.9 ± 49.3 | 34.8 ± 37.4 | **99.2 ± 1.2** | 75.2 ± 43.5 | 90.4 ± 3.1 |
| antmaze-medium-play | 94.4 ± 3.7 | 83.4 ± 2.9 | **97.4 ± 1.7** | 95.1 ± 7.8 | 93.4 ± 1.9 |
| antmaze-medium-diverse | **98.8 ± 3.1** | 86.6 ± 6.2 | **98.6 ± 1.7** | 96.3 ± 6.0 | **98.2 ± 1.8** |
| antmaze-large-play | 87.3 ± 7.0 | 56.0 ± 4.8 | **93.0 ± 3.1** | 75.0 ± 18.2 | 85.4 ± 6.3 |
| antmaze-large-diverse | 65.3 ± 35.1 | 60.4 ± 8.4 | **90.4 ± 4.8** | 74.4 ± 14.6 | 82.0 ± 5.6 |

In the AntMaze tasks, CPQL outperforms (or equal to) other baselines except for RLPD, with only a slight performance gap compared to RLPD. The advantage of CPQL becomes even more pronounced when compared to CQL. Taken together, the results from both the MuJoCo and AntMaze tasks demonstrate that our algorithm is more robust and delivers superior overall performance.

# H    ADDITIONAL RELATED WORKS

**Model-based Offline RL.** Model-based offline RL methods build dynamics and reward models from the offline dataset, leveraging state transitions and rewards of estimated model outputs for planning and policy improvements. They typically achieve this by penalizing the reward function with the error between the ground truth and estimated models (Yu et al., 2020; Kidambi et al., 2020; Rafailov et al., 2021; Lu et al., 2021; Kim & Oh, 2023; Sun et al., 2023), learning conservative Q-function within the model-based regime  (Yu et al., 2021), and training the policy and the dynamics model adversarially (Rigter et al., 2022). Algorithms that learn by planning synthetic trajectories under estimated dynamics typically perform policy evaluation using a single-step approach. However, applying CPQL in model-based offline RL settings can be particularly beneficial, similar to COMBO (Yu et al., 2021). It enables more conservative learning of the Q-function, mitigating overestimation issues. This suggests that CPQL has broad applicability and can enhance various aspects of model-based reinforcement learning.

Recently, Park & Lee (2025) considered a model-based offline RL approach, computing the target Q-function by applying lower expectile regression to $\lambda$-returns on synthetic trajectories planned from the estimated dynamics. This method differs from ours: we take a model-free offline RL approach, leveraging offline trajectories collected from the *actual* environment. Our method effectively enhances performance by utilizing real trajectories rather than relying on synthetic trajectories, which are subject to model estimation uncertainty. Furthermore, while they additionally employ lower expectile regression to obtain a conservative return estimate, CPQL derives a conservative value estimate solely by integrating the multi-step operator with a conservative estimation mechanism.

Kun et al. (2024) proposed Uni-O4, an offline policy evaluation method for safe multi-step policy improvement based on an approximate model and fitted Q evaluation. However, CPQL differs conceptually from Uni-O4. First, CPQL applies the PQL operator directly to offline trajectories from the real environment, whereas Uni-O4 computes its multi-step objective by rolling out an approximate transition model $\hat{T}$ trained on the offline dataset and aggregating fitted-Q estimates along simulated trajectories (AM-Q). Second, CPQL uses these multi-step returns in the critic update to obtain conservative value estimates for control, while Uni-O4 uses AM-Q only as an offline policy evaluation oracle that gates PPO-style policy updates. Finally, Uni-O4 trains an ensemble behavior-cloning policy to stay close to the data-support region and stabilize AM-Q, whereas CPQL does not require learning the behavior policy or a transition model.

**Offline Trajectory.** Several works (Yue et al., 2022; Liu et al., 2024; Xu et al., 2024) have attempted to handle offline trajectories in different ways to adaptively utilize information from past observations, where rewards have already been realized. They propose several methods, such as return-based data rebalancing in Yue et al. (2022), priority assignment based on trajectory quality using average, minimum, maximum, and quantile rewards in Liu et al. (2024), as well as a least-squares-based reward redistribution method for reward estimation in Xu et al. (2024). However, these methods are not applicable in sparse reward settings, such as AntMaze tasks, and were not empirically tested in such environments. In contrast, we show that CPQL achieves superior performance in sparse reward settings.

# I   ABLATION OF TD LOSS AND Q-VALUES

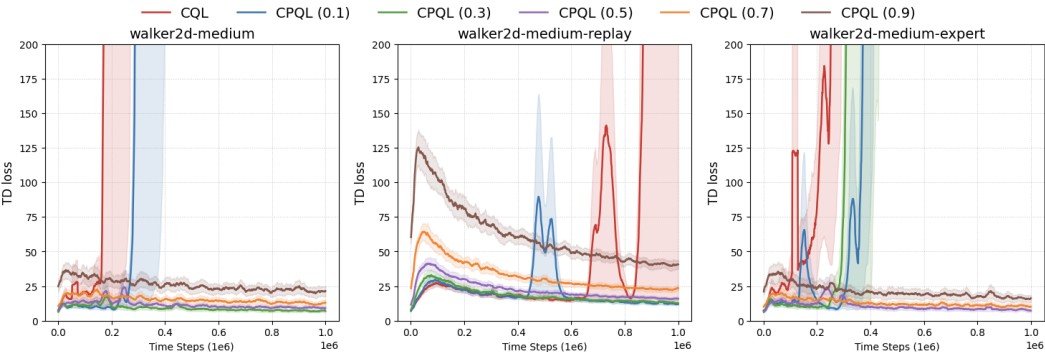

Figure 7:   TD loss of different $\lambda$ values

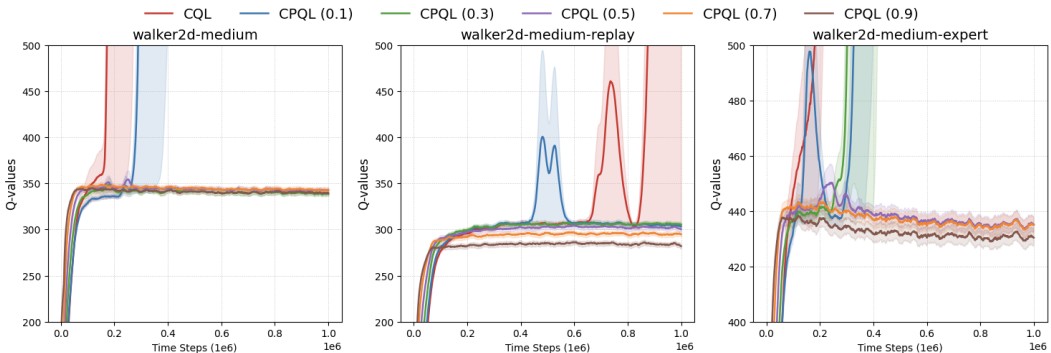

Figure 8:   Q-values of different $\lambda$ values

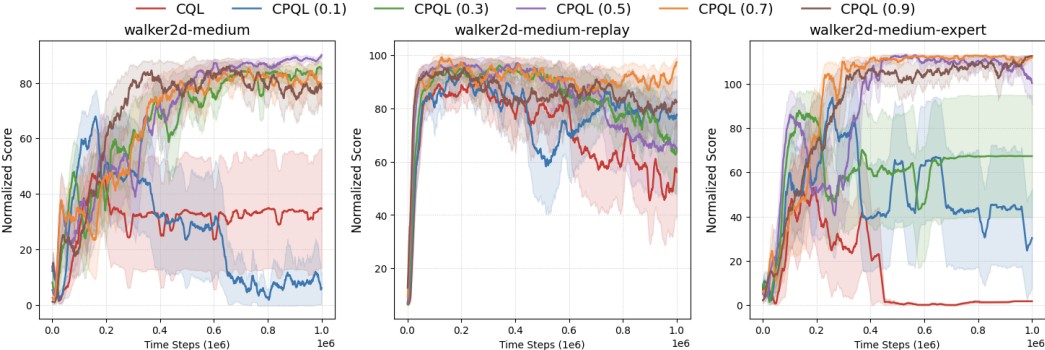

Figure 9:   Normalized scores of different $\lambda$ values

## J   ABLATION OF TRAJECTORY LENGTH

Table 18:   Comparison of CPQL with different trajectory lengths on MuJoCo locomotion tasks.
**Same** $(\alpha, \lambda)$ means that CPQL with $n = 10$ uses the same hyperparameters $(\alpha, \lambda)$ as $n = 5$, while
**Best** uses the best $(\alpha, \lambda)$ tuned for $n = 10$.

| Task | CPQL $n = 5$ | CPQL $n = 10$ | | Remarks |
|---|---|---|---|---|
| | | Same $(\alpha, \lambda)$ | Best $(\alpha, \lambda)$ | |
| halfcheetah-random | $38.8 \pm 1.0$ | $37.4 \pm 0.9$ | $37.4 \pm 0.9$ | - |
| hopper-random | $31.5 \pm 0.5$ | $31.5 \pm 0.5$ | $31.5 \pm 0.5$ | - |
| walker2d-random | $21.2 \pm 0.7$ | $21.3 \pm 0.5$ | $21.4 \pm 0.5$ | $(0.5, 0.9) \to (0.5, 0.95)$ |
| halfcheetah-medium | $66.6 \pm 0.9$ | $66.6 \pm 0.9$ | $66.6 \pm 0.9$ | - |
| hopper-medium | $99.7 \pm 2.0$ | $100.0 \pm 1.5$ | $100.0 \pm 1.5$ | $(0.1, 0.7) \to (0.1, 0.9)$ |
| walker2d-medium | $90.0 \pm 1.5$ | $89.4 \pm 1.3$ | $89.4 \pm 1.3$ | - |
| halfcheetah-medium-replay | $60.3 \pm 0.8$ | $60.5 \pm 0.7$ | $60.5 \pm 0.7$ | - |
| hopper-medium-replay | $103.0 \pm 0.6$ | $102.1 \pm 2.1$ | $103.2 \pm 0.8$ | $(0.5, 0.1) \to (0.5, 0.3)$ |
| walker2d-medium-replay | $97.4 \pm 4.0$ | $95.7 \pm 4.4$ | $95.7 \pm 4.4$ | - |
| halfcheetah-medium-expert | $95.3 \pm 0.6$ | $94.2 \pm 0.7$ | $95.4 \pm 0.6$ | $(10.0, 0.1) \to (10.0, 0.3)$ |
| hopper-medium-expert | $111.3 \pm 1.2$ | $106.7 \pm 6.0$ | $110.8 \pm 2.4$ | $(5.0, 0.1) \to (3.0, 0.7)$ |
| walker2d-medium-expert | $112.9 \pm 0.5$ | $112.8 \pm 0.4$ | $112.8 \pm 0.4$ | $(1.0, 0.95) \to (1.0, 0.99)$ |
| halfcheetah-expert | $98.0 \pm 1.6$ | $98.0 \pm 1.6$ | $98.7 \pm 1.0$ | $(3, 0.0) \to (3, 0.3)$ |
| hopper-expert | $112.0 \pm 0.6$ | $111.9 \pm 0.6$ | $111.9 \pm 0.6$ | - |
| walker2d-expert | $114.1 \pm 0.4$ | $114.3 \pm 0.4$ | $114.3 \pm 0.4$ | - |

## K   ABLATION OF CONSERVATISM PARAMETER

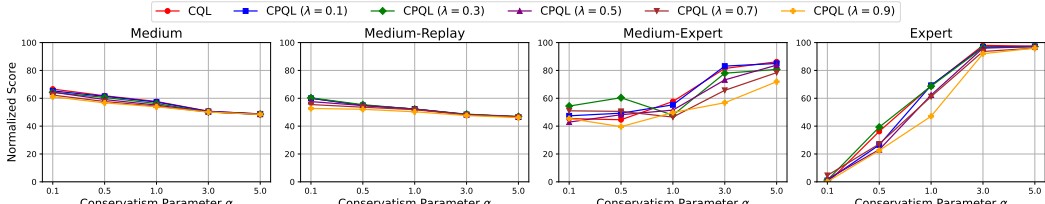

Figure 10:  Normalized scores for different conservatism parameters $\alpha$ in *Halfcheetah* tasks.

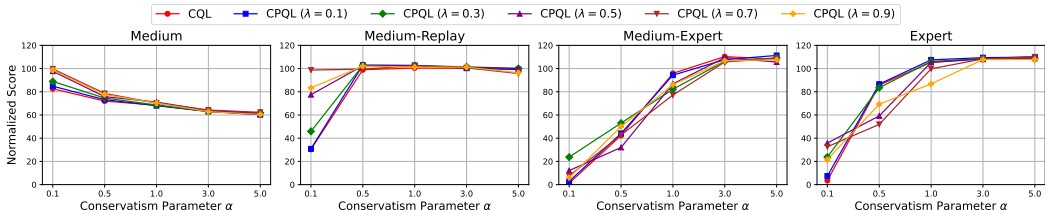

Figure 11:  Normalized scores for different conservatism parameters $\alpha$ in *Hopper* tasks.

