# OpenReview forum: "Peng's Q($\lambda$) for Conservative Value Estimation in Offline Reinforcement Learning"
_ICLR.cc/2026/Conference — ICLR 2026 Poster_

### Official Review · Reviewer_qHcg · 2025-10-28

**Soundness:** 3
**Presentation:** 3
**Contribution:** 3
**Rating:** 6
**Confidence:** 4

**Summary:**

The paper presents Conservative Peng’s $Q(\lambda)$ (CPQL), a model-free offline multi-step reinforcement learning algorithm that utilizes the Peng’s $Q(\lambda)$ operator for conservative value estimation. CPQL achieves superior performance on standard benchmarks.

**Strengths:**

1. The experimental evaluation is comprehensive.
2. The algorithm demonstrates superior performance across the majority of tasks.

**Weaknesses:**

1. A primary concern is that the claim of "multi-step" does not intuitively address the over-optimistic value estimation challenge in offline reinforcement learning. If the multi-step approach were fundamentally effective, it raises the question of why conservative value estimation, which is widely used in offline RL, is still necessary.

2. The authors assert that their method mitigates over-pessimistic Q-values, a challenge already addressed in other approaches such as MCQ. A comparison of the Q-value curves between CPQL and MCQ would help clarify the extent of this improvement.

3. Can CPQL be viewed as a result of simply modifying the target value $y$ in CQL? Would it be possible to perform a similar substitution in MCQ to further demonstrate the effectiveness of the "multi-step" approach? What impact would this have on the results?

4. While the authors conduct offline-to-online experiments, they do not compare their method against recent state-of-the-art algorithms, such as WSRL, which has demonstrated superior performance in the AntMaze domain.

**Questions:**

Please see weakness

---

> ### Author Response · Authors · 2025-11-21
> **Official Comment by Authors (1/2)**
>
> We appreciate your positive evaluation of our work and the constructive feedback! We will address your questions below and make improvements to our paper based on your suggestions.
>
> ---
>
> ## **PQL operator (multi-step) helps, but conservatism is still essential**
>
> **Our view is that the PQL operator and conservative regularization play complementary roles.**
> We agree that the PQL operator does not fully resolve the over-optimistic value estimation problem in offline RL.
> The PQL operator primarily mitigates over-optimism within the data support, while conservatism is required to control extrapolation on OOD actions.
> We illustrate this with additional experiments.
>
>
> We first construct customized offline datasets in the Walker2d environment.
> Using SAC, we collect $200$K samples with the policy obtained at the point where the normalized score reaches $20$ (the behavior policy).
> We then continue training SAC until the normalized score reaches $100$ and designate this policy as the optimal policy.
>
> |Toy example for Walker2d|Behavior policy|Optimal policy|SAC|PQL ($\lambda=0.3$)|PQL ($\lambda=0.7$)|
> |-|-|-|-|-|-|
> |Normalized Return|20|100|-0.5 $\pm$ 0.0|-0.5 $\pm$ 0.0|0.1 $\pm$ 0.1|
> |Average Q-values|$\approx$ 192.74|$\approx$ 267.76|1.6 $\times$ $10^{11}$ $\pm$ 5.2 $\times$ $10^{10}$|4.2 $\times$ $10^{10}$ $\pm$ 2.3 $\times$ $10^{9}$|438.7 $\pm$ 20.1|
>
> In the above table, SAC, PQL ($\lambda=0.3$), and PQL ($\lambda=0.7$) achieve low normalized returns on this dataset.
> However, their average Q-values differ significantly.
> **SAC shows severe overestimation on the order of $10^{11}$, whereas increasing $\lambda$ in the PQL operator increases the bias toward the behavior policy and leads to much more stable Q-values.**
> This shows that the PQL operator does mitigate over-optimistic value estimation within the support of the offline data.
>
> However, this mitigation is not sufficient to solve the overestimation challenge in offline RL by itself.
> Since the learned policy, weighted by $1-\lambda$, still queries OOD actions during policy evaluation, this can lead to an accumulation of extrapolation errors.
> Thus, it remains necessary to control the contribution of OOD actions in the Q-values.
> This is why we use a conservative value estimation method.
>
> To see how PQL and conservative value estimation interact, we next compare CQL and CPQL on the same toy Walker2d setup:
> |Toy example for Walker2d|CQL|CPQL ($\lambda=0.3$)|CPQL ($\lambda=0.7$)|
> |-|-|-|-|
> |Normalized Return|45.9 $\pm$ 9.5|63.5 $\pm$ 8.9|81.3 $\pm$ 4.5|
> |Average Q-values|75.9 $\pm$ 5.4|129.6 $\pm$ 5.9|174.3 $\pm$ 8.2|
>
> We set the conservatism parameter $\alpha$ to $5.0$ for both CQL and CPQL.
> By adding the conservatism term to PQL, CPQL alleviates the overly conservative Q-values observed in CQL and outperforms CQL in terms of performance.
> This improvement occurs because, **under the same conservatism parameter, CPQL has mildly conservative Q-values.**
> This aligns with the theoretical insights in Theorems 1-3.
>
> In summary, these results address the reviewer’s concern:
>
> - The PQL operator does substantially **reduce over-optimistic value estimates and stabilizes Q-values without estimating the behavior policy and requireing additional models.**
> - Conservative value estimation is still needed to handle OOD actions, and when combined with PQL in CPQL, **it avoids the over-conservatism of CQL while maintaining robustness in the offline RL setting.**
>
> ---
>
> ## **Comparison of the Q-value curves between CPQL and MCQ**
>
> We report the average Q-values of CPQL and MCQ [1] during training (1M gradient steps) on three *medium-replay* tasks in **Figure 12**, and the corresponding normalized scores in **Figure 13 from Appendix L**.
> Across all three datasets, CPQL consistently converges to higher Q-values (mildly conservative value estimation) than MCQ while maintaining equal or better normalized returns.
> For example, in *halfcheetah-medium-replay*, Q-values of MCQ plateau at a substantially lower level than those of CPQL.
> This gap is mirrored in the final performance ($\approx$ 54 vs. $\approx$ 60 normalized scores).
> A similar pattern appears in *hopper-medium-replay* and *walker2d-medium-replay*.
> These results support our claim that **CPQL mitigates over-pessimistic Q-values beyond what MCQ already addresses, without requiring behavior policy estimation.**

---

> > ### Author Response · Authors · 2025-11-21
> > **Official Comment by Authors (2/2)**
> >
> > ---
> >
> > ## **MCQ with the PQL operator**
> >
> > **The PQL operator demonstrates that a multi-step approach can also be effective within MCQ.**
> > To illustrate this, we adapt the PQL operator to MCQ and run experiments with three seeds on the *medium* and *medium-replay* datasets in the MuJoCo tasks.
> > In MCQ, the hyperparameter $\lambda_{\text{MCQ}}$ controls how the loss is weighted between in-distribution and OOD actions when computing the target values: $\lambda_{\text{MCQ}}$ is applied to the target values for in-distribution actions, and $1 - \lambda_{\text{MCQ}}$ is applied to the target values for OOD actions from the estimated behavior policy.
> > We conduct a grid search over $\lambda_{\text{MCQ}} \in \\{0.7, 0.8, 0.9\\}$, choosing values around the official $\lambda_{\text{MCQ}}$ reported in Table 5 of the MCQ paper (approximately within $\pm 0.1$).
> > We set the trajectory length to $5$ and perform a grid search over $\lambda_{\text{PQL}} \in \\{0.3, 0.7\\}$.
> > The normalized scores in the MCQ column are taken from the original MCQ paper.
> >
> > - m: medium / m-r: medium-replay
> >
> > |Task|MCQ|MCQ w/ PQL|
> > |-|:-:|:-:|
> > |Halfcheetah-m|**64.3 $\pm$ 0.2**|**64.5 $\pm$ 0.7**|
> > |Hopper-m|78.4 $\pm$ 4.3|**82.2 $\pm$ 15.8**|
> > |Walker2d-m|**91.0 $\pm$ 0.4**|**89.8 $\pm$ 7.8**|
> > |Halfcheetah-m-r|**56.8 $\pm$ 0.6**|**56.5 $\pm$ 0.7**|
> > |Hopper-m-r|**101.6 $\pm$ 0.8**|**101.7 $\pm$ 1.4**|
> > |Walker2d-m-r|91.3 $\pm$ 0.7|**99.5 $\pm$ 0.8**|
> >
> > MCQ with the PQL operator remains competitive and sometimes outperforms the original MCQ, while still recovering the original method when $\lambda = 0$.
> > Thus, these results suggest that the PQL operator can safely complement MCQ’s mildly conservative design.
> >
> >
> >
> > ---
> >
> > ## **Comparison to WSRL in offline-to-online experiments**
> >
> > In the WSRL [2] paper, WSRL is initialized from Cal-QL and then fine-tuned with an online SAC-style learner that uses an ensemble of $10$ Q-networks and a high update-to-data (UTD) ratio of $4$.
> > This fine-tuning phase includes a warm-up stage that seeds the replay buffer with $5,000$ rollouts from the pre-trained policy, without retaining the offline dataset.
> >
> > In our experiments, we set the UTD to $1$ to enable a fair comparison between WSRL and the other algorithms.
> > We evaluate our method and state-of-the-art baselines (e.g., WSRL) on the AntMaze navigation tasks by first pre-training them for $0.25$M offline steps and then fine-tuning them for $0.3$M online steps.
> >
> > - u: umaze / u-d: umaze-diverse / m-p: medium-play / m-d: medium-diverse / l-p: large-play / l-d: large-diverse
> >
> > |AntMaze|CQL|PEX|RLPD|Cal-QL|WSRL|CPQL|
> > |-|-|-|-|-|-|-|
> > |antmaze-u|**99.0 $\pm$ 0.7**|95.2 $\pm$ 2.0|**99.4 $\pm$ 1.0**|90.1 $\pm$ 13.4|**99.2 $\pm$ 1.0**|**98.2 $\pm$ 1.0**|
> > |antmaze-u-d|76.9 $\pm$ 49.3|34.8 $\pm$ 37.4|**99.2 $\pm$ 1.2**|75.2 $\pm$ 43.5|**98.4 $\pm$ 1.4**|90.4 $\pm$ 3.1|
> > |antmaze-m-p|94.4 $\pm$ 3.7|83.4 $\pm$ 2.9|**97.4 $\pm$ 1.7**|95.1 $\pm$ 7.8|**98.5 $\pm$ 1.3**|93.4 $\pm$ 1.9|
> > |antmaze-m-d|**98.8 $\pm$ 3.1**|86.6 $\pm$ 6.2|**98.6 $\pm$ 1.7**|96.3 $\pm$ 6.0|97.0 $\pm$ 1.4|**98.2 $\pm$ 1.8**|
> > |antmaze-l-p|87.3 $\pm$ 7.0|56.0 $\pm$ 4.8|**93.0 $\pm$ 3.1**|75.0 $\pm$ 18.2|**91.6 $\pm$ 2.8**|85.4 $\pm$ 6.3|
> > |antmaze-l-d|65.3 $\pm$ 35.1|60.4 $\pm$ 8.4|**90.4 $\pm$ 4.8**|74.4 $\pm$ 14.6|**92.0 $\pm$ 3.6**|82.0 $\pm$ 5.6|
> >
> > The key components of WSRL—such as the critic ensemble, higher UTD ratio, and warm-up phase—are orthogonal to our contribution and could also be applied to CPQL.
> > **Given that CPQL already performs well in the offline setting, we expect that combining CPQL with WSRL techniques would further improve its offline-to-online performance.**
> > Although WSRL is a general fine-tuning framework that can be paired with various offline RL algorithms rather than a direct competitor to CPQL, the above table shows that **CPQL achieves performance comparable to WSRL on the AntMaze tasks.**
> >
> > ---
> >
> > **Reference**
> >
> > [1] Lyu, Jiafei, et al. "Mildly conservative q-learning for offline reinforcement learning." Advances in Neural Information Processing Systems 35 (2022): 1711-1724.
> >
> > [2] Zhou, Zhiyuan, et al. "Efficient Online Reinforcement Learning Fine-Tuning Need Not Retain Offline Data." The Thirteenth International Conference on Learning Representations.

---

### Official Review · Reviewer_MQiE · 2025-10-30

**Soundness:** 3
**Presentation:** 2
**Contribution:** 1
**Rating:** 4
**Confidence:** 4

**Summary:**

CPQL (Conservative Peng's $\mathrm{Q}(\lambda)$ ) integrates the conservative Q penalty structure of Conservative Q-Learning (CQL) with the multi-step backup of Peng's $\mathrm{Q}(\lambda)$. To address the limitation of standard CQL, which suffers from restricted reward propagation leading to over-pessimism, CPQL introduces $\lambda$ to leverage trajectory-level information for long-term reward propagation. This allows the algorithm to stably utilize multi-step information without importance sampling while implicitly capturing the characteristics of the behavior policy in its Q-values. While empirical results demonstrates that CPQL outperforms existing baselines on the D4RL benchmark, the authors further provide theoretical evidence to prove that CPQL guarantees that the learned policy always performs at least as well as the behavior policy.

**Strengths:**

$\textbf{(1)}$

According to the authors, Conservative Peng's $Q(\lambda)$ (CPQL) represents the first attempt to incorporate multi-step temporal difference learning into the offline reinforcement learning setting. The method combines Peng's $\mathrm{Q}(\lambda)$ operator with a conservative Q-penalty, aiming to utilize multi-step information through the hyperparameter $\lambda$, while maintaining the stability of conservative value estimation through the hyperparameter $\alpha$. Unlike many prior approaches that rely on additional auxiliary networks-such as behavior policy estimators, importance-sampling correction models, or ensemble-based uncertainty estimators, CPQL performs this integration without additional components. This design allows the algorithm to propagate reward signals more effectively across trajectories, potentially capturing richer temporal dependencies without introducing additional instability.

$\textbf{(2)}$

CPQL extends the mixture policy convergence property of Peng's $Q(\lambda)$ to the offline reinforcement learning setting, providing a theoretical analysis of how multi-step learning interacts with conservative value estimation. The study examines both the role of the hyperparameter $\alpha$, which controls the degree of conservatism, and the parameter $\lambda$, which determines how strongly the mixture policy reflects the characteristics of the behavior policy. Specifically, while Theorem 2 proves that the policy learned by CPQL guarantees performance that is at least equal to or better than that of the behavior policy, Theorem 3 shows that, with an appropriately chosen $\lambda$, the sub-optimality gap can be reduced by effectively balancing conservatism and multi-step backup. Overall, these theoretical results strengthen the justification for CPQL's design choices and suggest that it provides an interpretable and mathematically grounded framework for policy improvement.


$\textbf{(3)}$

While the conservative Q-penalty in CQL plays an important role in suppressing overestimation of out-of-distribution actions, it often leads to an overly pessimistic value function, which can limit policy improvement. To mitigate this issue, CPQL introduces a multi-step temporal difference update, allowing reward signals to be propagated over longer horizons within offline trajectories. By leveraging trajectory-level information, the algorithm captures a more accurate relationship between actions and long-term returns, alleviating the problem of over-pessimism without compromising the stability provided by conservative regularization. Such balance between conservatism and optimism contributes to stable performance not only during purely offline training, but also in off-to-on fine-tuning, where the learned policy can adapt to online interaction without suffering from the sharp performance drop often observed in overly conservative methods.

**Weaknesses:**

$\textbf{(1)}$

The authors suggest that this work may represent an early attempt to extend Conservative Q-Learning into an N-step formulation, where multi-step temporal difference updates are incorporated within a conservative value estimation framework. However, the proposed method primarily integrates an existing algorithm, Peng's $Q(\lambda)$, into the established CQL framework, and its core idea appears to be more of an adaptation or refinement of prior methodologies rather than a fundamentally new algorithmic contribution. Moreover, much of the theoretical basis related to PQL has already been extensively discussed in previous studies, and the theoretical analyses of CPQL closely follow the analytical structure developed for CQL. Consequently, while the paper provides an interesting perspective on reinterpreting conservative value learning through a multi-step lens, its methodological originality appears to be relatively limited compared to earlier works.

$\textbf{(2)}$

According to the authors, the interaction between $\lambda$, which controls the degree to which the learned mixture policy follows the behavior policy, and $\alpha$, which determines the level of conservatism, is designed to regulate the balance between stability and policy improvement. Theoretically, a higher-quality behavior policy is expected to benefit from a larger $\lambda$, encouraging the learned policy to stay closer to the behavior policy, while a lower quality or inconsistent policy would require a smaller $\lambda$ to prevent the accumulation of erroneous transitions. However, the experimental results in domains such as HalfCheetah do not appear to consistently reflect this theoretically suggested relationship. Moreover, although Figure 1 claims that the presence of $\lambda$ makes $\alpha$ less sensitive in CPQL compared to CQL, it is not entirely convincing from the presented results that such reduced sensitivity actually holds in practice.

$\textbf{(3)}$

While CPQL conceptually allows for richer reward propagation across trajectories, its empirical validation has so far been limited to relatively simple or short-horizon tasks within the D4RL benchmark. As a result, it remains uncertain whether the proposed method can generalize its effectiveness to more complex offline RL environments that involve longer temporal dependencies or higher behavioral variability. For instance, evaluations on more diverse and realistic benchmarks-such as non-goal conditioned settings in frameworks like OGBENCH: BENCHMARKING OFFLINE GOAL-CONDITIONED RL[1]-could provide further insight into whether CPQL consistently maintains a stable balance between conservatism and policy performance across a wider range of task dynamics.

(reference)
[1] Park, Seohong, et al. "Ogbench: Benchmarking offline goal-conditioned rl." International Conference on Learning Representations (2025)

**Questions:**

$\textbf{(1)}$

Given that CPQL is a multi-step method for offline reinforcement learning, it seems reasonable to consider the trajectory length $n$ as an important hyperparameter in the implementation.
In the current paper, $n$ is fixed to 5 for all experiments, but I believe that both the performance and the behavior of the hyperparameters $\alpha$ and $\lambda$ could vary significantly depending on the choice of $n$.
Could the authors clarify the motivation for fixing $n=5$ in all experiments? Additionally, have the authors conducted (or considered) any experiments analyzing how performance or the interaction between $\alpha$ and $\lambda$ changes as $n$ varies?


$\textbf{(2)}$

In Figure 1, the experimental results on the sensitivity of the conservatism parameter $\alpha$ suggest that, for the same $\alpha$ values, CPQL tends to outperform CQL.
Moreover, it appears that as $\lambda$ increases, the sensitivity of CPQL to $\alpha$ becomes smaller, indicating that the method may be more robust to the choice of $\alpha$ under higher $\lambda$ values.
I am curious whether similar patterns were observed consistently across other tasks or environments.

Could the authors provide additional results or discussions that show whether this trend holds beyond the task presented in Figure 1?

---

> ### Author Response · Authors · 2025-11-21
> **Official Comment by Authors (1/3)**
>
> We appreciate your time to review our paper and your feedback. We would like to clarify some of the concerns regarding our theoretical and empirical results.
>
> ---
>
> ## **Novelty of CPQL**
>
> **CPQL goes beyond a simple integration of existing techniques and offers a conceptually different way to address overestimation in offline RL.**
> A key insight of CPQL is that the fixed point of the PQL operator lies closer to the value function of the behavior policy, thereby **inducing implicit behavior regularization.**
> Compared to other multi-step operators, the PQL operator yields a biased fixed point toward the behavior policy, which helps mitigate Q-value overestimation in offline RL.
> In practice, the PQL operator does not rely on accurate behavior-policy estimation (unlike Retrace), can be applied in continuous action spaces (unlike Tree-backup), and does not overly restrict the exploration of OOD actions (unlike the $n$-step return), making it well-suited for offline RL.
> In contrast to prior conservative value estimation methods that tackle Q-value overestimation by adding extra components, such as explicit behavior-policy estimation (MCQ, EPQ) or additional networks (CSVE), CPQL takes a different approach.
> We show that this behavior-regularization effect can be achieved by reinterpreting the PQL operator on offline trajectories, without requiring additional behavior-policy estimation, auxiliary networks, or extensive hyperparameter tuning compared to prior conservative value estimation algorithms.
>
>
> **Theorem 3 demonstrates that CPQL can effectively shrink the sub-optimality gap, compared to prior conservative value estimation methods such as CQL, MCQ, CSVE, and EPQ.**
> In these works, the main guarantee is that the policy learned by these algorithms achieves performance that is greater than (or equal to) that of the behavior policy, which is analogous to Theorem 2 in our paper.
> **However, none of these conservative value estimation algorithms explicitly shows that the sub-optimality gap itself can be reduced.**
> Our theoretical guarantees not only extend CQL’s analytical structure to the multi-step operator but also quantify the sub-optimality gap and clarify how CPQL can reduce it in a mathematically interpretable way.
>
>
> **Extensive numerical experiments on the D4RL benchmarks, together with OGBench results added during the rebuttal period, demonstrate that CPQL significantly outperforms existing offline single-step RL algorithms.**
> On the OGBench benchmark suite (antmaze, cube, scene, and puzzle), CPQL also achieves strong performance, consistently maintaining a balance between conservatism and policy performance across a wide range of task dynamics.
> Beyond mitigating overly pessimistic value estimation in offline RL, CPQL also contributes to offline-to-online learning.
> Initializing the online PQL agent with a Q-function pre-trained by CPQL helps avoid the performance drop typically observed at the start of the online phase and yields robust improvements in performance.
>
> ---
>
> ## **OGBench (Non-goal conditioned setting)**
>
> To demonstrate that CPQL maintains a consistent balance between conservatism and policy performance across a wider range of task dynamics, we evaluate it on the recently proposed OGBench tasks [1].
>
> We evaluate CPQL on five tasks: *antmaze-medium*, *antmaze-large*, *cube-single*, *scene-single*, and *puzzle-3 $\times$ 3-single*.
> All results are averaged over three random seeds.
> BC and IQL results are taken from [2], except for IQL on antmaze-medium, which we reproduced.
>
> |Task Category|BC|IQL|CPQL|
> |-|:-:|:-:|:-:|
> |OGBench antmaze-medium-singletask (5 tasks)|-|**82 $\pm$ 10**|**84 $\pm$ 11**|
> |OGBench antmaze-large-singletask (5 tasks)|11 $\pm$ 1|**53 $\pm$ 3**|**54 $\pm$ 6**|
> |OGBench cube-single-singletask (5 tasks)|5 $\pm$ 1|83 $\pm$ 3|**98 $\pm$ 2**|
> |OGBench scene-singletask (5 tasks)|5 $\pm$ 1|**28 $\pm$ 1**|**26 $\pm$ 1**|
> |OGBench puzzle-3 $\times$ 3-singletask (5 tasks)|2 $\pm$ 0|9 $\pm$ 1|**21 $\pm$ 2**|
>
> **CPQL achieves strong performance on the OGBench task suite.**

---

> > ### Author Response · Authors · 2025-11-21
> > **Official Comment by Authors (2/3)**
> >
> > ---
> >
> > ## **Change the trajectory length $n$**
> >
> > We evaluate CPQL with the trajectory length $n=10$ in the D4RL MuJoCo benchmark, which comprises a total of $15$ datasets from $3$ different environments (*HalfCheetah*, *Hopper*, and *Walker2d*), each with $5$ dataset types (*Random*, *Medium*, *Medium-Replay*, *Medium-Expert*, and *Expert*).
> >
> > - $n=5$: Original scores reported in the main table.
> > - $n=10$ / Same ($\alpha$, $\lambda$): Only increase the trajectory length from $5$ to $10$ while keeping ($\alpha$, $\lambda$) fixed to the values used for $n = 5$.
> > - $n=10$ / Best ($\alpha$, $\lambda$): Increase the trajectory length to 10 and then select the best-performing pair ($\alpha$, $\lambda$) from our hyperparameter search for each task.
> >
> >
> > |Task|$n=5$|$n=10$/ Same ($\alpha$, $\lambda$)|$n = 10$/ Best ($\alpha$, $\lambda$)|
> > |:-|:-:|:-:|:-:|
> > |  halfcheetah-r  | 38.8 $\pm$ 1.0  |   37.4 $\pm$ 0.9    |   37.4 $\pm$ 0.9    |
> > |    hopper-r     | 31.5 $\pm$ 0.5  |   31.5 $\pm$ 0.5    |   31.5 $\pm$ 0.5    |
> > |   walker2d-r    | 21.2 $\pm$ 0.7  |   21.3 $\pm$ 0.5    |   21.4 $\pm$ 0.5    |
> > |  halfcheetah-m  | 66.6 $\pm$ 0.9  |   66.6 $\pm$ 0.9    |   66.6 $\pm$ 0.9    |
> > |    hopper-m     | 99.7 $\pm$ 2.0  |   100.0 $\pm$ 1.5   |   100.0 $\pm$ 1.5   |
> > |   walker2d-m    | 90.0 $\pm$ 1.5  |   89.4 $\pm$ 1.3    |   89.4 $\pm$ 1.3    |
> > | halfcheetah-m-r | 60.3 $\pm$ 0.8  |   60.5 $\pm$ 0.7    |   60.5 $\pm$ 0.7    |
> > |   hopper-m-r    | 103.0 $\pm$ 0.6 |   102.1 $\pm$ 2.1   |   103.2 $\pm$ 0.8   |
> > |  walker2d-m-r   | 97.4 $\pm$ 4.0  |   95.7 $\pm$ 4.4    |   95.7 $\pm$ 4.4    |
> > | halfcheetah-m-e | 95.3 $\pm$ 0.6  |   94.2 $\pm$ 0.7    |   95.4 $\pm$ 0.6    |
> > |   hopper-m-e    | 111.3 $\pm$ 1.2 |   106.7 $\pm$ 6.0   |   110.8 $\pm$ 2.4   |
> > |  walker2d-m-e   | 112.9 $\pm$ 0.5 |   112.8 $\pm$ 0.4   |   112.8 $\pm$ 0.4   |
> > |  halfcheetah-e  | 98.0 $\pm$ 1.6  |   98.0 $\pm$ 1.6    |   98.7 $\pm$ 1.0    |
> > |    hopper-e     | 112.0 $\pm$ 0.6 |   111.9 $\pm$ 0.6   |   111.9 $\pm$ 0.6   |
> > |   walker2d-e    | 114.1 $\pm$ 0.4 |   114.3 $\pm$ 0.4   |   114.3 $\pm$ 0.4   |
> > - r: random / m: medium / m-r: medium-replay / m-e: medium-expert / e: expert
> >
> > We observe that **the choice of $(\alpha, \lambda)$ may not be sensitive to $n$ ($\ge 5$).**
> > Comparing trajectory lengths $5$ and $10$, the first to fourth transitions receive the same weights, namely $(1-\lambda)$, $\lambda(1-\lambda)$, $\lambda^{2}(1-\lambda)$, and $\lambda^{3}(1-\lambda)$.
> > For $n = 5$, the fifth step then receives the remaining weight $\lambda^{4}$, whereas for $n = 10$, the fifth step receives $\lambda^{4}(1-\lambda)$, while the remaining weights $\lambda^{5}(1-\lambda), \ldots, \lambda^{8}(1-\lambda), \lambda^{9}$ are assigned to later steps.
> > When $\lambda$ is relatively large, later transitions receive non-negligible weight, so using a longer trajectory length can further exploit information from the tail of the trajectory.
> > In such cases, one may find a different near-optimal choice of $(\alpha, \lambda)$ **(see Appendix J for details)**.
> > Thus, $n = 5$ already serves as a reasonable and convenient choice, even though larger multi-step horizons are also possible.
> > **If one wishes to further optimize performance, $(\alpha, \lambda)$ can easily be retuned for each trajectory length.**

---

> > > ### Author Response · Authors · 2025-11-21
> > > **Official Comment by Authors (3/3)**
> > >
> > > ---
> > >
> > > ## **Sensitivity of the conservatism parameter $\alpha$**
> > >
> > > **CPQL exhibits lower sensitivity to $\alpha$ across diverse datasets, and this trend is consistent with our theoretical characterization of the PQL operator.**
> > > As shown in Proposition 2, the fixed point of the PQL operator is $Q^{\lambda \pi_{\beta} + (1-\lambda)\pi_{\phi}}$.
> > > As $\lambda$ increases, this mixture policy becomes closer to the behavior policy $\pi_{\beta}$, and the influence of the learned policy $\pi_{\phi}$ on the fixed point diminishes.
> > > Consequently, the conservative parameter $\alpha$, which controls how strongly we penalize deviations from the behavior distribution, has a smaller marginal effect on the Q-values.
> > > In other words, because the backup places more weight on $\pi_{\beta}$ when $\lambda$ is large, CPQL becomes less sensitive to the precise choice of $\alpha$, which explains the robustness observed in Figure 1.
> > > In contrast, prior works [3, 4, 5] have pointed out that CQL ($\lambda = 0$) is extremely sensitive to the choice of $\alpha$, as even small changes can lead to significant performance differences.
> > > **In Appendix K (Figures 10 and 11)**, we report normalized scores for different values of the conservatism parameters $\alpha$ and $\lambda$ on *HalfCheetah* and *Hopper* tasks, which support the same trend.
> > >
> > > **We also emphasize that choosing an appropriate value of $\lambda$ depends on the environment, the type and quality of the offline dataset, and the scale of the conservative parameter $\alpha$.**
> > > Our theoretical analysis suggests that larger values of $\lambda$ can help when the behavior policy is truly near-optimal and the reward signal is sufficiently variable.
> > > However, *HalfCheetah-expert-v2* does not fully match this setting.
> > > First, **our analysis of the *HalfCheetah-expert-v2* dataset indicates that the behavior policy is not truly near-optimal** (the normalized score of the trajectories is about $85$, compared to about $100$ for *Hopper* and *Walker2d*).
> > > Second, **in *HalfCheetah*, episodes in the offline dataset never terminate early.**
> > > Once the agent reaches a stable running gait, the per-step reward is mostly determined by a relatively stable forward-velocity term, so the returns are much less variable than in *Hopper* or *Walker2d*.
> > > In this case, the PQL operator offers fewer advantages, because its multi-step targets have limited variability to smooth out, reducing the practical benefit of using larger $\lambda$.
> > > As a result, a large $\lambda$ may degrade performance in *Halfcheetah-expert-v2*.
> > > This explains why smaller $\lambda$ values work better overall in *HalfCheetah-expert-v2* and why the $(\lambda, \alpha)$ trend appears weaker than in *Hopper* and *Walker2d*.
> > >
> > >
> > > We are happy to include this discussion in Appendix C.3 of the revised version of the paper and thank reviewer MQiE for the helpful suggestion.
> > >
> > > ---
> > >
> > > **Reference**
> > >
> > > [1] Park, Seohong, et al. "OGBench: Benchmarking Offline Goal-Conditioned RL." The Thirteenth International Conference on Learning Representations.
> > >
> > > [2] Park, Seohong, et al. "Flow Q-learning." The Forty-Second International Conference on Machine Learning, 2025.
> > >
> > > [3] An, Gaon, et al. "Uncertainty-based offline reinforcement learning with diversified q-ensemble." Advances in neural information processing systems 34 (2021): 7436-7447.
> > >
> > > [4] Ghasemipour, Kamyar, Shixiang Shane Gu, and Ofir Nachum. "Why so pessimistic? estimating uncertainties for offline rl through ensembles, and why their independence matters." Advances in Neural Information Processing Systems 35 (2022): 18267-18281.
> > >
> > > [5] Tarasov, Denis, et al. "CORL: Research-oriented deep offline reinforcement learning library." Advances in Neural Information Processing Systems 36 (2023): 30997-31020.

---

### Official Review · Reviewer_W4Kk · 2025-10-31

**Soundness:** 3
**Presentation:** 3
**Contribution:** 3
**Rating:** 8
**Confidence:** 3

**Summary:**

The paper proposes Conservative Peng’s $Q(\lambda)$ (CPQL) for offline RL. The main idea is to replace the standard Bellman operator in the conservative Q-learning (CQL) objective with the Peng’s $Q(\lambda)$ operator. The authors present theoretical results showing that CPQL can achieve a tighter suboptimality gap than standard CQL, as the introduction of $\lambda$ could be used to mitigate the CQL 's excessive pessimism. They also provide extensive numerical experiments and ablation studies demonstrating the effectiveness of CPQL.

**Strengths:**

1. Interesting idea: Bringing Peng’s $Q(\lambda)$ from online RL to the offline setting (where the behavior policy is fixed offline) is interesting, and can potentially motivate future research.

2. Mitigates over-pessimism and improves robustness: Introducing $\lambda$ alongside $\alpha$ adds a practical control knob that balances conservatism and extrapolation risk, helping to mitigate excessive pessimism and showing reduced hyperparameter sensitivity in experiments.

3. Theory is solid and sound.

4. Comprehensive experiments and thorough ablations

**Weaknesses:**

1. Although the authors note in the last paragraph of Section 4.1 that we should focus on how an appropriately chosen $\lambda$ mitigates Q-value overestimation rather than on the additional bias it introduces, I would still suggest adding a discussion of the impact of this introduced bias. One potential direction in my mind is to analyze how the bias affects the suboptimality gap of the learned policy $\hat{\pi}$ (not just the mixture policy), but other forms of analysis are also welcome. In short, it would help to quantify how much we gain vs. how much we sacrifice, by introducing Peng's $Q(\lambda)$ into offline RL.

2. In Algorithm 1, it is not fully clear which policy is ultimately evaluated/deployed: the mixture $\lambda \pi_\beta +(1-\lambda)\pi_\phi$ or the learned actor $\pi_\phi$ itself. I suggest to clarify this explicitly in Algorithm 1

**Questions:**

Could you discuss more on the trade-offs introduced by incorporating Peng’s $Q(\lambda)$ into offline RL? In addition, please clarify the main limitations of the proposed framework in the "Conclusion" section.

---

> ### Author Response · Authors · 2025-11-21
> **Official Comment by Authors**
>
> Thank you very much for your positive evaluation of our paper and for your insightful and valuable feedback!
> We will address your questions below and make improvements to our paper based on your suggestions.
>
> ---
>
> ## **Clarify the updated policy in Algorithm 1**
>
> The evaluated/deployed policy is **the learned actor $\pi_{\phi}$**, not the mixture $\lambda \pi_{\beta} + (1-\lambda)\pi_{\phi}$. We use the Peng’s Q($\lambda$) (PQL) operator, whose the fixed point corresponds to $Q^{\lambda \pi_{\beta} + (1-\lambda)\pi_{\phi}}$, so the learned Q-function approximates the value of this mixture policy.
> In particular, $\pi_{\phi}$ is the one of the greedy policies with respect to $Q^{\lambda \pi_{\beta} + (1-\lambda)\pi_{\phi}}$.
> Thus, we clarify this explicitly in Algorithm 1 in the revised manuscript.
> We sincerely appreciate the reveiwer's careful reading and constructive feedback.
>
> ---
>
> ## **Trade-offs by incorporating Peng's Q($\lambda$) into offline RL**
>
> With additional experiments, **we analyze how the bias introduced by the PQL operator affects both the sub-optimality gap of the learned policy $\pi_{\phi}$ and the average Q-values of the critic network.**
> First, we focus on comparing SAC and PQL.
> In the table below, simply applying the PQL operator already improves the normalized return as $\lambda$ increases.
> We can expect that the bias introduced by PQL has a substantial impact in the offline RL setting.
>
> |Hopper-medium-replay-v2|SAC|PQL ($\lambda=0.3$)|PQL ($\lambda=0.7$)|
> |-|-|-|-|
> |Normalized Return|7.4 $\pm$ 0.5|24.9 $\pm$ 10.8|45.3 $\pm$ 27.9|
>
> To detect the role of $\lambda$, we constructed customized offline datasets in the *Walker2d* environment.
> Using SAC, we collected $200$K samples with the policy obtained at the point where the normalized score reached $20$.
> We continued training until the normalized score reached $100$, designating this policy as the optimal policy.
>
> |Toy example for Walker2d|Behavior policy|Optimal policy|SAC|PQL ($\lambda=0.3$)|PQL ($\lambda=0.7$)|
> |-|-|-|-|-|-|
> |Normalized Return|20|100|-0.5 $\pm$ 0.0|-0.5 $\pm$ 0.0|0.1 $\pm$ 0.1|
> |Average Q-values|$\approx$ 192.74|$\approx$ 267.76|1.6 $\times$ $10^{11}$ $\pm$ 5.2 $\times$ $10^{10}$|4.2 $\times$ $10^{10}$ $\pm$ 2.3 $\times$ $10^{9}$|438.7 $\pm$ 20.1|
>
> In the above table, SAC, PQL ($\lambda=0.3$), and PQL ($\lambda=0.7$) achieve low normalized returns.
> **SAC shows severe overestimation on the order of $10^{11}$ in the average Q-values, whereas increasing $\lambda$ increases the bias toward the behavior policy and leads to much more stable Q-values.**
> This shows the mitigation of over-optimistic value estimation with the PQL operator.
>
> However, this mitigation is not sufficient to solve offline RL by itself.
> Since the learned policy, weighted by $1-\lambda$, still queries OOD actions during policy evaluation, this can lead to an accumulation of extrapolation errors.
> Thus, it remains necessary to control the contribution of OOD actions in the Q-values, which is precisely the goal of conservative value estimation.
>
> To see how PQL and conservatism interact, we next compare CQL and CPQL on the same toy *Walker2d* setup:
> |Toy example for Walker2d|CQL|CPQL ($\lambda=0.3$)|CPQL ($\lambda=0.7$)|
> |-|-|-|-|
> |Normalized Return|45.9 $\pm$ 9.5|63.5 $\pm$ 8.9|81.3 $\pm$ 4.5|
> |Average Q-values|75.9 $\pm$ 5.4|129.6 $\pm$ 5.9|174.3 $\pm$ 8.2|
>
> We set the conservatism parameter $\alpha$ to $5.0$ for both CQL and CPQL.
> By adding the conservatism term to PQL, CPQL alleviates the overly conservative Q-values observed in CQL and outperforms CQL in terms of performance.
> This improvement occurs because, **under the same conservatism parameter, CPQL has mildly conservative Q-values and reduce the sub-optimality gap.**
> **This aligns with the theoretical insights in Theorems 1–3.**
>
> In summary, we gain:
> - A substantial reduction of overestimation and more stable Q-values through the bias introduced by the PQL operator.
> - Improved performance without excessive conservatism when the introduced bias is combined with a conservative penalty in CPQL, which alleviates the over-conservatism of CQL in offline RL.
> - Faster convergence (better sample efficiency)
>
> In contrast, we sacrifice:
> - Additional computational cost due to multi-step backups (with only minor overhead)
> - Potential performance degradation on low-quality datasets, where single-step updates can be preferable to multi-step operators.
>
> ---
>
> ## **Clarify the main limitations of CPQL**
>
> In the revised manuscript, we include a discussion of the limitations of CPQL in the Conclusion section, explicitly addressing the above weaknesses.

---

### Official Review · Reviewer_jSat · 2025-11-01

**Soundness:** 3
**Presentation:** 3
**Contribution:** 3
**Rating:** 6
**Confidence:** 4

**Summary:**

The paper proposes Conservative Peng’s Q(λ) (CPQL), an offline RL algorithm that replaces the single-step Bellman operator in CQL with the multi-step Peng’s Q(λ) operator. The authors prove conservative lower bounds on value estimates, establish policy-improvement guarantees w.r.t. the behavior policy, and derive a λ–α sub-optimality trade-off. This creates implicit behavior regularization without explicit policy constraints or extra networks, reducing both overestimation (from OOD actions) and over-pessimism.

**Strengths:**

1. First (to my knowledge) to pair PQL with conservative value estimation and analyze it theoretically for offline RL. Theorems 1–3 extend CQL-style results: (i) conservative lower bounds; (ii) policy improvement over behavior; (iii) an explicit λ–α trade-off with detailed proofs.

2. Algorithm 1 is straightforward; no behavior-policy estimation (unlike MCQ/CSVE/EPQ).

3. On MuJoCo, Adroit, AntMaze, CPQL is best or second-best on 22/29 tasks; sensitivity to α is substantially lower than CQL (clear practical win). Alternative multi-step operators and offline-to-online results reinforce the story.

**Weaknesses:**

1. Theory mostly treats the exact evaluation case; neural critics + replay introduce approximation error and potential instability (classic issue with λ-returns). Provide a finite-sample / approximation note (even informal), or an experiment tracking Bellman error vs. λ and critic drift under function approximation. A short stability plot (loss, TD error, divergence events) would help.

2. Recent multi-step offline methods (e.g., Uni-O4) are conceptually close. Either add Uni-O4 (or the closest available multi-step competitor) to Table/Fig. 2 or clearly justify incompatibility. A paragraph contrasting CPQL’s no-IS λ-returns vs. methods relying on IS/retrace would clarify the novelty.

3. Multi-step targets add compute; the paper omits time-to-X (e.g., to 1M steps) vs. CQL/IQL. Add a runtime table and learning-curve area-under-curve (AUC). If CPQL converges faster for similar wall-clock, that strengthens the practical claim.

**Questions:**

1. Algorithm 1 line 7 subtracts γⁿ α_td log π, but α_td = 0 except at the final step. What role does this term play? Would omitting it entirely affect stability or theoretical guarantees?

2. What is the runtime cost of CPQL relative to CQL and IQL? Please report wall-clock time (to 1 M steps) and sample-efficiency comparisons.

3. Are there tasks where single-step conservative Bellman updates (λ = 0) outperform CPQL? If so, how can practitioners detect when multi-step λ > 0 is harmful?

4.Could you compare to recent multi-step offline RL algorithms such as Uni-O4 (2024) or clarify conceptual differences?

---

> ### Author Response · Authors · 2025-11-21
> **Official Comment by Authors (1/2)**
>
> We thank the reviewer for taking the time to review our paper and for recognizing its value! Regarding concerns and questions, we provide the detailed responses below.
>
> ---
>
> ## **Line 7 in Algorithm 1**
>
> **Removing the last-step entropy term will make the learned Q-function sharper and the resulting policy more unstable in practice.**
> CPQL is based on the Soft Actor-Critic (SAC) algorithm.
> Line 7 in Algorithm 1 shows the multi-step soft TD target, where the coefficient $\alpha_{\text{td}}$ controls the entropy bonus that is added to the target Q-function, analogous to SAC.
> However, we found that adding this entropy term at every step makes the target scale grow with the horizon, which destabilizes learning, consistent with the observations of [1].
> For this reason, we set $\alpha_{\text{td}} = 0$ at all intermediate steps and $\alpha_{\text{td}} = \alpha_{\text{pol}}$ only at the last step, so that $\gamma^n \alpha_{\text{td}}\log\pi(s_n,a_n)$ reduces to a standard terminal entropy bouns while avoiding cumulative entropy inflation.
>
> This design choice does not affect our theoretical guarantees (Theorems 1–3).
> However, removing the last-step entropy term would turn the backup into a standard Peng’s Q($\lambda$) target without entropy regularization.
> This would remove a useful source of smoothing in the value targets and is likely to make the learned Q-function sharper and the resulting policy more brittle in practice.
>
> ---
>
> ## **Runtime cost and sample-efficiency of CPQL relative to CQL and IQL**
>
> **CPQL achieves substantially better performance than CQL and IQL for similar wall-clock (normalized by the number of gradient steps)**.
> **In Appendix C.4**, we already address the computational cost of CQL and CPQL.
> We add the runtime of IQL in the table below.
> In summary, we run a comparison on the *Hopper-Medium-v2* dataset using a single GeForce RTX 3090 GPU.
> We measure **the average runtime per epoch (1K gradient steps), excluding the evaluation step**.
>
> ||IQL|CQL|CPQL(n=5)|
> |-|-|-|:-:|
> |Epoch runtime|12.8 sec|40.6 sec|42.4 sec|
>
> Because the runtime per 1K gradient steps differs by approximately a factor of **3.2** between IQL and CQL/CPQL, we compare their sample efficiency under a comparable wall-clock budget.
> We therefore report **the normalized scores** after **1M gradient steps** for IQL, **0.32M** for CQL, and **0.3M** for CPQL.
> The normalized scores of IQL are taken from [2].
>
> |Normalized Scores per Tasks|IQL (1M grad steps)|CQL (0.32M grad steps)|CPQL (0.3M grad steps)|
> |:-|:-:|:-:|:-:|
> | halfcheetah-r|13.1|25.1 $\pm$ 1.6 | **37.5 $\pm$ 1.2**  |
> | hopper-r|7.9|8.2 $\pm$ 1.1| **20.2 $\pm$ 0.2**  |
> | walker2d-r|5.4| 2.5 $\pm$ 5.5| **13.4 $\pm$ 8.4**  |
> | halfcheetah-m|47.4|   48.4 $\pm$ 0.3| **64.4 $\pm$ 1.0**  |
> | hopper-m|66.2|62.6 $\pm$ 6.1| **102.0 $\pm$ 1.6** |
> | walker2d-m|78.3|**81.8 $\pm$ 1.9** | **82.0 $\pm$ 2.4**  |
> | halfcheetah-m-r|44.2| 46.6 $\pm$ 0.3| **59.1 $\pm$ 1.7**  |
> | hopper-m-r|94.7|99.6 $\pm$ 1.9| **103.4 $\pm$ 0.6** |
> | walker2d-m-r|73.8|84.7 $\pm$ 4.5| **95.9 $\pm$ 3.2**  |
> | halfcheetah-m-e|**86.7** |70.3 $\pm$ 9.2|   76.2 $\pm$ 10.9   |
> | hopper-m-e| **91.5** |88.0 $\pm$ 22.0 | **90.4 $\pm$ 23.4** |
> | walker2d-m-e| **109.6**|**110.4 $\pm$ 0.2** | **110.0 $\pm$ 0.7** |
>
> CPQL achieves substantially better performance than CQL and IQL when normalized by the number of gradient steps, which further strengthens the practical claim of our algorithm.
>
> ---
> ## **When is the single-step preferable to the multi-step?**
>
> There can be instances where single-step conservative updates may outperform multi-step conservative updates.
> In Tables 1 and 4, the *Hopper-random-v2* and *HalfCheetah-medium-v2* datasets achieve higher normalized returns with $\lambda = 0$ than with $\lambda > 0$.
> However, we emphasize that **our algorithm can subsume the single-step operator by setting $\lambda = 0$** in any environment.
> More generally, choosing an appropriate value of $\lambda$ requires considering several factors, including the environment, the type of offline dataset, and whether the conservative parameter $\alpha$ is set to be relatively small or large.
> We observe the following consideration across environments, datasets, and $\alpha$:
> - **Low-quality trajectories**  (e.g., random datasets):
> When the offline dataset is very noisy or suboptimal, performance tends to be better as $\lambda$ approaches $0$.
> Because multi-step backups propagate errors from low-quality transitions and thereby degrade performance.
> - **Weak conservatism**:
> When $\alpha$ is set too small, using a larger $\lambda$ helps reduce overestimated Q-values,
> as multi-step targets provide stronger regularization by aggregating information from the behavior policy along the trajectory.
> - **Overly strong conservatism**:
> When $\alpha$ is excessively large, performance becomes relatively insensitive to the choice of $\lambda$, with different values of $\lambda$ yielding similar returns of the behavior policy (Theorem 2).

---

> > ### Author Response · Authors · 2025-11-21
> > **Official Comment by Authors (2/2)**
> >
> > ---
> > ## **Comparison to Uni-O4**
> >
> > Uni-O4 [3] has proposed an offline policy evaluation method for safe multi-step policy improvement, which uses a multi-step objective based on an approximate model and fitted Q evaluation.
> > However, our algorithm (CPQL) differs conceptually from Uni-O4 in several key aspects.
> >
> > - CPQL uses the multi-step operator (PQL operator) over offline trajectories collected from the real environment, whereas Uni-O4 computes its multi-step objective, $\mathbb{E}\_{(s,a)\sim(\hat{T}, \pi)}\left[\sum\_{t=0}^{H-1}\hat{Q}\_{\tau}(s_{t},a_{t})\right]$, by **rolling out an approximate transition model (AM) $\hat{T}$** trained via maximum-likelihood estimation on the offline dataset and **aggregating fitted-Q estimates** along the simulated trajectories (AM-Q).
> > - CPQL calculates the multi-step return in the critic update to obtain a conservative value estimate for control, whereas Uni-O4 uses AM-Q only as **an offline policy evaluation oracle to determine whether a PPO-style policy update should be accepted**, thereby implementing safe multi-step policy improvement.
> > - Additionally, Uni-O4 optimizes an ensemble behavior-cloning model to approximate the behavior policy.
> > The learned behavior policy is used to keep policy updates close to the data-support region and to mitigate distributional shift when querying the AM-Q estimator, which is why Uni-O4 resorts to an ensemble of behavior-cloning models to **better capture the behavior policy and stabilize offline policy evaluation**.
> > In contrast, CPQL does not require learning the behavior policy and estimating the transition model.
> >
> > Consequently, although both methods leverage multi-step information, CPQL is a conservative value-estimation algorithm based on a multi-step operator, whereas Uni-O4 uses a model-based offline policy evaluation method (AM-Q) for safe multi-step policy improvement.
> > We clarify these conceptual differences in the additional related work section **(Appendix H)** of the revised manuscript.
> >
> >
> > ---
> >
> > ## **Provide an experiment tracking TD loss and Q-values**
> >
> > To address the reviewer’s suggestion about approximation error and stability under function approximation, we add new analysis for the TD loss, Q-values, and normalized scores in **Appendix I (Figures 7–9)**.
> > In these experiments, we track, for CQL ($\lambda = 0$) and CPQL with $\lambda \in \\{0.1, 0.3, 0.5, 0.7, 0.9\\}$ on *medium*, *medium-replay*, and *medium-expert* for *Walker2d*:
> > - TD loss (Figure 7),
> > - Average Q-values of the critic (Figure 8)
> > - Normalized return during training (Figure 9)
> >
> > We set the conservatism parameter $\alpha$ to $1.0$.
> > We observe that for $\lambda = 0$ (CQL) and $\lambda = 0.1$ (CPQL), the TD loss and the average Q-values grow rapidly and eventually diverge on all three datasets.
> > On Walker2d-medium-expert, even $\lambda = 0.3$ shows a clear increase in TD loss and Q-values.
> > These results indicate that a small $\lambda$ is insufficient to mitigate the TD loss and the overestimation of Q-values under function approximation.
> >
> > As $\lambda$ becomes larger (e.g., $\lambda = 0.7, 0.9$), the TD loss increases and becomes noisier, which is consistent with the classic behavior of multi-step TD methods: using longer trajectories increases the variance of the targets and propagates approximation errors further.
> > **Nevertheless, even for large $\lambda$ the curves remain bounded and move toward a convergent regime rather than exploding.**
> > This suggests that CPQL can simultaneously prevent unbounded overestimation and avoid overly conservative value estimates by appropriately utilizing offline trajectory information.
> >
> > Overall, these results the view that **CPQL exhibits the usual bias–variance trade-off of $\lambda$-returns under function approximation, while still offering a reasonably wide range of $\lambda$ values in which it improves performance and mitigates critic divergence.**
> > Within this range, CPQL maintains stable TD errors and Q-value estimates.
> >
> > ---
> >
> > **Reference**
> >
> > [1] Kozuno, Tadashi, et al. "Revisiting Peng’s Q ($λ$) for Modern Reinforcement Learning." International Conference on Machine Learning. PMLR, 2021.
> >
> > [2] Kostrikov, Ilya, Ashvin Nair, and Sergey Levine. "Offline Reinforcement Learning with Implicit Q-Learning." International Conference on Learning Representations.
> >
> > [3] Lei, Kun, et al. "Uni-O4: Unifying Online and Offline Deep Reinforcement Learning with Multi-Step On-Policy Optimization." ICLR. 2024.

---

### Author Response · Authors · 2025-11-21

We thank the reviewers for recognizing our contribution and for their thoughtful and detailed feedback.
We have updated a revised version of the paper, with changes highlighted in blue.
In particular, we have added a detailed discussion in the Appendix of additional experiments on TD loss, Q-values, trajectory length, the conservatism parameter, and other baselines.

---

### Comment · Area_Chair_qDpG · 2025-11-23
**Subject: Follow-up Reviews Required to Proceed**

Dear Reviewers,
The authors have submitted their rebuttal, and we now require your follow-up assessments to move the decision process forward. Please review the authors’ responses and update your evaluations accordingly.
Your prompt follow-up is necessary for us to finalize the meta-review.
Kindly submit your updates as soon as possible.
Best,
Area Chair

---

### Meta-Review · Area_Chair_v7v6 · 2026-01-06

**Summary:**

CPQL is a clean and well-motivated extension of CQL that replaces the single-step Bellman backup with Peng’s Q(λ) in order to use multi-step trajectory information as implicit behavior regularization, aiming to reduce both OOD overestimation and CQL’s over-pessimism without learning a behavior policy or adding auxiliary networks. The paper backs this with solid CQL-style theory (lower bounds, policy improvement over behavior, λ–α trade-off), and strong empirical results across D4RL suites, with additional diagnostics, runtime comparisons, Uni-O4 discussion, MCQ comparisons, and OGBench/offline-to-online results provided in the rebuttal.

**Reviewer Concerns:**

Most key concerns are addressed convincingly: the authors added stability diagnostics under function approximation (TD loss/Q-value tracking vs. λ/α), clarified the entropy term in the multi-step soft target and why it is only applied at the last step, provided runtime comparisons showing CPQL is close to CQL in wall-clock and substantially stronger in performance under matched compute, and clarified conceptual differences to Uni-O4 (model-based evaluation vs. trajectory-based multi-step conservative control). Additional experiments and discussion also quantify the trade-offs of λ (when λ=0 can be preferable on low-quality data) and strengthen the “reduced α sensitivity” claim across more tasks, while comparisons to MCQ and WSRL-style fine-tuning reduce concerns about over-pessimism and offline-to-online competitiveness.

**Reviewer Scores:**

The panel appears to trend positive overall: one reviewer is clearly strong accept, and multiple marginal reviews were driven by missing comparisons/diagnostics and practical cost reporting that were directly addressed in the rebuttal with concrete tables and plots. The remaining skepticism is mostly about “novelty as integration” and scope (broader benchmarks), but the added OGBench results, MCQ-with-PQL experiment, and the explicit λ–α theoretical trade-off materially raise the contribution beyond a superficial swap of targets. Net, I would expect borderline reviewers to move upward or at least remain supportive given the strengthened evidence.

---

### Decision · Program_Chairs · 2026-01-26

Accept (Poster)